# AliO: Output Alignment Matters in Long-Term Time Series Forecasting

**Kwangryeol Park**
Artificial Intelligence Graduate School
Ulsan National Institute of Science & Technology (UNIST), South Korea
pkr7098@unist.ac.kr

**Jaeho Kim**[*]
Artificial Intelligence
Korea University, South Korea
kjh3690@korea.ac.kr

**Seulki Lee**[†]
Department of Computer Science and Engineering
Ulsan National Institute of Science & Technology (UNIST), South Korea
seulki.lee@unist.ac.kr

## Abstract

Long-term Time Series Forecasting (LTSF) tasks, which leverage the current data sequence as input to predict the future sequence, have become increasingly crucial in real-world applications such as weather forecasting and planning of electricity consumption. However, state-of-the-art LTSF models often fail to achieve prediction output alignment for the same timestamps across lagged input sequences. Instead, these models exhibit low output alignment, resulting in fluctuation in prediction outputs for the same timestamps, undermining the model's reliability. To address this, we propose AliO (Align Outputs), a novel approach designed to improve the output alignment of LTSF models by reducing the discrepancies between prediction outputs for the same timestamps in both the time and frequency domains. To measure output alignment, we introduce a new metric, TAM (Time Alignment Metric), which quantifies the alignment between prediction outputs, whereas existing metrics such as MSE only capture the distance between prediction outputs and ground truths. Experimental results show that AliO effectively improves the output alignment, i.e., up to 58.2% in TAM, while maintaining or enhancing the forecasting performance (up to 27.5%). This improved output alignment increases the reliability of the LTSF models, making them more applicable in real-world scenarios. The code implementation is on the GitHub repository[3].

## 1 Introduction

The task of long-term time series forecasting (LTSF) is essential in various fields such as prediction of electricity demand [13], weather forecasting [37], health data [20], and so on. Recently, deep neural network models [40, 30, 33, 43] have shown strong performance in predicting long-term time series

---

[*]Work done at UNIST

[†]Corresponding author

[3]https://github.com/eai-lab/AliO

39th Conference on Neural Information Processing Systems (NeurIPS 2025).

based on historical information, which typically aim to minimize the error between the prediction output and the ground-truth sequence as a regression task. However, in real-world applications, merely minimizing the prediction error in LTSF is insufficient. It is equally important to ensure that the forecasting model generates consistent (aligned) prediction outputs for overlapping timestamps across lagged input sequences. For example, Fig. 1 illustrates a scenario in which a trained forecasting model predicts future electricity usage in two instances: (1) for the period from April to November (Prediction 1, purple), using input data from January to March, and (2) for the period from May to December (Prediction 2, yellow), using input data from February to April, where the forecasting periods overlap between May and November. A reliable model should provide consistent predictions for these overlapping months, regardless of the partially differing input sequences. If the model produces inconsistent predictions on the electricity usage for the same timestamps (i.e., May to November) between two input sequences, it could result in significant time and financial costs for rescheduling budget allocation and undermine the reliability of the predictions.

We refer to this phenomenon as the *output alignment* problem, which has not been adequately acknowledged and addressed by existing LTSF studies [25, 7, 18], despite its significance and substantial impact on real-world applications. To the best of our knowledge, state-of-the-art LTSF models [37, 33, 30] often fail to maintain the prediction output consistency, and none of existing works has explicitly recognized or attempted to address this inconsistency in LTSF tasks.

In this paper, we present *AliO (Align Outputs)*, a novel method designed to enhance output alignment in LTSF models. For the first time, AliO enables LTSF models to produce consistent predictions for overlapping timestamps across lagged input sequences. By aligning predictions for overlapping timestamps through the minimization of discrepancies in both time and frequency domains, AliO improves output consistency even when input sequences lagged. AliO achieves this by directing the model's predictions toward the ground-truth sequences while simultaneously minimizing discrepancies across multiple predictions obtained from a set of lagged input sequences. It allows AliO to integrate seamlessly with the model's forecasting objectives, such as regression loss (e.g., MSE), without adding implementation complexity or requiring modifications to the model. As a result, AliO enhances the reliability of forecasts by improving output alignment (consistency), while maintaining or even improving overall forecasting performance.

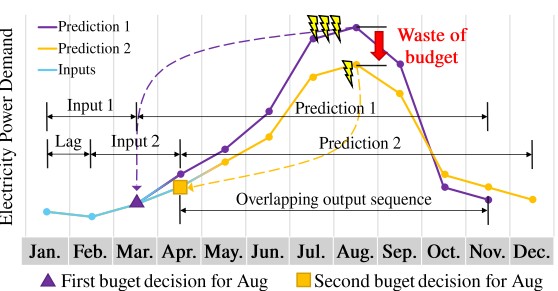

Figure 1: An example of low output alignment between two predictions on future electricity usages: (1) Prediction 1 (purple) forecast spanning from Apr to Nov and (2) Prediction 2 (yellow) forecast spanning from May to Dec. These two predictions are generated from two partially overlapping input sequences—(1) one from Jan to Mar (Input 1), and (2) the other from Feb to Apr (Input 2). The inconsistency between these two prediction outputs over the same timestamps (i.e., May to Nov) leads to differing budget allocation plannings for electricity power consumption in Aug, resulting in time and financial waste due to rescheduling resource allocation.

This represents a significant advancement over existing methods [37, 40] that focus solely on minimizing the forecasting objective without considering prediction output alignment.

To quantify output alignment, which represents the consistency of a model's predictions across lagged input sequences, we propose a new metric, Time Alignment Metric (TAM). TAM quantitatively assesses the model's output alignment by measuring discrepancies between predictions for overlapping timestamps for multiple input sequences. To the best of our knowledge, the proposed TAM is the first metric designed to measure the output alignment.

We experiment with AliO on representative LTSF tasks, including ETT{h1, h2, m1, m2}, Electricity (ECL), Traffic, Weather, and ILI dataset [37], using various state-of-the-art LTSF models such as CycleNet [27], GPT4S [43], iTransformer [30], PatchTST [33], TimesNet [36], DLinear [40] and Autoformer [37]. The evaluation results demonstrate that AliO effectively aligns predictions over overlapping timestamps, i.e., improving TAM up to 58.2%, while maintaining or enhancing

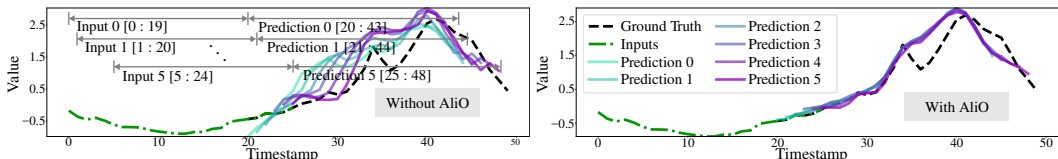

Figure 2: (**Left**) Six prediction outputs from the DLinear [40] model trained with regression loss (MSE) only, forecasting a length-24 sequence for ILI data [37], where six input sequences (green dotted line) are lagged one timestamp apart. (**Right**) Six predicted sequences from DLinear trained with AliO, under the same conditions. The predicted sequences exhibit improved output alignment, showing a closer match to the ground truth for all six predictions, which significantly reduces the prediction shifts observed when using regression loss alone.

forecasting regression accuracy up to 17.4% measured in MSE. The detailed descriptions on additional datasets and experimental configurations are provided in Secs. G to I.

## 2  Motivation

A system with low prediction consistency leads to user distrust and increased costs. For instance, studies on the consistency of weather forecasts have reported cases where consistent forecasts lead to greater trust in the system, whereas inconsistent forecasts result in user distrust [5, 31, 4]. Furthermore, accroding to research in non-weather domains, consumers tend to perceive consistency among multiple estimates from the same system as a signal of skill [11, 5, 3]. The inconsistency in predictions necessitates replanning, which consumes unnecessary organizational time. The resulting resource reallocation leads to the investment of more resources, contrary toe th original goal of Rolling forecasting, which aims for resource savings [16, 14]. This can be described as an additional sunk cost [35], as frequent fluctuations in predictions can induce irrational decision-making and result in lost opportunity costs [35, 6].

While modern Long-Term Times Series Forecasting (LTSF) models demonstrate high accuracy, research on the aforementioned consistency has been insufficient. Consequently, we examined the consistency of existing models through rolling forecasting on representative LTSF datasets. The left panel of Fig. 2 shows the prediction results of DLinear [37] on the ILI dataset [37] using only regression loss (MSE), revealing a significant lack of consistency. In contrast, the right panel, where AliO is applied, shows that consistency is improved while accuracy is maintained. As can be seen in Sec. I, this is a phenomenon observed across other models and datasets.

Although both consistency and accuracy are crucial for addressing users' psychological and economic concerns, our experiments revealed that while modern LTSF models achieve high accuracy, they fall short in terms of consistency. AliO, as a loss function used in conjunction with regression loss, aims to achieve two goals: maintaining the high accruacy of existing LTSF models while improving their prediction consistency. Furthermore, to quantify consistency, we propose the Time Alignment Metric (TAM). Regression metrics such as MSE, MAE, and DTW [34] represent accuracy, which is the distance between predictions and ground truth. Therefore, they have limitations in measuring consistency, which is a measure of performance between predictions. Complementing this, TAM measures the distance between predictions, serving as an indicator that represents consistency.

Through Sec. 5 and Sec. I, we show that AliO successfully achieves the dual goals of maintaining/improveing the high accuracy of existing models and enhancing consistency. Additionally, in **??**, we experimentally verify the relationship between the inconsistency value (TAM) of existing models and the MSE improvement in models trained with AliO. To the best our knowledge, this is the first study to address the improvemnet of consistency in LTSF.

## 3  Time alignment metric (TAM)

We first define the concept of output alignment and then introduce Time Alignment Metric (TAM), which enables a quantitative assessment of output alignment.

**Definition 3.1.** (*Output Alignment*) *Output alignment* refers to the property wherein the prediction outputs of a model, derived from a set of lagged input sequences, exhibit consistent alignment, characterized by uniform patterns across overlapping timestamps in the prediction outputs.

For instance, as illustrated in Fig. 2, the six prediction outputs for the overlapping timestamps (i.e., from 25 to 43) in the left figure demonstrate a lower degree of output alignment. Conversely, the six prediction outputs in the right figure exhibit similar and consistent patterns, demonstrating a high degree of output alignment. To quantify the output alignment (Theorem 3.1) in model predictions, we propose Time Alignment Metric (TAM). We begin by defining the necessary concepts, including lagged input and output sequences, and overlapping output sequences, as provided below.

**Definition 3.2.** (*Input and Output Sequence*) Given a time-series sequence $\boldsymbol{X} \in \mathbb{R}^{c \times d}$, where $c$ is the input channel, and $d$ is the length of the sequence, respectively, the $n$-th *input sequence* $\boldsymbol{X}^n \in \mathbb{R}^{c \times d'}$ of the length $d' \leq d$ is defined as the segment $\boldsymbol{X}_{s:s+d'-1}$, where $s$ is the starting timestamp, and $s + d' - 1$ is the ending timestamp.

Then, taking $\boldsymbol{X}^n \in \mathbb{R}^{c \times d'}$ as input, $f(\boldsymbol{X}^n; \boldsymbol{\theta}) = \boldsymbol{Y}^n \in \mathbb{R}^{c' \times h}$ is defined as the prediction *output sequence* of the length $h$ provided by the forecasting model $f$, where $c'$ is the output channel, and $\boldsymbol{\theta}$ is the model parameter. Having the input sequence $\boldsymbol{X}^n$ and the corresponding output sequence $\boldsymbol{Y}^n$ of the model $f$, the lagged input and output sequences, $\boldsymbol{X}^{n+1}$ and $\boldsymbol{Y}^{n+1}$, are derived by shifting their timestamps by the lag parameter $l$, as described below.

**Definition 3.3.** (*Lagged Input and Output Sequence*) Given the $n$-th input sequence $\boldsymbol{X}^n \in \mathbb{R}^{c \times d'}$ of the segment $\boldsymbol{X}_{s:s+d'-1}$, the sequence $\boldsymbol{X}^{n+1} \in \mathbb{R}^{c \times d'}$ of the segment $\boldsymbol{X}_{s+l:s+d'+l-1}$ is defined as the *lagged input sequence* of $\boldsymbol{X}^n$ with $l$ being the lag parameter.

Then, given the lagged input sequence $\boldsymbol{X}^{n+1} \in \mathbb{R}^{c \times d'}$, the prediction output of the length $h$, $f(\boldsymbol{X}^{n+1}; \boldsymbol{\theta}) = \boldsymbol{Y}^{n+1} \in \mathbb{R}^{c' \times h}$, is defined as the model $f$'s *lagged output sequence*. From $\boldsymbol{Y}^n$ and $\boldsymbol{Y}^{n+1}$, the overlapping output sequences $\boldsymbol{P}^n$ and $\boldsymbol{P}^{n+1}$ are derived to compute TAM, as follows.

**Definition 3.4.** (*Overlapping Output Sequence*) Given two output sequences $\boldsymbol{Y}^n \in \mathbb{R}^{c' \times h}$ and $\boldsymbol{Y}^{n+1} \in \mathbb{R}^{c' \times h}$ of the model $f$ generated from $\boldsymbol{X}^n$ and $\boldsymbol{X}^{n+1}$ as input, respectively, $\boldsymbol{P}^n \in \mathbb{R}^{c' \times h'}$ and $\boldsymbol{P}^{n+1} \in \mathbb{R}^{c' \times h'}$ are defined as the *overlapping output sequences* between $\boldsymbol{Y}^n$ and $\boldsymbol{Y}^{n+1}$, whose segments are given by $\boldsymbol{Y}^n_{1+l:h}$ and $\boldsymbol{Y}^{n+1}_{1:h-l}$, respectively, where $h' = h - l$ is the overlapping length.

From these, Time Alignment Metric (TAM) is defined to quantify output alignment by evaluating the consistency between overlapping output sequences $\boldsymbol{P}^n$ and $\boldsymbol{P}^{n+1}$, as follows.

**Definition 3.5.** (*TAM; Time Alignment Metric*) Given $N$ overlapping output sequences $\boldsymbol{P}^n \in \mathbb{R}^{c' \times h'}$ for $n = 1, 2, \ldots, N$, where the input and output sequences, $\boldsymbol{X}^{n+1}$ and $\boldsymbol{Y}^{n+1}$, are offset from $\boldsymbol{X}^n$ and $\boldsymbol{Y}^n$ by the lag $l$, $TAM_N$ (*Time Alignment Metric*) is calculated as the average of the MAE values between all pairs of $\{\boldsymbol{P}^n, \boldsymbol{P}^m\}$, which is given by:

$$TAM_N \triangleq \frac{(N-1)N}{2} \sum_{n=1}^{N-1} \sum_{m=n+1}^{N} \frac{|\boldsymbol{P}^n - \boldsymbol{P}^m|_1}{h'} \tag{1}$$

TAM offers a distinct advantage over traditional forecasting regression metrics, such as the MSE or MAE between model outputs and ground truth, which only measure forecasting performance and fail to capture prediction fluctuations across different predictions. Unlike these metrics, TAM evaluates the distances between overlapping predictions, allowing for an assessment of how predictions evolve over time. For instance, regression metrics do not determine whether overlapping predictions remain consistent or exhibit smooth transitions, making it challenging to assess data-model robustness. In contrast, TAM computes the distance between overlapping predictions, providing a robust measure of data-model consistency. A lower TAM value reflects improved alignment and data-model robustness.

## 4 Improving output alignment (AliO)

To enhance the output alignment of LTSF models, we propose AliO (Align Outputs), which minimizes the discrepancy between overlapping predictions (Theorem 3.4). AliO is designed to simultaneously

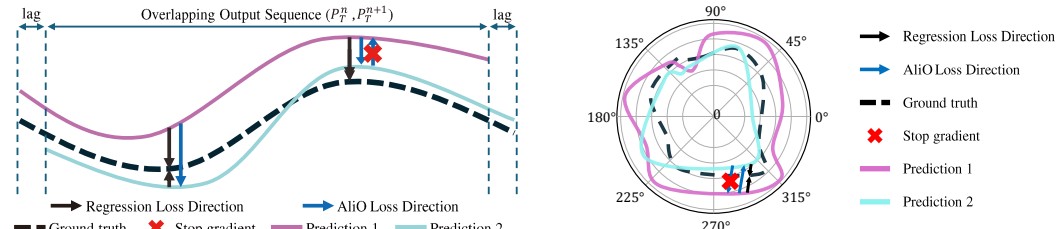

Figure 3: **Output alignments in both time and frequency domains. (Left)** In the time domain, AliO minimizes the difference between two predictions in the overlapping timestamps, i.e., $\boldsymbol{P}_T^n$ and $\boldsymbol{P}_T^{n+1}$ ($T$ means time domain). To preserve the regression loss direction while improving the output alignment, the regression pulling (Sec. 4.2) is applied through the stop-gradient operation to each time point, ensuring that the overall loss is calculated in the direction towards the ground truth. **(Right)** In the frequency domain, AliO aligns both the phase and amplitude components of $\boldsymbol{P}_F^n$ and $\boldsymbol{P}_F^{n+1}$ (the transformed frequency domain of $\boldsymbol{P}_T^n$ and $\boldsymbol{P}_T^{n+1}$), applied with the stop-gradient operation. The figure is represented in polar coordinates, where the angle indicates the phase, and the distance from the center point corresponds to the amplitude

.

achieve two key objectives: (1) improving output alignment by reducing discrepancies between the overlapping predictions, and (2) maintaining the model's forecasting regression performance by aligning the model's prediction outputs with the ground truths. The two objectives are achieved through the technique that we call regression pulling with the stop-gradient operation, which ensures that the forecasting regression loss remains unaffected while improving output alignment simultaneously.

Given two overlapping predictions, $\boldsymbol{P}^n = f(\boldsymbol{X}^n; \boldsymbol{\theta})_{1+l:h}$ and $\boldsymbol{P}^{n+1} = f(\boldsymbol{X}^{n+1}; \boldsymbol{\theta})_{1:h-l}$ defined in Theorem 3.4, AliO aligns these predictions such that $\boldsymbol{P}^n \simeq \boldsymbol{P}^{n+1}$ for $n = 1, 2, \ldots, N$ by minimizing the following objective with respect to the model $f$'s parameter $\boldsymbol{\theta}$, as:

$$\min_{\boldsymbol{\theta}} D(f(\boldsymbol{X}^n; \boldsymbol{\theta})_{1+l:h}, f(\boldsymbol{X}^{n+1}; \boldsymbol{\theta})_{1:h-l}) \tag{2}$$

where $D$ denotes a distance function and subscripts denote the segment ranges of the model $f$'s prediction output sequences, i.e., $1 + l : h$ and $1 : h - l$ correspond to the same timestamps for $\boldsymbol{X}^n$ and $\boldsymbol{X}^{n+1}$, respectively. Eq. (2) can be easily extended to handle non-consecutive overlapping output sequences, enabling its application to TAM (Theorem 3.5). AliO aligns the overlapping predictions $\boldsymbol{P}^n$ and $\boldsymbol{P}^{n+1}$ in both time and frequency domains, as illustrated in Fig. 3 with further details provided in the following subsections. The algorithmic procedure of AliO is summarized in Alg. 1.

### 4.1 Time domain alignment

To enhance the temporal alignment, AliO aligns the overlapping predictions, $\boldsymbol{P}_T^n$ and $\boldsymbol{P}_T^m$ for $n, m = 1, 2, \ldots, N$, in the time domain, where the subscript $T$ denotes the time domain. This alignment can be achieved using a distance function, $D_T$, such as MSE or Dynamic Time Warping (DTW) [34]. The computed distance is back-propagated to encourage the model $f$ to produce overlapping predictions that align with each other. However, solely aligning the overlapping predictions by minimizing the time domain alignment loss $\mathcal{L}_T = D_T(\boldsymbol{P}_T^n, \boldsymbol{P}_T^m)$ may lead to both prediction outputs deviating from the ground truth, degrading the forecasting regression performance of the model $f$.

### 4.2 Regression pulling (RegPull)

To maintain forecasting regression performance while aligning predictions, we propose *Regression Pulling (RegPull)*, which identifies which prediction output points of $\boldsymbol{P}_T^n$ and $\boldsymbol{P}_T^m$ are relatively further from the ground truth than the other at the same timestamp, indicated by the index variable $idx_T$, as shown on line (1) in Alg. 1. By applying the stop-gradient operation, denoted as $sg(\cdot)$, to the prediction output points farther from the ground truth, the time domain alignment loss $\mathcal{L}_T$ pulls these distant points closer to the ground truth, aligning its optimization direction with that of the forecasting regression loss and reinforcing to minimize the regression loss. Consequently, regression pulling

**Algorithm 1** The procedure of AliO. $\odot$ denotes element-wise multiplication. $FFT(\cdot, \cdot, \cdot)$ returns the frequency domain representation of each signal sequences, $sg(\cdot)$ is stop-gradient operator.

---

**Input:** The number of predictions $N$, consecutive predictions $\boldsymbol{Y}^n$ with at time lag of $l$, and their ground truth $\hat{\boldsymbol{Y}}^n$, where $n \in [1, N]$, and distance function $D_T$ and $D_F$ for the time and frequency domain, respectively. The subscript $T$ and $F$ denotes the time and frequency domain, respectively.
**Output:** The time domain alignment loss $\mathcal{L}_T$ and frequency domain alignment loss $\mathcal{L}_F$.
Initialize $\mathcal{L}_t = 0, \mathcal{L}_f = 0, count = 0$
**for** $n = 1$ **to** $N - 1$ **do**
   **for** $m = n + 1$ **to** $N$ **do**
      $gap = |m - n| \times l$
      $(\boldsymbol{P}_T^n, \boldsymbol{P}_T^m, \boldsymbol{GT}_T) = (\boldsymbol{Y}_{gap:}^n, \boldsymbol{Y}_{:-gap}^m, \hat{\boldsymbol{Y}}_{gap:}^n)$
      $idx_T = \text{Index}(|\boldsymbol{P}_T^n - \boldsymbol{GT}_T| > |\boldsymbol{P}_T^m - \boldsymbol{GT}_T|)$                  (1)
      $\mathcal{L}_t = \mathcal{L}_t + D_T(\boldsymbol{P}_{T, i \in idx_T}^n, sg(\boldsymbol{P}_{T, i \in idx_T}^m)) + D_T(sg(\boldsymbol{P}_{T, i \notin idx_T}^n), \boldsymbol{P}_{T, i \notin idx_T}^m)$   [RegPull]   (2)
      $(\boldsymbol{P}_F^n, \boldsymbol{P}_F^m, \boldsymbol{GT}_F) = FFT(\boldsymbol{P}_T^n, \boldsymbol{P}_T^m, \boldsymbol{GT}_T)$                    (3)
      $idx_F = \text{Index}(|\boldsymbol{P}_F^n - \boldsymbol{GT}_F| > |\boldsymbol{P}_F^m - \boldsymbol{GT}_F|)$                (4)
      $\mathcal{L}_f = \mathcal{L}_f + D_F(\boldsymbol{P}_{F, i \in idx_F}^n, sg(\boldsymbol{P}_{F, i \in idx_F}^m)) + D_F(sg(\boldsymbol{P}_{F, i \notin idx_F}^n), \boldsymbol{P}_{F, i \notin idx_F}^m)$   [RegPull]   (5)
      $count = count + 1$
   **end for**
**end for**
$(\mathcal{L}_T, \mathcal{L}_F) = (\mathcal{L}_t / count, \mathcal{L}_f / count)$
**Return:** $\mathcal{L}_T$ and $\mathcal{L}_F$

---

effectively reduces the misalignment between prediction outputs with the forecasting regression performance being unaffected, inducing overall predictions closely aligned with the ground truths.

For each prediction output point $i \in [1, h']$ in Alg. 1, $D_T(\boldsymbol{P}_{T, i \in idx_T}^n, sg(\boldsymbol{P}_{T, i \in idx_T}^m))$ on line (2) encourages the prediction $\boldsymbol{P}_T^n$ to move closer to the prediction $\boldsymbol{P}_T^m$. On the other hand, $D_T(sg(\boldsymbol{P}_{T, i \notin idx_T}^n), \boldsymbol{P}_{T, i \notin idx_T}^m)$ on the same line (2) promotes the alignment of $\boldsymbol{P}_T^m$ towards $\boldsymbol{P}_T^n$. The direction of alignment is determined by the index variable, $idx_T$. These two distance values are combined into the time domain alignment loss $\mathcal{L}_T$, which is expressed as:

$$\mathcal{L}_T = D_T(\boldsymbol{P}_{T, i \in idx_T}^n, sg(\boldsymbol{P}_{T, i \in idx_T}^m)) + D_T(sg(\boldsymbol{P}_{T, i \notin idx_T}^n), \boldsymbol{P}_{T, i \notin idx_T}^m) \tag{3}$$

Consequently, Eq. (3) aligns the direction of the time domain alignment loss $\mathcal{L}_T$ with the forecasting regression loss while improving output alignment. The regression pulling can also be applied in the same way to the frequency domain; Fig. 3 shows a visual illustration of output alignment in the time (left) and frequency domain (right) using regression pulling (red cross marks).

### 4.3 Frequency domain alignment

In addition to the time domain alignment, we propose frequency domain alignment as a complementary approach that supports and enhances time-domain alignment by aligning overlapping predictions in the frequency representation. It promotes the alignment of both the phase and the amplitude components, improving overall consistency in the time domain.

As shown in Alg. 1, the time domain overlapping predictions, $\boldsymbol{P}_T^n$ and $\boldsymbol{P}_T^m$, along with the ground truths $\boldsymbol{GT}_T$, are first transformed into frequency domain representations, resulting in $\boldsymbol{P}_F^n$, $\boldsymbol{P}_F^m$, and $\boldsymbol{GT}_F$ on line (3), where the subscript $F$ denotes the frequency domain. Subsequently, the index variable $idx_F$ is determined by identifying output points further from $\boldsymbol{GT}_F$ between $\boldsymbol{P}_F^n$ and $\boldsymbol{P}_F^m$ on line (4). The distance function $D_F$ is then applied with regression pulling to facilitate alignment in the frequency domain on lines (5) in the same manner to the time domain. From this, the frequency domain alignment loss $\mathcal{L}_F$ is obtained as:

$$\mathcal{L}_F = D_F(\boldsymbol{P}_{F, i \in idx_F}^n, sg(\boldsymbol{P}_{F, i \in idx_F}^m)) + D_F(sg(\boldsymbol{P}_{F, i \notin idx_F}^n), \boldsymbol{P}_{F, i \notin idx_F}^m) \tag{4}$$

We use MSE (mean squared error) as the main distance function for the frequency domain alignment, i.e., $D_F = \|\boldsymbol{P}_F^n - \boldsymbol{P}_F^m\|_2^2 / h'$. The following Theorem 4.1 demonstrates that applying MSE in the frequency domain facilitates phase alignment between prediction outputs.

**Theorem 4.1.** *Given two frequency domain prediction vectors, $\boldsymbol{p}_F^n$ and $\boldsymbol{p}_F^m$ in $\mathbb{C}^{h'}$, where $\mathbb{C}$ means the set of complex numbers, minimizing their MSE, $\|\boldsymbol{p}_F^n - \boldsymbol{p}_F^m\|_2^2$, results in a reduction of the difference*

*in the phase components, $\angle p_F^n$ and $\angle p_F^m$, as shown below (the proof is provided in Sec. B.1):*

$$\frac{1}{h'}\|p_F^n - p_F^m\|_2^2 \to 0 \implies |\angle p_F^n - \angle p_F^m| \to 0 \quad (5)$$

Fig. 3 (Right) shows a polar-coordinate depiction of frequency alignment conducted through Eq. (5) in the frequency domain. Theorem 4.2 shows that minimizing MSE, i.e., $D_F = \|P_F^n - P_F^m\|_2^2$, also reduces the differences in amplitude once the phase components $\angle P_F^n$ and $\angle P_F^m$ have been aligned.

**Theorem 4.2.** *Once the phases of two frequency domain prediction vectors, $p_F^n$ and $p_F^m$ in $\mathbb{C}^{h'}$, are aligned, i.e., $\angle p_F^n \simeq \angle p_F^m$, the minimization of MSE between the two prediction outputs, $\|p_F^n - p_F^m\|_2^2$, leads to a reduction in the amplitude difference, as shown below (the proof is provided in Sec. B.2):*

$$\frac{1}{h'}\|p_F^n - p_F^m\|_2^2 \to 0 \implies |p_F^n| - |p_F^m| \to 0 \quad (6)$$

### 4.4 AliO loss

By combining the time $\mathcal{L}_T$ in Eq. (3) and frequency domain alignment loss $\mathcal{L}_F$ in Eq. (4), we derive the AliO loss, $\mathcal{L}_{AliO}$, which is incorporated with the forecasting regression loss, $\mathcal{L}_{reg}$ as shown below, where $\lambda_T$ and $\lambda_F$ control the extent of alignment in the time and frequency domain, respectively.

$$\mathcal{L}_{AliO} = \lambda_T \mathcal{L}_T + \lambda_F \mathcal{L}_F \qquad \therefore \mathcal{L}_{total} = \mathcal{L}_{reg} + \mathcal{L}_{AliO} \quad (7)$$

## 5 Experiment

We evaluate AliO on representative LTSF models and datasets. In model training, MSE is employed as the regression loss. The model performance is assessed using both MSE and $TAM_N$ for output alignment evaluation, with $N = 2$. To reproduce experimental results of baselines and ensure a fair evaluation, we follow the same hyper-parameters presented in each model papers [37, 27, 40, 43, 30, 33, 36]. For AliO, we set $N = 2$ and $l = 1$, covering a wide prediction range. The search space for $\lambda_T$ and $\lambda_F$ are $\{1.0, 2.0, 5.0\}$ and $\{0.0, 0.5, 1.0, 2.0\}$, respectively, and we report the best results in the main results. All results are reported as the average performance across all prediction lengths. Detailed descriptions of the experiment and results are provided in Secs. G to I.

**Models.** We apply AliO to various LTSF architectures, experimenting it with a diverse range of LTSF approaches, categorized into four groups: (1) Transformer-based models, i.e., Autoformer [37], PatchTST [33], and iTransformer [30], (2) Linear-based models, i.e., DLinear [40] and CycleNet [27], (3) CNN-based models, i.e., TimesNet [36], and (4) LLM-based models, i.e., GPT4TS [43].

**Datasets.** On the main text of the paper, we report results on representative LTSF datasets, i.e., Electricity (ECL) [13], ETT {h1, h2, m1, and m2} [42], Traffic, [37], Weather [37], and National-Illness (ILI) dataset [37]. The experiments results on full datasets can be found in Sec. I.

**Context Length** Following the official settings for each model, we set the context length to 336 for PatchTST and DLinear, and 96 for the other models. For the ILI dataset specifically, the context lengths were set to 104 for PatchTST and DLinear, and 36 for TimesNet.

**Prediction Length** Furthermore, as per the official settings, the prediction lengths were set to $\{96, 192, 336, 720\}$ for most datasets. The exceptions were the PEMS-related datasets with $\{12, 24, 48, 96\}$, the ILI dataset with $\{24, 36, 48, 60\}$, and the Autoformer model on the ETT datasets, for which we used $\{24, 48, 168, 336, 720\}$.

### 5.1 Output alignment and forecasting performance

Tab. 1 summarizes the performance of LTSF models on seven datasets (excluding ILI) for multivariate LTSF tasks, comparing baselines (without AliO) and AliO-integrated models in terms of MSE and TAM. As shown in the table, integrating AliO with the regression loss substantially improves alignment performance (TAM), achieving gains of up to 70.5% for CycleNet, 45.8% for GPT4TS, 45.8% for iTransformer, 36.8% for PatchTST, 45.6% for TimesNet, 64.0% for DLinear, and 39.0% for Autoformer. Simultaneously, forecasting performance (MSE) also improves by up to 11.5%. As summarized in Tab. 2, for the ILI dataset exhibiting insufficient initial output alignment, AliO achieves up to a 17.4% higher MSE improvement compared to other benchmarks.

Table 1: The forecasting and alignment performance, measured by MSE and TAM$_2$, respectively, are compared between the baseline (MSE only) and AliO, with best results indicated in **bold**. The results are averaged over three random seeds and prediction lengths. The more metrics (MAE, MAPE, and RMSE) with standard deviations are provided in Sec. I.

| Models | | CycleNet | | GPT4TS | | iTransformer | | PatchTST | | TimesNet | | DLinear | | Autoformer |
|---|---|---|---|---|---|---|---|---|---|---|---|---|---|---|
| Metric | | MSE | TAM$_2$ | MSE | TAM$_2$ | MSE | TAM$_2$ | MSE | TAM$_2$ | MSE | TAM$_2$ | MSE | TAM$_2$ | MSE | TAM$_2$ |
| ECL | baseline | 0.171 | 0.016 | 0.167 | 0.036 | 0.176 | 0.050 | 0.162 | 0.037 | 0.196 | 0.040 | 0.166 | 0.013 | 0.229 | 0.041 |
| | AliO | **0.170** | **0.014** | **0.167** | **0.025** | **0.172** | **0.031** | **0.161** | **0.024** | **0.191** | **0.028** | **0.166** | **0.010** | **0.221** | **0.036** |
| ETTh1 | baseline | 0.432 | 0.044 | 0.424 | 0.056 | 0.455 | 0.088 | 0.418 | 0.076 | 0.476 | 0.05 | 0.422 | 0.025 | 0.477 | 0.05 |
| | AliO | **0.429** | **0.013** | **0.417** | **0.039** | **0.438** | **0.052** | **0.415** | **0.048** | **0.468** | **0.032** | **0.419** | **0.014** | **0.444** | **0.046** |
| ETTh2 | baseline | 0.385 | 0.047 | 0.363 | 0.048 | 0.382 | 0.074 | 0.341 | 0.061 | 0.416 | 0.079 | 0.449 | 0.054 | 0.401 | 0.041 |
| | AliO | **0.381** | **0.025** | **0.354** | **0.033** | **0.377** | **0.052** | **0.341** | **0.050** | **0.402** | **0.043** | **0.446** | **0.033** | **0.377** | **0.025** |
| ETTm1 | baseline | 0.386 | 0.032 | 0.351 | 0.044 | 0.407 | 0.060 | 0.352 | 0.050 | 0.412 | 0.045 | 0.358 | 0.024 | 0.488 | 0.070 |
| | AliO | **0.383** | **0.014** | **0.346** | **0.031** | **0.396** | **0.033** | **0.349** | **0.032** | **0.398** | **0.026** | **0.354** | **0.010** | **0.448** | **0.053** |
| ETTm2 | baseline | 0.272 | 0.027 | 0.270 | 0.042 | 0.292 | 0.048 | 0.257 | 0.059 | 0.296 | 0.036 | 0.289 | 0.037 | 0.271 | 0.052 |
| | AliO | **0.271** | **0.019** | **0.265** | **0.028** | **0.289** | **0.026** | **0.255** | **0.039** | **0.295** | **0.022** | **0.268** | **0.020** | **0.254** | **0.032** |
| Traffic | baseline | 0.487 | 0.021 | 0.411 | 0.059 | **0.422** | 0.061 | 0.389 | 0.041 | 0.626 | 0.040 | 0.436 | 0.025 | 0.634 | 0.048 |
| | AliO | **0.481** | **0.008** | **0.405** | **0.032** | 0.423 | **0.044** | **0388** | **0.027** | **0.554** | **0.030** | **0.434** | **0.009** | **0.624** | **0.045** |
| Weather | baseline | **0.255** | 0.010 | 0.227 | 0.024 | 0.260 | 0.024 | 0.229 | 0.018 | 0.262 | 0.018 | 0.245 | 0.007 | 0.342 | 0.056 |
| | AliO | 0.255 | **0.008** | **0.225** | **0.011** | **0.259** | **0.017** | **0.228** | **0.012** | **0.262** | **0.014** | **0.244** | **0.005** | **0.311** | **0.042** |

## 5.2 How AliO maintains or improves regression performance

Tab. 2 shows AliO's effectiveness in improving both forecasting accuracy and alignment performance on the ILI dataset [37], which exhibits poor output alignment in the absence of AliO. The proposed regression pulling enhances model training by analyzing prediction distances at each timestamp, adaptively strengthening the effect of regression loss to align predictions with each other and ground truth. On ILI, which exhibits poor initial alignment, AliO achieves 17.4% MSE and 58.1% TAM improvement. Conversely, datasets with better initial alignment, presented in Tab. 1, achieve comparatively smaller MSE improvements. Figs. 4 and 13 shows this trend; higher initial TAM (poor alignment) correlates with greater MSE gains, demonstrating AliO's efficacy in suboptimal alignment conditions.

Figs. 2 and 5 visualize DLinear predictions on the ILI dataset. The left figures (without AliO) exhibit prediction shifts [18] in both time (Fig. 2) and frequency (Fig. 5) domains, reflected in high TAM values. In contrast, the right figures (with AliO) show improved alignment in both domains, which reduces TAM value and enhances regression performance (MSE).

Table 2: The forecasting regression and alignment performance on ILI [37], measured in MSE, and TAM$_2$, are compared between the baseline (without AliO) and AliO. The best results are highlighted in **bold**. For robust evaluation, we conduct experiments with three random seeds. Results are rounded to three decimal places for simplicity.

| Models | GPT4TS | | PatchTST | | TimesNet | | DLinear | |
|---|---|---|---|---|---|---|---|---|
| Metric | MSE | TAM$_2$ | MSE | TAM$_2$ | MSE | TAM$_2$ | MSE | TAM$_2$ |
| base | 1.898 | 0.117 | 1.813 | 0.110 | 2.176 | 0.322 | 2.247 | 0.152 |
| AliO | **1.755** | **0.098** | **1.497** | **0.046** | **1.963** | **0.258** | **2.187** | **0.119** |

## 5.3 Hyper-parameter Analysis

**Time and Frequency Domain Coefficients.** AliO utilizes the two coefficients, $\lambda_T$ and $\lambda_F$ in Eq. (7), to balance the regression performance and output alignment by adjusting the alignment strengths over the time and frequency domains, respectively. To evaluate AliO with respect to these two coefficients, we provide Fig. 6, which presents heatmaps of the normalized MSE and TAM for various coefficient values. The x-axis represents $\lambda_F$ in $\{0.0, 0.5, 1.0, 2.0\}$, and the y-axis corresponds to $\lambda_T$ in $\{1.0, 2.0, 5.0\}$. The left heatmap shows normalized MSE, where the maximum MSE in each dataset is scaled to 1.0 and the minimum to 0, followed by averaging across all models. The right heatmap presents normalized TAM in the same manner. Additional results are in Sec. F.

As AliO is designed to enhance alignment, a clear trend is observed in TAM where increasing the time domain alignment coefficient $\lambda_T$ from 1.0 to 5.0 leads to a consistent improvement. Interestingly, the normalized MSE also shows a general improvement as $\lambda_T$ increases, suggesting that the regression pulling effectively aligns the directions of the AliO loss with those of the regression loss. The influence of the frequency domain alignment coefficient $\lambda_F$ is less pronounced compared to the time

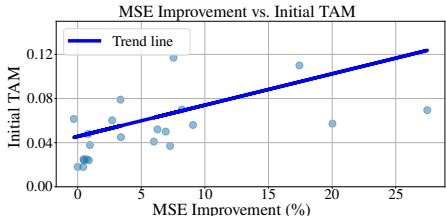

Figure 4: The relationship between initial TAM and MSE improvement (%). The correlation coefficient is 0.33, indicating a positive trend; when the initial alignment is poor (i.e., higher TAM values), the MSE improvement tends to be greater. For more detailed information, please check Sec. E.

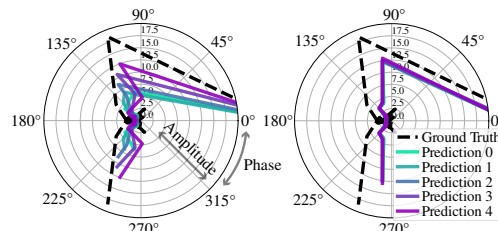

Figure 5: **(Left)** The frequency domain of five predictions in polar coordinates from DLinear [40] trained with regression loss, forecasting the ILI dataset. **(Right)** Five phases from the same setup, with AliO applied. AliO effectively aligns both the phase and amplitude of the model's predictions in the frequency domain.

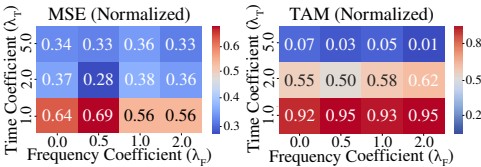

Figure 6: Comparison of the normalized MSE (**left**) and TAM (**right**) over different settings of $\lambda_T$ and $\lambda_F$. The x-axis represents $\lambda_F$ ($\{0.0, 0.5, 1.0, 2.0\}$), and the y-axis represents $\lambda_T$ ($\{1.0, 2.0, 5.0\}$). Increasing $\lambda_T$ consistently improves TAM performance, while $\lambda_F$ shows its impact depending on the value of $\lambda_T$. Blue represents better performance.

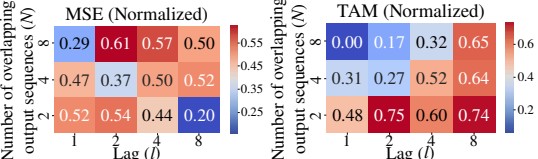

Figure 7: Comparison of the normalized MSE (**left**) and TAM (**right**) over different settings of the overlapping predictions $N$ and the lag $l$. The x-axis represents the lag $l$, and the y-axis represents the number of overlapping predictions $N$. Increasing $N$ improves TAM, while increasing $l$ degrades the performance. No distinct tendencies are observed regarding MSE. Blue represents better performance.

coefficient $\lambda_T$. However, in cases where $\lambda_T$ alone cannot improve performance, $\lambda_F$ shows its impact on both the MSE and TAM metrics, as exemplified by values of $0.28$ and $0.50$, respectively. This demonstrates the significance of the frequency domain alignment to overcome limitations in the time domain alignment, indicating that both phase and amplitude contribute to performance enhancements.

**Number of Overlapping Output Sequences and Lag Size.** We investigate how the model's MSE and TAM vary with different numbers of overlapping output sequences $N$ and the lag $l$. Fig. 7 shows the normalized MSE and TAM with a range between 0 and 1.0. TAM is improved as the lag decreases, which can be attributed to an increased length of overlapping timestamps at smaller lag value. This allows AliO to align longer overlapping sequence effectively. Additionally, the model performance improves as $N$ increases. As shown in Alg. 1, a higher $N$ enables AliO to align predictions from distant timestamps, significantly enhancing TAM. In contrast, no clear trend is observed for MSE, since the window size increases as both $N$ and $l$ grow. Prior studies [12, 2, 17, 32] suggest that large window sizes can cause overfitting, whereas an optimal window size leads to better performance. As shown in Fig. 7, when both $N$ and $l$ are large, MSE tends to increase, indicating degraded performance. Conversely, lower MSE values ($0.29, 0.37, 0.20$) are observed along the diagonal elements, representing proper window sizes, such as $(l, N) = (1, 8), (2, 4), (8, 2)$, which indicates that balanced combinations of $l$ and $N$ yield better performance.

## 6 Related works

To the best of our knowledge, no prior work has adequately identified the output alignment problem and provided a solution for it, as AliO proposed in this paper.

**Data-Model Robustness.** While existing research [30] primarily focuses on model initialization robustness, i.e., the consistency of model performance across randomly initialized weights [30], the output alignment emphasizes data-model robustness. This aspect underscores the model's ability

to produce consistent prediction outputs for lagged input sequences. The data-model robustness is equally critical for practical applications as discussed in this paper and can be quantified using the proposed TAM, which can be enhanced through the proposed time and frequency domain alignment. We anticipate this aspect of LTSF study will inspire further active research in the field.

**Contrastive Learning.** Contrastive learning [41, 8, 1, 9, 38, 22] employ pretraining strategies that encourage augmented positive pairs to be closely aligned in the representation space. However, they differ from AliO in several key aspects. First, they operate in the representation space rather than in the output space where AliO functions. Second, they minimize the distance between whole vectors rather than focusing on overlapping timestamps. Third, they do not incorporate regression-aware algorithms like regression pulling. These differences highlight the distinctive approach of AliO in output alignment, operating independently of existing methodes and can thus be integrated with them.

## 7 Limitations and discussions

**On Volatile Datasets.** TAM is a metric designed to assess the output alignment of LTSF models. However, its application should be approached with caution on highly volatile datasets that fall outside the typical scope of LTSF tasks, as model predictions may experience considerable volatility. Nevertheless, as demonstrated in Sec. I, the results on the Exchange Rate dataset [37], which is influenced by abrupt external shocks, suggest that TAM retains potential applicability even in volatile settings. Designing more robust metrics for inherently volatile data is our next research objective.

**Distance Functions.** AliO improves regression performance under high initial TAM conditions (a low degree of output alignment). This is likely attributable to its use of MSE as the distance function, which is consistent with the regression loss employed during model training. To further enhance performance, alternative distance functions such as [25, 29, 15, 7], which capture different characteristics of time series data (e.g., shape), may serve as promising directions for future exploration.

## 8 Conclusion

In this paper, we investigate the output alignment problem, which can commonly arise in LTSF tasks and propose Time Alignment Metric (TAM) as a quantitative measure for this problem. To solve the output alignment problem, we propose AliO (Output Alignment) applied in both time and frequency domains. AliO achieves up to 27.5% improvement in MSE across various LTSF models and datasets, and up to 58.2% improvement in TAM, effectively addressing the output alignment problem.

## Acknowledgment

This work was supported by the Institute of Information & communications Technology Planning & Evaluation(IITP) grant funded by the Korea government(MSIT)(No.RS-202400508465) and Institute of Information & communications Technology Planning & Evaluation(IITP) grant funded by the Korea government(MSIT) (No.RS-2020-II201336, Artificial Intelligence Graduate School Program(UNIST)).

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

## A  Exemplar illustrations on importance of output alignment

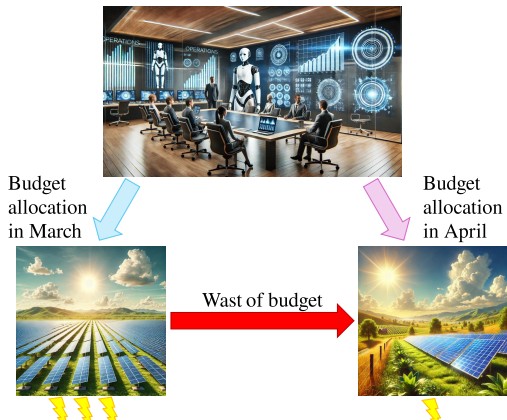

Budget allocation in March

Budget allocation in April

Wast of budget

Figure 8: A simple example illustrating the importance of *output alignment* in electricity power demand forecasting. The two predictions differ, causing confusion among people, and they realize that the budget allocated in March was a waste.

## B  Proof

### B.1  Proof of Theorem 4.1

*Proof.* Rewriting the equation on the left side of the theorem, we can express it form the perspective of individual elements:

$$\frac{|\boldsymbol{p}_F^1 - \boldsymbol{p}_F^2|_2^2}{h'} = \frac{1}{h'} \sum_i^{h'} |\boldsymbol{p}_{F,i}^1 - \boldsymbol{p}_{F,i}^2|^2 \tag{8}$$

The squared magnitude of the difference between two complex numbers $\boldsymbol{p}_{F,i}^1$ and $\boldsymbol{p}_{F,i}^2$ are given as follows:

$$|\boldsymbol{p}_{F,i}^1 - \boldsymbol{p}_{F,i}^2|^2 = (\boldsymbol{p}_{F,i}^1 - \boldsymbol{p}_{F,i}^2)(\overline{\boldsymbol{p}^1}_{F,i} - \overline{\boldsymbol{p}^2}_{F,i}) \tag{9}$$

Where $\overline{\boldsymbol{p}^1}_{F,i}$ and $\overline{\boldsymbol{p}^2}_{F,i}$ are the complex conjugate of $\boldsymbol{p}_{F,i}^1$ and $\boldsymbol{p}_{F,i}^2$. Now, expressing $\boldsymbol{p}_{F,i}^1$ and $\boldsymbol{p}_{F,i}^2$ including the complex conjugate in polar form:

$$\boldsymbol{p}_{F,i}^1 = |\boldsymbol{p}_{F,i}^1| e^{j\theta_{\boldsymbol{p}_{F,i}^1}}, \quad \boldsymbol{p}_{F,i}^2 = |\boldsymbol{p}_{F,i}^2| e^{j\theta_{\boldsymbol{p}_{F,i}^2}} \tag{10}$$

$$\overline{\boldsymbol{p}^1}_{F,i} = |\boldsymbol{p}_{F,i}^1| e^{-j\theta_{\boldsymbol{p}_{F,i}^1}}, \quad \overline{\boldsymbol{p}^2}_{F,i} = |\boldsymbol{p}_{F,i}^2| e^{-j\theta_{\boldsymbol{p}_{F,i}^2}} \tag{11}$$

where $j = \sqrt{-1}$ and $\theta$ is the angle (phase) for the corresponding complex number. We substitute into the expression and simplify the expression:

$$|\boldsymbol{p}_{F,i}^1 - \boldsymbol{p}_{F,i}^2|^2 = (|\boldsymbol{p}_{F,i}^1| e^{j\theta_{\boldsymbol{p}_{F,i}^1}} - |\boldsymbol{p}_{F,i}^2| e^{j\theta_{\boldsymbol{p}_{F,i}^2}})(|\boldsymbol{p}_{F,i}^1| e^{-j\theta_{\boldsymbol{p}_{F,i}^1}} - |\boldsymbol{p}_{F,i}^2| e^{-j\theta_{\boldsymbol{p}_{F,i}^2}}) \tag{12}$$

$$= |\boldsymbol{p}_{F,i}^1|^2 + |\boldsymbol{p}_{F,i}^2|^2 - |\boldsymbol{p}_{F,i}^1||\boldsymbol{p}_{F,i}^2|(e^{j(\theta_{\boldsymbol{p}_{F,i}^1} - \theta_{\boldsymbol{p}_{F,i}^2})} + e^{-j(\theta_{\boldsymbol{p}_{F,i}^1} - \theta_{\boldsymbol{p}_{F,i}^2})}) \tag{13}$$

Using Euler's formula $e^{j\theta} = cos(\theta) + jsin(\theta)$ [10], we can convert the exponential expressions to trigonometric expressions:

$$|\boldsymbol{p}_{F,i}^1 - \boldsymbol{p}_{F,i}^2|^2 = |\boldsymbol{p}_{F,i}^1|^2 + |\boldsymbol{p}_{F,i}^2|^2 - |\boldsymbol{p}_{F,i}^1||\boldsymbol{p}_{F,i}^2|\{cos(\theta_{\boldsymbol{p}_{F,i}^1} - \theta_{\boldsymbol{p}_{F,i}^2}) + jsin(\theta_{\boldsymbol{p}_{F,i}^1} - \theta_{\boldsymbol{p}_{F,i}^2}) \tag{14}$$

$$+ cos(-(\theta_{\boldsymbol{p}_{F,i}^1} - \theta_{\boldsymbol{p}_{F,i}^2})) + jsin(-(\theta_{\boldsymbol{p}_{F,i}^1} - \theta_{\boldsymbol{p}_{F,i}^2}))\} \tag{15}$$

$$= |\boldsymbol{p}_{F,i}^1|^2 + |\boldsymbol{p}_{F,i}^2|^2 - 2|\boldsymbol{p}_{F,i}^1||\boldsymbol{p}_{F,i}^2|cos(\theta_{\boldsymbol{p}_{F,i}^1} - \theta_{\boldsymbol{p}_{F,i}^2}) \tag{16}$$

Utilizing the angle (phase) notation $\angle$, we can express the equation as:

$$|\boldsymbol{p}_{F,i}^1 - \boldsymbol{p}_{F,i}^2|^2 = |\boldsymbol{p}_{F,i}^1|^2 + |\boldsymbol{p}_{F,i}^2|^2 - 2|\boldsymbol{p}_{F,i}^1||\boldsymbol{p}_{F,i}^2|cos(\angle\boldsymbol{p}_{F,i}^1 - \angle\boldsymbol{p}_{F,i}^2) \tag{17}$$

Reducing the final equation implies increasing the cosine term, which in turn signifies aligning the two phases, since one of predictions is constant by stop-gradient $sg(\cdot)$ operator in Alg. 1.

$$|\boldsymbol{p}_{F,i}^1 - \boldsymbol{p}_{F,i}^2|^2 \to 0 \implies cos(\angle \boldsymbol{p}_{F,i}^1 - \angle \boldsymbol{p}_{F,i}^2) \to 1 \tag{18}$$

$$cos(\angle \boldsymbol{p}_{F,i}^1 - \angle \boldsymbol{p}_{F,i}^2) \to 1 \implies |\angle \boldsymbol{p}_F^1 - \angle \boldsymbol{p}_F^2| \to 0 \tag{19}$$

$$\therefore |\boldsymbol{p}_{F,i}^1 - \boldsymbol{p}_{F,i}^2|^2 \to 0 \implies |\angle \boldsymbol{p}_F^1 - \angle \boldsymbol{p}_F^2| \to 0 \tag{20}$$

$\square$

### B.2 Proof of Theorem 4.2

*Proof.* Since the two phases, i.e., $\angle \boldsymbol{p}_{F,i}^1$ and $\angle \boldsymbol{p}_{F,i}^2$, are aligned, the Eq. (17) in Sec. B.1 is modified as follows:

$$|\boldsymbol{p}_{F,i}^1 - \boldsymbol{p}_{F,i}^2|^2 = |\boldsymbol{p}_{F,i}^1|^2 + |\boldsymbol{p}_{F,i}^2|^2 - 2|\boldsymbol{p}_{F,i}^1||\boldsymbol{p}_{F,i}^2| \tag{21}$$

By factoring, it can be expressed in the following perfect square form.

$$|\boldsymbol{p}_{F,i}^1 - \boldsymbol{p}_{F,i}^2|^2 = (|\boldsymbol{p}_{F,i}^1|^2 - |\boldsymbol{p}_{F,i}^2|^2)^2 \tag{22}$$

Since one of predictions is constant by stop-gradient $sg(\cdot)$ operator in Alg. 1, minimizing $|\boldsymbol{p}_{F,i}^1 - \boldsymbol{p}_{F,i}^2|^2$ leads to a reduction in the amplitude difference $\square$

## C   Visualization of alignment

In this section, we visualize the model's prediction outputs and alignment differences when trained solely with regression loss versus with both regression loss and AliO. Each figure is arranged in a 2×2 grid: the top row shows results using regression loss alone, while the bottom row presents results incorporating AliO; the left column illustrates the time domain, and the right column displays the frequency domain visualized in a polar coordinate system, where the azimuth angle represents the phase and the radial distance from the origin corresponds to the amplitude. Overlapping timestamps of prediction outputs for five lagged inputs are shown, with the ground truth indicated by black dashed lines. Figs. 9 to 12 show the results from DLinear [40], PatchTST [33], TimesNet [36], and iTransformer [30].

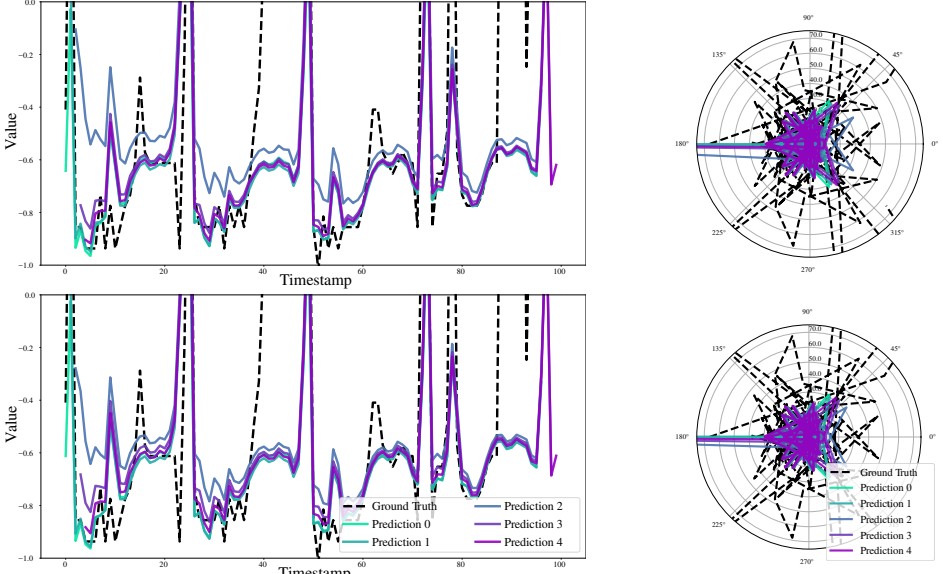

Figure 9: **Visualization**: **Upper** shows the prediction of DLinear [40] using only the regression loss on the ECL dataset [13], while **Bottom** shows the result under the same conditions when AliO is also applied. It is observed that AliO improves alignment performance for overlapping timestamps.

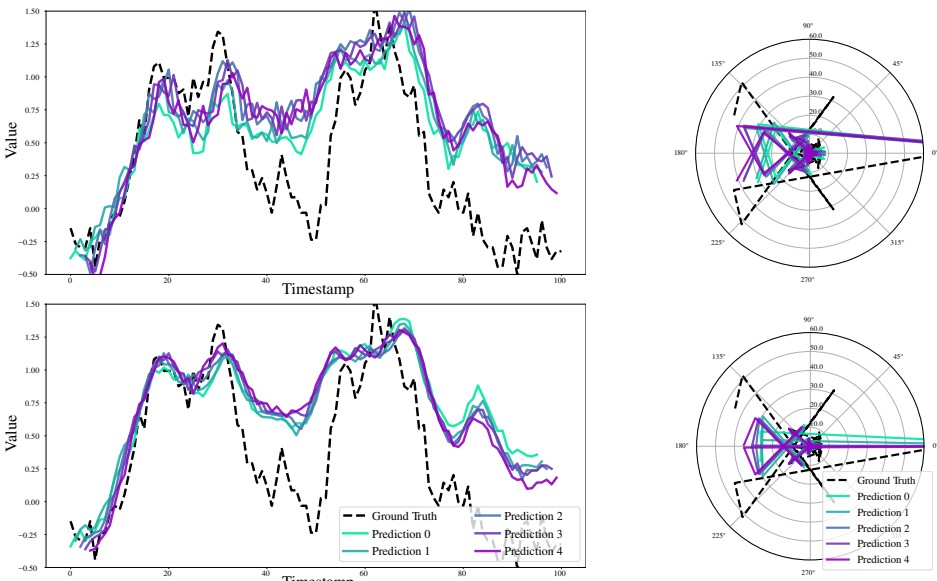

Figure 10: **Visualization**: **Upper** shows the prediction of PatchTST [33] using only the regression loss on the ETTm1 dataset [42], while **Bottom** shows the result under the same conditions when AliO is also applied. It is observed that AliO improves alignment performance for overlapping timestamps.

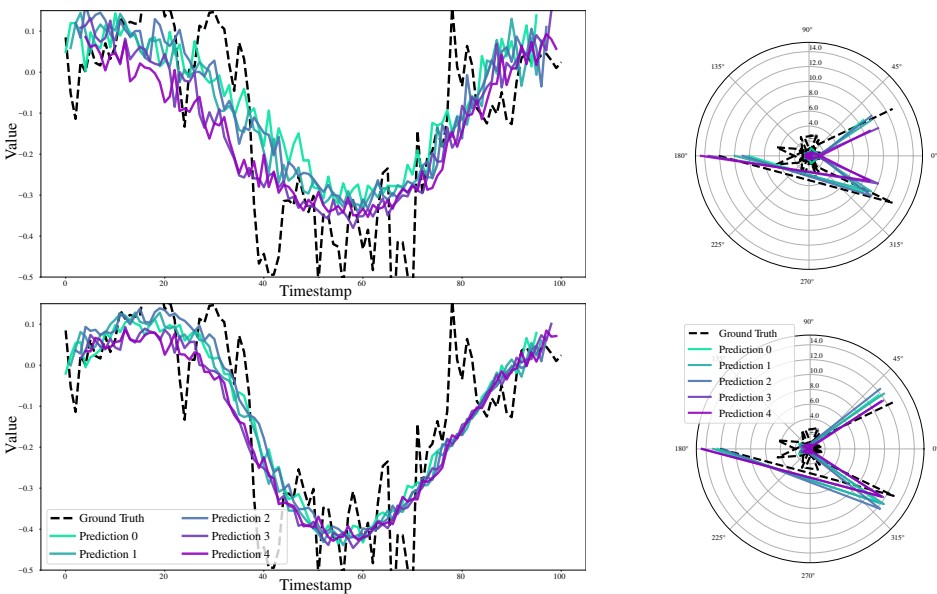

Figure 11: **Visualization**: **Upper** shows the prediction of TimesNet [36] using only the regression loss on the ETTm2 dataset [42], while **Bottom** shows the result under the same conditions when AliO is also applied. It is observed that AliO improves alignment performance for overlapping timestamps.

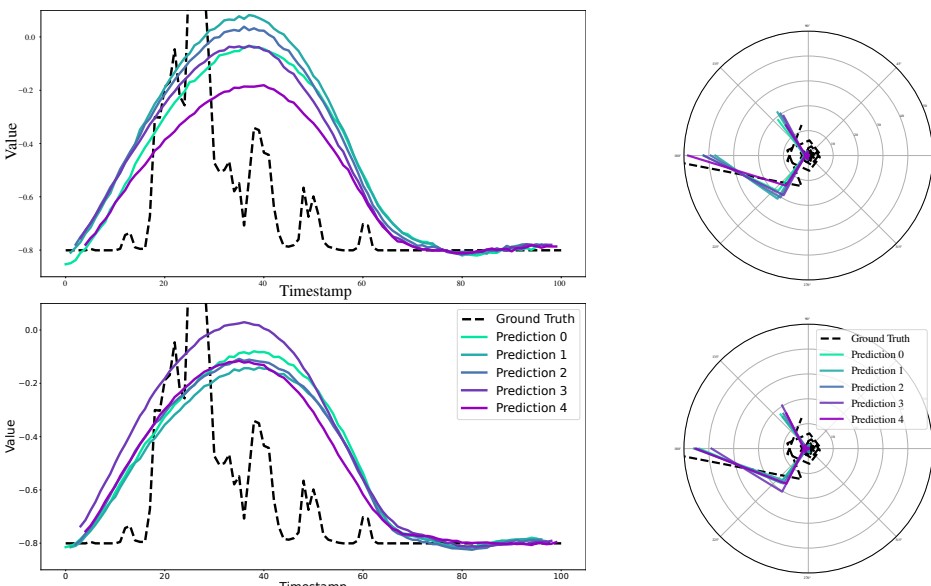

Figure 12: **Visualization**: **Upper** shows the prediction of iTransformer [30] using only the regression loss on the Solar dataset [24], while **Bottom** shows the result under the same conditions when AliO is also applied. It is observed that AliO improves alignment performance for overlapping timestamps.

# D  On sudden event

This section expands on the sudden events mentioned in Sec. 7.

**Definition D.1.** The sudden event refers to an anomalous data point in the input sequence that can be interpreted as out-of-distribution (OOD), causing perturbations in the output sequence. Such event occur when input-level perturbations lead to unpredictable output patterns.

These scenarios deviate from the core objective of Long-term Time Series Forecasting (LTSF), which focuses on predictable sequences. Instead, they align more closely with:

- Domain Adaptation [19]
- Anomaly Detection [39]
- Test-Time Adaptation [21]

The TAM metric is specifically designed for LTSF tasks with predictable, stable output sequences (i.e., non-perturbed scenarios). In contexts involving sudden events or OOD (out-of-distribution) data, the inherent assumptions of TAM-particularly its reliance on sequence stability-may not hold, necessitating cautious application. However, if a sudden event includes precursor signals (i.e., detectable input patterns) and the model can sufficiently predict the output sequence using this information, TAM remains valid even under such conditions.

# E  Initial TAM vs. MSE improvement

Fig. 4 may be difficult to interpret as there is no explicit distinction between models and datasets. In this section [Fig. 13], to facilitate easier interpretation, models are represented by different shapes and datasets are represented by different colors.

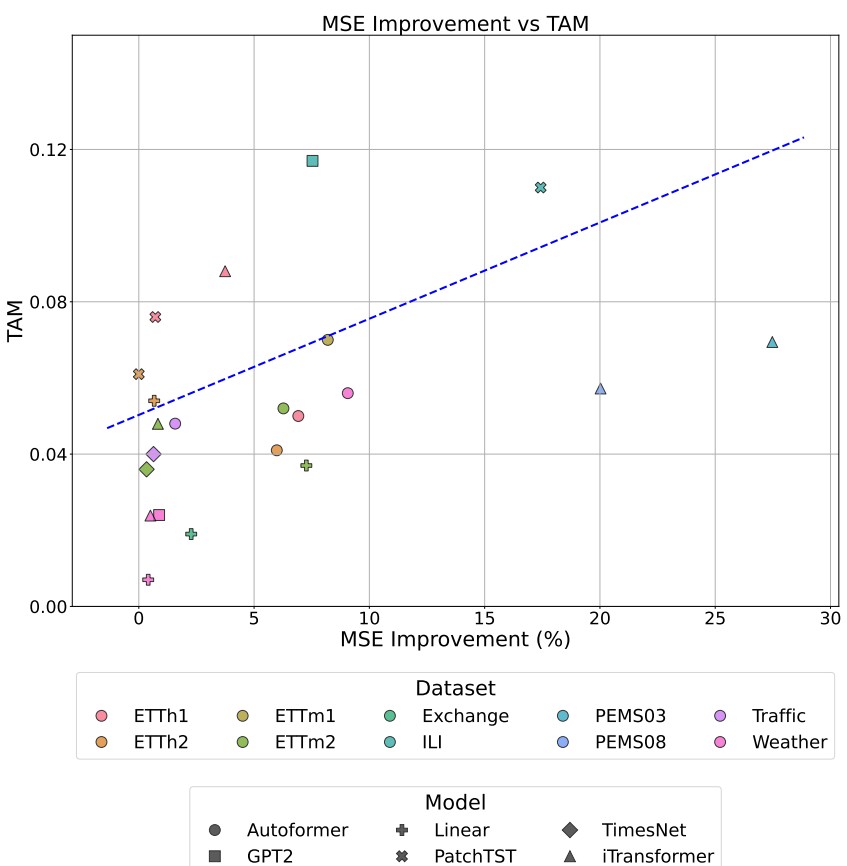

Figure 13: The relationship between initial TAM and MSE improvement (%).

# F  Zero coefficient of time domain alignment ($\lambda_T$)

By default, AliO operates in the time domain, so scenarios where the time-scaling coefficient is zero were excluded from the main context. However, we conducted experiments on CycleNet [27], iTransformer [30], TimesNet [36] under conditions with a zero time-scaling coefficient. The results, shown in Figs. 14 to 25, are normalized between 0 and 1. Since the condition ($\lambda_T = 0$, $\lambda_F = 0$) implies that AliO is not used, so it is empty. As demonstrated in Fig. 6 increasing $\lambda_T$ concsistently improves TAM performance. When it comes to MSE performance, there's a general trend for it to improve as the coefficient increases, similar to what's seen with TAM (I'm assuming this refers to a specific model or method you're using). However, in some environments, the coefficient can become excessively large, dominating the regression loss and causing performance to degrade. Despite this, our experiments confirmed that an appropriate, non-zero coefficient leads to improved performance. The tables show the results where the lag is 1 and the number of sequences is 2 (default).

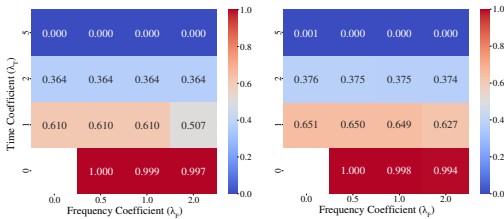

Figure 14: Comparison of the normalized MSE (**left**) and TAM (**right**) of CycleNet trained on ETTm1 dataset. The x-axis represents $\lambda_F$ ($\{0.0, 0.5, 1.0, 2.0\}$), and the y-axis represents $\lambda_T$ ($\{0.0, 1.0, 2.0, 5.0\}$).

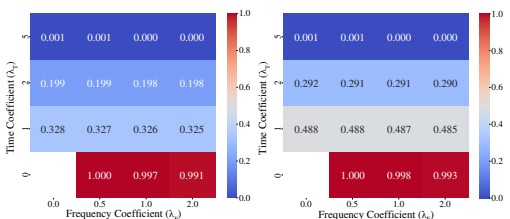

Figure 15: Comparison of the normalized MSE (**left**) and TAM (**right**) of CycleNet trained on ETTm2 dataset. The x-axis represents $\lambda_F$ ($\{0.0, 0.5, 1.0, 2.0\}$), and the y-axis represents $\lambda_T$ ($\{0.0, 1.0, 2.0, 5.0\}$).

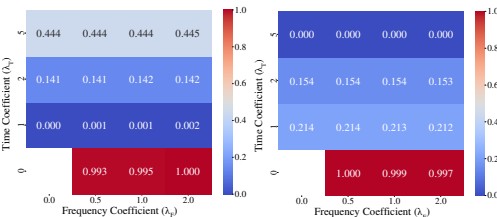

Figure 16: Comparison of the normalized MSE (**left**) and TAM (**right**) of CycleNet trained on Traffic dataset. The x-axis represents $\lambda_F$ ($\{0.0, 0.5, 1.0, 2.0\}$), and the y-axis represents $\lambda_T$ ($\{0.0, 1.0, 2.0, 5.0\}$).

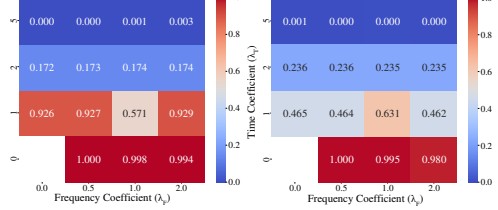

Figure 17: Comparison of the normalized MSE (**left**) and TAM (**right**) of iTransformer trained on ETTh1 dataset. The x-axis represents $\lambda_F$ ($\{0.0, 0.5, 1.0, 2.0\}$), and the y-axis represents $\lambda_T$ ($\{0.0, 1.0, 2.0, 5.0\}$).

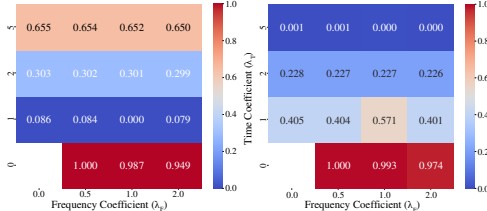

Figure 18: Comparison of the normalized MSE (**left**) and TAM (**right**) of iTransformer trained on ETTh2 dataset. The x-axis represents $\lambda_F$ ($\{0.0, 0.5, 1.0, 2.0\}$), and the y-axis represents $\lambda_T$ ($\{0.0, 1.0, 2.0, 5.0\}$).

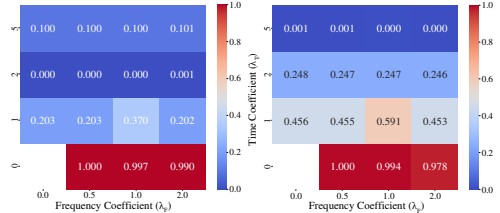

Figure 19: Comparison of the normalized MSE (**left**) and TAM (**right**) of iTransformer trained on ETTm1 dataset. The x-axis represents $\lambda_F$ ($\{0.0, 0.5, 1.0, 2.0\}$), and the y-axis represents $\lambda_T$ ($\{0.0, 1.0, 2.0, 5.0\}$).

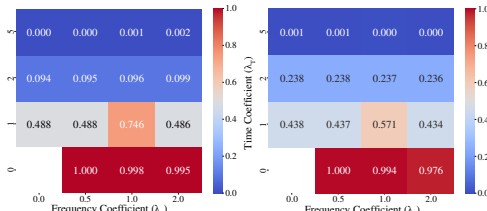

Figure 20: Comparison of the normalized MSE (**left**) and TAM (**right**) of iTransformer trained on ETTm2 dataset. The x-axis represents $\lambda_F$ ($\{0.0, 0.5, 1.0, 2.0\}$), and the y-axis represents $\lambda_T$ ($\{0.0, 1.0, 2.0, 5.0\}$).

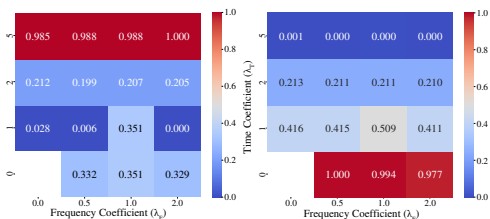

Figure 21: Comparison of the normalized MSE (**left**) and TAM (**right**) of iTransformer trained on Traffic dataset. The x-axis represents $\lambda_F$ ($\{0.0, 0.5, 1.0, 2.0\}$), and the y-axis represents $\lambda_T$ ($\{0.0, 1.0, 2.0, 5.0\}$).

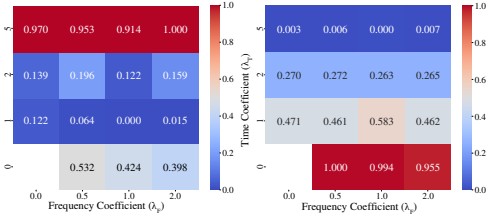

Figure 22: Comparison of the normalized MSE (**left**) and TAM (**right**) of iTransformer trained on Weather dataset. The x-axis represents $\lambda_F$ ($\{0.0, 0.5, 1.0, 2.0\}$), and the y-axis represents $\lambda_T$ ($\{0.0, 1.0, 2.0, 5.0\}$).

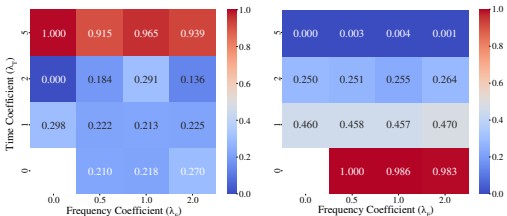

Figure 23: Comparison of the normalized MSE (**left**) and TAM (**right**) of TimesNet trained on ETTh1 dataset. The x-axis represents $\lambda_F$ ($\{0.0, 0.5, 1.0, 2.0\}$), and the y-axis represents $\lambda_T$ ($\{0.0, 1.0, 2.0, 5.0\}$).

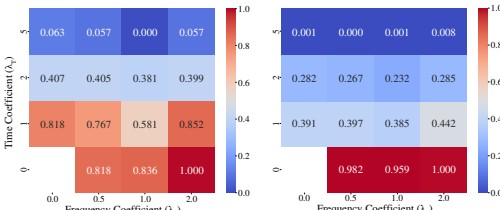

Figure 24: Comparison of the normalized MSE (**left**) and TAM (**right**) of TimesNet trained on ETTm1 dataset. The x-axis represents $\lambda_F$ ($\{0.0, 0.5, 1.0, 2.0\}$), and the y-axis represents $\lambda_T$ ($\{0.0, 1.0, 2.0, 5.0\}$).

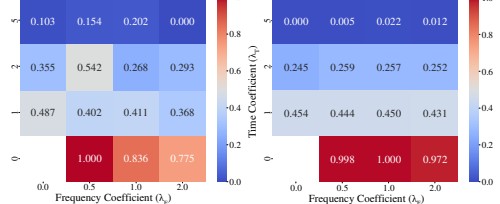

Figure 25: Comparison of the normalized MSE (**left**) and TAM (**right**) of TimesNet trained on ETTm2 dataset. The x-axis represents $\lambda_F$ ($\{0.0, 0.5, 1.0, 2.0\}$), and the y-axis represents $\lambda_T$ ($\{0.0, 1.0, 2.0, 5.0\}$).

## G   Dataset explanation

We provide a brief description of each dataset used in our experiments, as referenced in Secs. 5 and I.

- **Electricity transformer temperature (ETT)** [42]: This dataset consists of two years of data collected from two counties in China. It is divided into four subsets: ETTh1 and ETTh2 (sampled hourly), and ETTm1 and ETTm2 (sampled every 15 minutes). Each record contains six power load features and one target variable representing the oil temperature.

- **Exchange** [37]: Contains daily exchange rates from eight countries, spanning from 1990 to 2016.

- **Electricity (ECL)** [13]: Comprises electric power consumption data sampled every minute over four years for a single household.

- **ILI** (https: / / github.com / thuml / Autoformer): Weekly records from 2002 to 2021, provided by the US Centers for Disease Control and Prevention, representing the number of influenza-like illness patients.

- **PEMS**: Contains traffic information measured at 5-minute intervals on California highways. The subsets PEMS03, PEMS04, PEMS07, and PEMS08 correspond to different regions and time spans, with varying numbers of sensors [28].

- **Solar-Energy** [24]: Includes solar energy production data from 137 plants in 2006, measured at 10-minute intervals.

- **Traffic** [37]: Consists of hourly traffic congestion data collected by 862 sensors on San Francisco freeways from January 2015 to December 2016.

To ensure fair performance evaluation, we adopted the sequence length, prediction length, and label length settings used in recent LTSF models, such as Autoformer [37], DLinear [40], PatchTST [33], TimesNet [36], iTransformer [30], GPT4TS [43], CycleNet [27]. Following previous works [37, 30, 28]. Tab. 3 shows the descriptions of datasets including the number of variate in each dataset, prediction length, dataset size, sampling frequency, and domain. Since the official Autoformer implementation provides the results in different prediction length (24, 48, 168, 336, 720), we follow their implementation and show the results on Sec. I.

Table 3: Descriptions of datasets. # of vars means the number of variate in each dataset.

| Dataset | # of vars | Prediction Length | Dataset size (Train / Validation / Test) | Frequency | Domain |
|---------|-----------|-------------------|------------------------------------------|-----------|--------|
| ETTh1 | 7 | 24, 48, 168, 336, 720 (Autoformer) 96, 192, 336, 720 (other models) | 8545 / 2881 / 2881 | Hourly | Temperature |
| ETTh2 | 7 | 24, 48, 168, 336, 720 (Autoformer) 96, 192, 336, 720 (other models) | 8545 / 2881 / 2881 | Hourly | Temperature |
| ETTm1 | 7 | 24, 48, 168, 336, 720 (Autoformer) 96, 192, 336, 720 (other models) | 34465 / 11521 / 11521 | 15 min | Temperature |
| ETTm2 | 7 | 24, 48, 168, 336, 720 (Autoformer) 96, 192, 336, 720 (other models) | 34465 / 11521 / 11521 | 15 min | Temperature |
| Weather | 21 | 96, 192, 336, 720 | 36792 / 5271 / 10540 | 10 min | Weather |
| Electricity (ECL) | 321 | 96, 192, 336, 720 | 18317 / 2633 / 5261 | Hourly | Electricity |
| Traffic | 862 | 96, 192, 336, 720 | 12185 / 1757 / 3509 | Hourly | Transportation |
| ILI | 7 | 24, 36, 48, 60 | 617 / 74 / 170 | Weekly | Health |
| Exchange Rate (Exchange) | 8 | 96, 192, 336, 720 | 5120 / 665 / 1422 | Daily | Economy |
| Solar | 137 | 96, 192, 336, 720 | 36601 / 5161 / 10417 | 10 min | Energy |
| PEMS03 | 358 | 12, 24, 48, 96 | 15617 / 5135 / 5135 | 5 min | Transportation |
| PEMS04 | 307 | 12, 24, 48, 96 | 10172 / 3375 / 3375 | 5 min | Transportation |
| PEMS07 | 883 | 12, 24, 48, 96 | 16911 / 5622 / 5622 | 5 min | Transportation |
| PEMS08 | 170 | 12, 24, 48, 96 | 10690 / 3548 / 3548 | 5 min | Transportation |

## H  Experiment configurations

To ensure fair model evaluation, we utilized the official GitHub codes provided by six benchmark models: Autoformer [37], DLinear [40], PatchTST [33], TimesNet [36], iTransformer [30], GPT4TS [43], CycleNet [27]. For all experiments, we adopted the same prediction length, label length, and sequence length as the official implementations, and maintained the original architecture of each model.

**Context lengths.**  According to the papers [37, 40, 33, 36, 30, 43, 27], the used context lenghts are followed by:

- CycleNet, Autoformer, iTransformer: 96

- TimesNet: 96 (36 for the ILI dataset)

- PatchTST, DLinear: 336 (104 for the ILI dataset)

**Prediction lengths.** We primarily used prediction lengths of $\{96, 192, 336, 720\}$. For the ILI dataset (https://github.com/thuml/Autoformer), we followed prior works and used $\{24, 36, 48, 60\}$. As an exception, for the ETT$\{$h1, h2, m1, m2$\}$ datasets, we followed the Autoformer paper and used $\{24, 48, 168, 336, 720\}$ (for Autoformer) and $\{96, 192, 336, 720\}$ (for other models).

**Optimization.** The optimizer and scheduler were used under the same conditions as specified in the official codes for each model.

**Our method.** For our method (AliO), the hyperparameters—the number of samples $N$ in Alg. 1 and lag $l$ in Alg. 1—were set to their default values of 2 and 1, respectively. The Mean Squared Error (MSE) function was used as the distance function for both the time and frequency domains. The coefficients for the time domain ($\lambda_T$) and frequency domain ($\lambda_F$) were selected from the ranges $\{1.0, 2.0, 5.0\}$ and $\{0.0, 0.5, 1.0, 2.0\}$, respectively (default values are $\lambda_T = 1$ and $\lambda_F = 0$ since the frequency domain is optional). Additionally, the AWL [26] technique was employed to automatically tune these coefficients and report the best performance.

**Evaluation.** We used Mean Squared Error (MSE) as our baseline loss function and reported forecasting performance using MSE, Mean Absolute Error (MAE), Mean Absolute Percentage Error (MAPE), Root Mean Squared Error (RMSE), and TAM. For all these metrics, lower values indicate better performance.

**Robustness.** To ensure robust results, we used three random seeds (2023, 2024, 2025) for initialization and report the standard deviation using $\pm$. All experiments were conducted on NVIDIA RTX 3090 and A6000 GPUs. We conducted all experiments using the same GPU when comparing the baseline and AliO under the same conditions.

**Training hyper-parameters.** We used the same learning rate, batch size, and epoch as the official implementation of each model for reproducibility and fair comparison. The implementation code for each model is as follows.

- Autoformer: `https://github.com/thuml/Autoformer`

- DLinear: `https://github.com/vivva/DLinear`

- PatchTST: `https://github.com/yuqinie98/PatchTST`

- TimesNet: `https://github.com/thuml/TimesNet`

- iTransformer: `https://github.com/thuml/iTransformer`

- GPT4TS: `https://github.com/DAMO-DI-ML/NeurIPS2023-One-Fits-All`

- CycleNet: `https://github.com/ACAT-SCUT/CycleNet`

We followed the implementation code listed above (GitHub) and used the same learning rate, batch size, and number of epochs as shown in Tabs. 4 to 11. The optimizer we used is Adam [23].

Table 4: Learning rate and batch size of Autoformer used in each datasets.

| Dataset | ECL | ETTh1 | ETTh2 | ETTm1 | ETTm2 | Exchange | Traffic | Weather |
|---|---|---|---|---|---|---|---|---|
| **Learning rate** | 0.0001 | 0.0001 | 0.0001 | 0.0001 | 0.0001 | 0.0001 | 0.0001 | 0.0001 |
| **batch size** | 32 | 32 | 32 | 32 | 32 | 32 | 32 | 32 |
| **Epoch** | 10 | 10 | 10 | 10 | 10 | 10 | 3 | 10 |

Table 5: Learning rate and batch size of DLinear used in each datasets.

| Dataset | ECL | ETTh1 | ETTh2 | ETTm1 | ETTm2 | Exchange | ILI | Traffic | Weather |
|---|---|---|---|---|---|---|---|---|---|
| **Learning rate** | 0.001 | 0.0001 | 0.05 | 0.0001 | 0.001 0.01 | 0.0005 0.005 | 0.05 | 0.05 | 0.0001 |
| **batch size** | 16 | 8 | 32 | 8 | 32 | 8 32 | 32 | 16 | 16 |
| **Epoch** | 10 | 10 | 10 | 10 | 10 | 10 | 10 | 10 | 10 |

Table 6: Learning rate and batch size of PatchTST used in each datasets.

| Dataset | ECL | ETTh1 | ETTh2 | ETTm1 | ETTm2 | ILI | Traffic | Weather |
|---|---|---|---|---|---|---|---|---|
| **Learning rate** | 0.0001 | 0.0001 | 0.0001 | 0.0001 | 0.0001 | 0.0025 | 0.0001 | 0.0001 |
| **batch size** | 32 | 128 | 128 | 128 | 128 | 16 | 6 | 128 |
| **Epoch** | 100 | 100 | 100 | 100 | 100 | 100 | 100 | 100 |

Table 7: Learning rate and batch size of TimesNet used in each datasets.

| Dataset | ECL | ETTh1 | ETTh2 | ETTm1 | ETTm2 | ILI | Exchange | Traffic | Weather |
|---|---|---|---|---|---|---|---|---|---|
| **Learning rate** | 0.0001 | 0.0001 | 0.0001 | 0.0001 | 0.0001 | 0.0001 | 0.0001 | 0.0001 | 0.0001 |
| **batch size** | 32 | 32 | 32 | 32 | 32 | 32 | 32 | 16 | 32 |
| **Epoch** | 10 | 10 | 10 | 10 | 10 | 10 | 10 | 10 | 10 |

Table 8: Learning rate and batch size of iTransformer used in each datasets (excluding PEMS and Solar).

| Dataset | ECL | ETTh1 | ETTh2 | ETTm1 | ETTm2 | ILI | Exchange | Traffic | Weather |
|---|---|---|---|---|---|---|---|---|---|
| **Learning rate** | 0.0005 | 0.0001 | 0.0001 | 0.0001 | 0.0001 | 0.0001 | 0.0001 | 0.001 | 0.0001 |
| **batch size** | 16 | 32 | 32 | 32 | 32 | 32 | 32 | 16 | 32 |
| **Epoch** | 10 | 10 | 10 | 10 | 10 | 10 | 10 | 10 | 10 |

Table 9: Learning rate and batch size of iTransformer used in each datasets (including PEMS and Solar.

| Dataset | PEMS03 | PEMS04 | PEMS07 | PEMS08 | Solar |
|---|---|---|---|---|---|
| **Learning rate** | 0.001 | 0.0005 | 0.001 | 0.0001 0.001 | 0.0005 |
| **batch size** | 32 | 32 | 32 16 | 32 16 | 32 |
| **Epoch** | 10 | 10 | 10 | 10 | 10 |

Table 10: Learning rate and batch size of GPT4TS used in each datasets.

| Dataset | ECL | ETTh1 | ETTh2 | ETTm1 | ETTm2 | ILI | Traffic | Weather |
|---|---|---|---|---|---|---|---|---|
| **Learning rate** | 0.0001 | 0.0001 | 0.0001 | 0.0001 | 0.0001 | 0.0001 | 0.001 | 0.0001 |
| **batch size** | 512 | 256 | 256 | 256 | 256 | 16 | 256 | 512 |
| **Epoch** | 10 | 10 | 10 | 10 | 10 | 10 | 10 | 10 |

Table 11: Learning rate and batch size of CycleNet used in each datasets.

| Dataset | ECL | ETTh1 | ETTh2 | ETTm1 | ETTm2 | Traffic | Weather | Solar |
|---|---|---|---|---|---|---|---|---|
| **Learning rate** | 0.0001 | 0.0001 | 0.0001 | 0.0001 | 0.0001 | 0.0001 | 0.0001 | 0.0001 |
| **batch size** | 128 | 128 | 128 | 128 | 128 | 128 | 128 | 128 |
| **Epoch** | 30 | 30 | 30 | 30 | 30 | 30 | 30 | 30 |

# I  Full experimental results

This section presents the comprehensive results of each Long-Term Series Forecasting (LTSF) models, Autoformer [37] (Tabs. 12 and 13), DLinear [40] (Tabs. 14 and 15), PatchTST [33] (Tabs. 16 and 17), TimesNet [36] (Tabs. 18 and 19), iTransformer [30] (Tabs. 20 to 22), GPT4TS [43] (Tabs. 23 and 24), and CycleNet [27] (Tabs. 25 and 26). The evaluation metrics include Mean Squared Error (MSE), Temporal Alignment Metric (TAM), Mean Absolute Error (MAE), Mean Absolute Percentage Error (MAPE), and Root Mean Squared Error (RMSE). The $\pm$ symbol denotes the standard deviation across multiple seeds (we used three random seeds for robust experiment). AVG represents the average value across all prediction lengths, with the best performance highlighted in **bold**.

Table 12: Full results of the proposed AliO method on the ECL, ETTh1, ETTh2, ETTm1, ETTm2, Exchange, Traffic, and Weather datasets and Autoformer [37]. The results are reported in terms of MSE, TAM, MAE, MAPE, and RMSE. The best results are highlighted in **bold**. We use official GitHub code (https://github.com/thuml/Autoformer) to train the model with the official configuration. The average improvement of AliO over the baseline is MSE: 6.98%, TAM: 24.24%, MAE: 3.38%, MAPE: 3.76%, and RMSE: 3.12%. The maximum improvement of AliO over the baseline is MSE: 27.90%, TAM: 81.18%, MAE: 15.17%, MAPE: 15.42%, and RMSE: 8.58%. Exchange, Traffic, and Weather are in Tab. 13.

| Models | | Autoformer | | | | | | | | | |
|---|---|---|---|---|---|---|---|---|---|---|---|
| Method | | Baseline | | | | | AliO | | | | |
| Metric | | MSE↓ | TAM↓ | MAE↓ | MAPE↓ | RMSE↓ | MSE↓ | TAM↓ | MAE↓ | MAPE↓ | RMSE↓ |
| ECL | 96 | 0.204±0.008 | 0.035±0.002 | 0.319±0.008 | 3.398±0.230 | 0.452±0.009 | **0.193±0.003** | **0.029±0.001** | **0.308±0.003** | **3.296±0.132** | **0.440±0.004** |
| | 192 | 0.219±0.005 | 0.042±0.002 | 0.329±0.003 | 3.416±0.185 | 0.468±0.005 | **0.215±0.003** | **0.038±0.004** | **0.324±0.004** | **3.402±0.091** | **0.464±0.004** |
| | 336 | 0.232±0.004 | 0.040±0.004 | 0.340±0.004 | 3.466±0.224 | 0.481±0.004 | **0.223±0.007** | **0.034±0.004** | **0.331±0.005** | **3.350±0.136** | **0.472±0.007** |
| | 720 | 0.260±0.013 | 0.046±0.007 | 0.362±0.009 | 3.501±0.118 | 0.509±0.013 | **0.253±0.008** | **0.042±0.002** | **0.356±0.008** | **3.404±0.035** | **0.503±0.008** |
| | AVG | 0.229±0.018 | 0.041±0.003 | 0.337±0.015 | 3.445±0.036 | 0.478±0.019 | **0.221±0.019** | **0.036±0.004** | **0.330±0.015** | **3.363±0.040** | **0.469±0.020** |
| ETTh1 | 24 | 0.408±0.008 | 0.063±0.005 | 0.427±0.002 | 10.853±0.413 | 0.639±0.006 | **0.373±0.008** | **0.051±0.002** | **0.416±0.007** | **10.747±0.477** | **0.611±0.007** |
| | 48 | 0.408±0.014 | 0.048±0.004 | 0.431±0.008 | 11.632±0.477 | 0.639±0.011 | **0.395±0.024** | **0.037±0.002** | **0.420±0.012** | **10.889±0.175** | **0.628±0.019** |
| | 168 | 0.502±0.028 | 0.051±0.012 | 0.482±0.016 | 11.810±0.142 | 0.708±0.020 | **0.445±0.014** | **0.036±0.005** | **0.451±0.010** | **11.573±0.593** | **0.667±0.011** |
| | 336 | 0.517±0.013 | **0.039±0.003** | 0.491±0.006 | **11.786±0.326** | 0.719±0.009 | **0.484±0.013** | 0.048±0.002 | **0.479±0.008** | 11.957±0.223 | **0.696±0.009** |
| | 720 | 0.549±0.016 | **0.047±0.002** | 0.527±0.008 | **12.533±0.659** | 0.741±0.011 | **0.521±0.009** | 0.057±0.009 | **0.516±0.010** | 13.217±1.015 | **0.722±0.007** |
| | AVG | 0.477±0.053 | 0.050±0.007 | 0.472±0.034 | 11.723±0.488 | 0.689±0.039 | **0.444±0.050** | **0.046±0.008** | **0.456±0.034** | **11.677±0.811** | **0.665±0.038** |
| ETTh2 | 24 | 0.280±0.009 | 0.054±0.004 | 0.356±0.006 | 1.524±0.038 | 0.529±0.008 | **0.265±0.003** | **0.034±0.003** | **0.340±0.003** | **1.392±0.028** | **0.515±0.003** |
| | 48 | 0.335±0.029 | 0.051±0.009 | 0.390±0.021 | 1.624±0.067 | 0.578±0.025 | **0.301±0.009** | **0.025±0.002** | **0.360±0.006** | **1.422±0.054** | **0.549±0.008** |
| | 168 | 0.442±0.008 | 0.044±0.009 | 0.448±0.009 | 1.868±0.150 | 0.665±0.006 | **0.421±0.008** | **0.027±0.002** | **0.427±0.005** | **1.580±0.084** | **0.649±0.006** |
| | 336 | 0.479±0.012 | 0.031±0.012 | 0.482±0.012 | 2.003±0.194 | 0.692±0.009 | **0.448±0.002** | **0.018±0.004** | **0.455±0.002** | **1.749±0.020** | **0.669±0.001** |
| | 720 | 0.470±0.022 | 0.025±0.005 | 0.483±0.013 | 2.273±0.166 | 0.685±0.016 | **0.450±0.005** | **0.020±0.000** | **0.466±0.003** | **2.044±0.024** | **0.670±0.004** |
| | AVG | 0.401±0.072 | 0.041±0.010 | 0.432±0.046 | 1.858±0.245 | 0.630±0.059 | **0.377±0.071** | **0.025±0.005** | **0.410±0.046** | **1.637±0.219** | **0.610±0.060** |
| ETTm1 | 24 | 0.377±0.015 | 0.092±0.003 | 0.415±0.005 | 2.563±0.005 | 0.614±0.012 | **0.351±0.023** | **0.084±0.001** | **0.394±0.011** | **2.455±0.019** | **0.592±0.019** |
| | 48 | 0.447±0.048 | 0.079±0.014 | 0.452±0.017 | 2.789±0.130 | 0.668±0.035 | **0.381±0.002** | **0.053±0.001** | **0.412±0.003** | **2.458±0.074** | **0.617±0.001** |
| | 96 | 0.493±0.054 | 0.074±0.010 | 0.478±0.019 | 2.834±0.081 | 0.701±0.038 | **0.440±0.029** | **0.055±0.006** | **0.445±0.012** | **2.579±0.078** | **0.663±0.022** |
| | 288 | 0.579±0.030 | 0.070±0.013 | 0.513±0.012 | 2.851±0.060 | 0.760±0.020 | **0.515±0.003** | **0.041±0.002** | **0.479±0.004** | **2.604±0.049** | **0.718±0.002** |
| | 672 | **0.544±0.013** | 0.036±0.001 | 0.501±0.005 | 2.788±0.062 | **0.737±0.009** | 0.554±0.013 | **0.031±0.002** | 0.501±0.009 | 2.728±0.044 | 0.744±0.009 |
| | AVG | 0.488±0.065 | 0.070±0.017 | 0.472±0.032 | 2.765±0.095 | 0.696±0.047 | **0.448±0.070** | **0.053±0.016** | **0.446±0.036** | **2.565±0.093** | **0.667±0.053** |
| ETTm2 | 24 | 0.161±0.013 | 0.078±0.010 | 0.268±0.010 | 1.081±0.039 | 0.401±0.016 | **0.148±0.005** | **0.057±0.005** | **0.259±0.005** | **1.054±0.041** | **0.385±0.007** |
| | 48 | 0.217±0.012 | 0.080±0.005 | 0.308±0.009 | 1.196±0.046 | 0.466±0.013 | **0.182±0.006** | **0.041±0.009** | **0.279±0.007** | **1.094±0.033** | **0.426±0.007** |
| | 96 | 0.252±0.029 | 0.058±0.008 | 0.324±0.015 | 1.302±0.024 | 0.502±0.028 | **0.217±0.003** | **0.030±0.006** | **0.298±0.004** | **1.158±0.019** | **0.466±0.003** |
| | 288 | 0.319±0.005 | 0.027±0.004 | 0.362±0.003 | 1.407±0.029 | 0.565±0.005 | **0.313±0.001** | **0.019±0.005** | **0.354±0.001** | **1.307±0.013** | **0.559±0.001** |
| | 672 | **0.408±0.002** | 0.017±0.003 | 0.410±0.001 | 1.577±0.014 | **0.639±0.001** | 0.409±0.003 | **0.012±0.004** | **0.409±0.002** | **1.538±0.047** | 0.640±0.002 |
| | AVG | 0.271±0.078 | 0.052±0.024 | 0.334±0.044 | 1.313±0.156 | 0.514±0.075 | **0.254±0.087** | **0.032±0.015** | **0.320±0.050** | **1.230±0.161** | **0.495±0.084** |

Table 13: Full results of the proposed AliO method on the ECL, ETTh1, ETTh2, ETTm1, ETTm2, Exchange, Traffic, and Weather datasets and Autoformer [37]. The results are reported in terms of MSE, TAM, MAE, MAPE, and RMSE. The best results are highlighted in **bold**. We use official GitHub code (https://github.com/thuml/Autoformer) to train the model with the official configuration. The average improvement of AliO over the baseline is MSE: 6.98%, TAM: 24.24%, MAE: 3.38%, MAPE: 3.76%, and RMSE: 3.12%. The maximum improvement of AliO over the baseline is MSE: 27.90%, TAM: 81.18%, MAE: 15.17%, MAPE: 15.42%, and RMSE: 8.58%. ECL, ETTh1, ETTh2, ETTm1, and ETTm2 are in Tab. 12.

| Models | | Autoformer | | | | | | | | | | |
| Method | | Baseline | | | | | AliO | | | | |
| Metric | | MSE↓ | TAM↓ | MAE↓ | MAPE↓ | RMSE↓ | MSE↓ | TAM↓ | MAE↓ | MAPE↓ | RMSE↓ |
|---|---|---|---|---|---|---|---|---|---|---|---|
| Exchange | 96 | 0.155±0.012 | 0.031±0.016 | 0.286±0.011 | 1.700±0.053 | 0.394±0.015 | **0.139±0.005** | **0.018±0.007** | **0.270±0.005** | **1.648±0.010** | **0.373±0.007** |
| | 192 | 0.284±0.017 | **0.014±0.004** | 0.389±0.009 | 2.345±0.025 | 0.532±0.016 | **0.275±0.012** | 0.015±0.008 | **0.379±0.009** | **2.338±0.025** | **0.525±0.011** |
| | 336 | 0.447±0.041 | 0.020±0.007 | 0.496±0.023 | 3.403±0.130 | 0.668±0.031 | **0.436±0.012** | **0.011±0.000** | **0.488±0.007** | **3.369±0.002** | **0.660±0.009** |
| | 720 | 1.110±0.035 | 0.037±0.044 | **0.814±0.022** | 6.640±0.123 | 1.053±0.017 | **0.800±0.011** | **0.007±0.002** | 1.079±0.022 | **6.627±0.094** | **1.039±0.011** |
| | AVG | 0.499±0.329 | 0.025±0.008 | **0.496±0.177** | 3.522±1.699 | 0.662±0.220 | **0.413±0.221** | **0.013±0.004** | 0.554±0.280 | **3.495±1.707** | **0.649±0.221** |
| Traffic | 96 | 0.637±0.018 | 0.051±0.005 | 0.400±0.014 | 4.222±0.261 | 0.798±0.011 | **0.613±0.009** | **0.041±0.003** | **0.375±0.002** | **3.782±0.014** | **0.783±0.006** |
| | 192 | 0.626±0.009 | 0.055±0.006 | 0.389±0.006 | **4.104±0.090** | 0.791±0.006 | **0.621±0.008** | **0.051±0.005** | **0.386±0.006** | 4.130±0.079 | **0.788±0.005** |
| | 336 | 0.623±0.006 | **0.042±0.002** | 0.386±0.005 | **4.207±0.091** | 0.789±0.004 | **0.614±0.003** | 0.048±0.006 | **0.379±0.003** | 4.313±0.167 | **0.784±0.002** |
| | 720 | 0.651±0.010 | 0.045±0.004 | 0.398±0.006 | 4.326±0.033 | 0.807±0.006 | **0.648±0.005** | **0.039±0.001** | **0.386±0.003** | **3.970±0.063** | **0.805±0.003** |
| | AVG | 0.634±0.010 | 0.048±0.005 | 0.393±0.005 | 4.215±0.070 | 0.796±0.006 | **0.624±0.013** | **0.045±0.004** | **0.382±0.004** | **4.049±0.175** | **0.790±0.008** |
| Weather | 96 | 0.269±0.032 | 0.069±0.011 | 0.340±0.023 | **13.016±1.557** | 0.518±0.031 | **0.224±0.009** | **0.051±0.003** | **0.288±0.011** | 13.076±0.455 | **0.474±0.010** |
| | 192 | 0.298±0.013 | 0.055±0.003 | 0.356±0.008 | **14.001±0.960** | 0.546±0.012 | **0.282±0.007** | **0.043±0.005** | **0.332±0.006** | 15.517±0.589 | **0.531±0.007** |
| | 336 | 0.367±0.026 | 0.046±0.006 | 0.398±0.020 | **13.587±2.218** | 0.606±0.021 | **0.338±0.011** | **0.045±0.014** | **0.368±0.008** | 14.002±0.378 | **0.581±0.010** |
| | 720 | 0.434±0.014 | 0.056±0.004 | 0.441±0.013 | **11.666±0.645** | 0.659±0.011 | **0.401±0.005** | **0.027±0.007** | **0.410±0.004** | 12.468±0.723 | **0.633±0.004** |
| | AVG | 0.342±0.057 | 0.056±0.007 | 0.384±0.035 | **13.068±0.788** | 0.582±0.049 | **0.311±0.058** | **0.042±0.008** | **0.350±0.040** | 13.766±1.028 | **0.555±0.053** |

Table 14: Full results of the proposed AliO method on the ECL, ETTh1, ETTh2, ETTm1, ETTm2, Exchange, ILI, Traffix, and Weather datasets and DLinear [40]. The results are reported in terms of MSE, TAM, MAE, MAPE, and RMSE. The best results are highlighted in **bold**. We use official GitHub code (https://github.com/vivva/DLinear) to train the model with the official configuration. The average improvement of AliO over the baseline is MSE: 5.62%, TAM: 36.21%, MAE: -3.82%, MAPE: 2.25%, and RMSE: 0.79%. The maximum improvement of AliO over the baseline is MSE: 36.07%, TAM: 77.73%, MAE: 8.21%, MAPE: 15.19%, and RMSE: 5.61%. Exchange ILI Traffic Weather are in Tab. 15.

| Models | | | DLinear | | | | | | | | | |
|---|---|---|---|---|---|---|---|---|---|---|---|---|
| Method | | | Baseline | | | | | AliO | | | | |
| Metric | | | MSE↓ | TAM↓ | MAE↓ | MAPE↓ | RMSE↓ | MSE↓ | TAM↓ | MAE↓ | MAPE↓ | RMSE↓ |
| ECL | 96 | | **0.140±0.000** | 0.017±0.000 | 0.237±0.000 | **2.149±0.005** | **0.374±0.000** | **0.140±0.000** | **0.013±0.000** | **0.237±0.000** | 2.149±0.003 | 0.374±0.000 |
| | 192 | | **0.153±0.000** | 0.013±0.000 | 0.250±0.000 | **2.300±0.003** | **0.391±0.000** | **0.153±0.000** | **0.010±0.000** | **0.250±0.000** | 2.301±0.001 | 0.391±0.000 |
| | 336 | | 0.169±0.000 | 0.011±0.000 | 0.268±0.000 | 2.283±0.004 | 0.411±0.000 | **0.169±0.000** | **0.008±0.000** | **0.267±0.000** | **2.283±0.001** | **0.411±0.000** |
| | 720 | | 0.204±0.000 | 0.011±0.001 | 0.301±0.000 | 2.457±0.002 | 0.451±0.000 | **0.203±0.000** | **0.008±0.001** | **0.300±0.000** | **2.451±0.004** | **0.451±0.000** |
| | AVG | | 0.166±0.021 | 0.013±0.002 | 0.264±0.021 | 2.297±0.098 | 0.407±0.026 | **0.166±0.021** | **0.010±0.002** | **0.264±0.021** | **2.296±0.096** | **0.407±0.026** |
| ETTh1 | 96 | | 0.371±0.000 | 0.028±0.001 | 0.395±0.001 | **8.691±0.011** | 0.609±0.000 | **0.369±0.000** | **0.019±0.000** | **0.392±0.000** | 8.743±0.003 | **0.608±0.000** |
| | 192 | | 0.408±0.002 | 0.028±0.002 | 0.419±0.002 | **8.481±0.081** | 0.638±0.002 | **0.404±0.000** | **0.015±0.000** | **0.413±0.000** | 8.496±0.006 | **0.635±0.000** |
| | 336 | | 0.435±0.002 | 0.024±0.005 | 0.439±0.003 | 8.545±0.053 | 0.660±0.001 | **0.432±0.000** | **0.013±0.000** | **0.435±0.000** | **8.489±0.001** | **0.658±0.000** |
| | 720 | | 0.474±0.001 | 0.022±0.003 | 0.493±0.001 | 9.459±0.056 | 0.689±0.001 | **0.469±0.000** | **0.010±0.000** | **0.488±0.000** | **9.429±0.001** | **0.685±0.000** |
| | AVG | | 0.422±0.034 | 0.025±0.003 | 0.436±0.032 | 8.794±0.350 | 0.649±0.026 | **0.419±0.033** | **0.014±0.003** | **0.432±0.032** | **8.789±0.343** | **0.646±0.025** |
| ETTh2 | 96 | | 0.291±0.003 | 0.069±0.016 | 0.354±0.002 | 1.359±0.048 | 0.539±0.003 | **0.290±0.006** | **0.046±0.010** | **0.353±0.006** | **1.321±0.028** | **0.538±0.006** |
| | 192 | | 0.381±0.008 | 0.049±0.005 | 0.416±0.006 | **1.369±0.007** | 0.617±0.007 | **0.369±0.007** | **0.023±0.004** | **0.405±0.006** | 1.376±0.022 | **0.607±0.006** |
| | 336 | | **0.438±0.011** | **0.039±0.001** | **0.456±0.007** | **1.494±0.008** | **0.662±0.008** | 0.454±0.009 | 0.050±0.027 | 0.465±0.004 | 1.497±0.041 | 0.674±0.007 |
| | 720 | | 0.687±0.044 | 0.057±0.020 | 0.586±0.018 | 1.605±0.033 | 0.828±0.026 | **0.672±0.025** | **0.013±0.002** | **0.577±0.013** | **1.544±0.019** | **0.820±0.015** |
| | AVG | | 0.449±0.131 | 0.054±0.010 | 0.453±0.076 | 1.457±0.090 | 0.662±0.095 | **0.446±0.128** | **0.033±0.014** | **0.450±0.075** | **1.434±0.080** | **0.660±0.093** |
| ETTm1 | 96 | | 0.300±0.001 | 0.031±0.002 | 0.344±0.001 | 2.021±0.011 | 0.548±0.001 | **0.296±0.000** | **0.014±0.000** | **0.338±0.000** | **1.983±0.001** | **0.544±0.000** |
| | 192 | | 0.338±0.000 | 0.026±0.001 | 0.369±0.001 | 2.079±0.008 | 0.581±0.000 | **0.333±0.000** | **0.010±0.000** | **0.361±0.000** | **2.068±0.001** | **0.577±0.000** |
| | 336 | | 0.369±0.000 | 0.020±0.001 | 0.386±0.001 | 2.131±0.008 | 0.608±0.000 | **0.367±0.000** | **0.008±0.000** | **0.382±0.000** | **2.124±0.001** | **0.606±0.000** |
| | 720 | | 0.427±0.000 | 0.019±0.001 | 0.422±0.001 | 2.240±0.018 | 0.653±0.000 | **0.422±0.000** | **0.006±0.000** | **0.416±0.000** | **2.237±0.001** | **0.650±0.000** |
| | AVG | | 0.358±0.041 | 0.024±0.004 | 0.380±0.025 | 2.118±0.072 | 0.597±0.034 | **0.354±0.042** | **0.010±0.002** | **0.374±0.025** | **2.103±0.082** | **0.594±0.035** |
| ETTm2 | 96 | | 0.171±0.002 | 0.034±0.002 | 0.265±0.004 | **1.047±0.009** | 0.413±0.002 | **0.166±0.001** | **0.018±0.001** | **0.258±0.001** | 1.056±0.003 | **0.407±0.001** |
| | 192 | | 0.233±0.004 | 0.031±0.004 | 0.312±0.004 | **1.153±0.024** | 0.483±0.004 | **0.229±0.002** | **0.015±0.001** | **0.307±0.002** | 1.162±0.005 | **0.478±0.002** |
| | 336 | | 0.311±0.022 | 0.053±0.023 | 0.366±0.017 | 1.325±0.085 | 0.557±0.019 | **0.284±0.005** | **0.026±0.007** | **0.344±0.004** | **1.274±0.018** | **0.533±0.005** |
| | 720 | | 0.443±0.012 | 0.030±0.003 | 0.453±0.008 | **1.315±0.019** | 0.665±0.009 | **0.394±0.004** | **0.020±0.005** | **0.415±0.003** | 1.384±0.007 | **0.628±0.003** |
| | AVG | | 0.289±0.091 | 0.037±0.009 | 0.349±0.062 | **1.210±0.104** | 0.530±0.083 | **0.268±0.075** | **0.020±0.004** | **0.331±0.051** | 1.219±0.110 | **0.512±0.072** |

Table 15: Full results of the proposed AliO method on the ECL, ETTh1, ETTh2, ETTm1, ETTm2, Exchange, ILI, Traffix, and Weather datasets and DLinear [40]. The results are reported in terms of MSE, TAM, MAE, MAPE, and RMSE. The best results are highlighted in **bold**. We use official GitHub code (https://github.com/vivva/DLinear) to train the model with the official configuration. The average improvement of AliO over the baseline is MSE: 5.62%, TAM: 36.21%, MAE: -3.82%, MAPE: 2.25%, and RMSE: 0.79%. The maximum improvement of AliO over the baseline is MSE: 36.07%, TAM: 77.73%, MAE: 8.21%, MAPE: 15.19%, and RMSE: 5.61%. ECL, ETTh1, ETTh2, ETTm1, and ETTm2 are in Tab. 14.

| Models | | DLinear | | | | | | | | | |
| --- | --- | --- | --- | --- | --- | --- | --- | --- | --- | --- | --- |
| Method | | Baseline | | | | | AliO | | | | |
| Metric | | MSE↓ | TAM↓ | MAE↓ | MAPE↓ | RMSE↓ | MSE↓ | TAM↓ | MAE↓ | MAPE↓ | RMSE↓ |
| Exchange | 96 | 0.086±0.006 | 0.017±0.000 | 0.208±0.008 | 1.198±0.036 | 0.293±0.010 | **0.078±0.000** | **0.015±0.000** | **0.201±0.000** | **1.120±0.001** | **0.280±0.000** |
| | 192 | 0.160±0.002 | 0.015±0.001 | 0.295±0.002 | **1.584±0.033** | 0.400±0.003 | **0.157±0.002** | **0.014±0.001** | **0.291±0.002** | 1.594±0.019 | **0.396±0.002** |
| | 336 | 0.330±0.032 | **0.012±0.005** | 0.434±0.016 | **2.120±0.139** | 0.574±0.028 | **0.320±0.034** | 0.012±0.003 | **0.425±0.022** | 2.222±0.207 | **0.565±0.030** |
| | 720 | 0.833±0.129 | **0.033±0.017** | 0.689±0.050 | 2.298±0.441 | 0.910±0.072 | **0.822±0.135** | 0.054±0.033 | **0.679±0.051** | **2.186±0.258** | **0.904±0.074** |
| | AVG | 0.352±0.260 | **0.019±0.007** | 0.407±0.163 | 1.800±0.390 | 0.544±0.209 | **0.344±0.259** | 0.024±0.016 | **0.399±0.161** | **1.781±0.407** | **0.536±0.210** |
| ILI | 24 | 2.335±0.149 | 0.196±0.039 | 1.078±0.060 | 4.052±0.445 | 1.527±0.049 | **2.229±0.019** | **0.131±0.003** | **1.036±0.005** | **3.689±0.059** | **1.493±0.006** |
| | 36 | **2.072±0.026** | **0.144±0.009** | **1.018±0.012** | **2.497±0.075** | **1.439±0.009** | 2.076±0.036 | 0.153±0.063 | 1.024±0.005 | 2.507±0.190 | 1.441±0.013 |
| | 48 | 2.269±0.077 | 0.136±0.020 | 1.091±0.032 | 2.685±0.249 | 1.506±0.026 | **2.161±0.016** | **0.078±0.003** | **1.049±0.003** | **2.278±0.016** | **1.470±0.005** |
| | 60 | 2.315±0.022 | 0.132±0.028 | 1.088±0.006 | 2.542±0.176 | 1.521±0.007 | **2.280±0.080** | **0.112±0.021** | **1.078±0.019** | **2.412±0.050** | **1.510±0.027** |
| | AVG | 2.247±0.093 | 0.152±0.023 | 1.069±0.027 | 2.944±0.576 | 1.498±0.031 | **2.187±0.069** | **0.119±0.025** | **1.047±0.018** | **2.721±0.505** | **1.478±0.023** |
| Traffic | 96 | 0.412±0.001 | 0.024±0.001 | **0.286±0.001** | 3.123±0.030 | **0.642±0.001** | **0.269±0.000** | **0.007±0.000** | 0.426±0.000 | **2.749±0.000** | 0.653±0.000 |
| | 192 | 0.424±0.000 | 0.026±0.003 | **0.291±0.000** | 3.073±0.018 | **0.651±0.000** | **0.276±0.000** | **0.007±0.002** | 0.430±0.000 | **2.718±0.006** | 0.656±0.000 |
| | 336 | 0.438±0.001 | 0.023±0.000 | **0.299±0.001** | 3.047±0.008 | **0.662±0.000** | **0.282±0.000** | **0.006±0.002** | 0.440±0.000 | **2.690±0.006** | 0.664±0.000 |
| | 720 | 0.468±0.001 | 0.026±0.004 | **0.319±0.001** | 3.139±0.022 | 0.684±0.001 | **0.299±0.000** | **0.009±0.002** | 0.466±0.000 | **2.769±0.005** | **0.683±0.000** |
| | AVG | 0.436±0.019 | 0.025±0.001 | **0.299±0.011** | 3.095±0.033 | **0.660±0.014** | **0.282±0.010** | **0.007±0.001** | 0.441±0.014 | **2.731±0.027** | 0.664±0.010 |
| Weather | 96 | **0.175±0.001** | 0.009±0.001 | 0.237±0.003 | **10.585±0.247** | **0.419±0.001** | 0.175±0.001 | **0.007±0.000** | **0.236±0.002** | 10.679±0.122 | 0.419±0.001 |
| | 192 | **0.216±0.001** | 0.007±0.000 | 0.275±0.001 | **10.767±0.080** | **0.465±0.001** | 0.217±0.000 | **0.005±0.000** | **0.273±0.000** | 10.893±0.009 | 0.466±0.000 |
| | 336 | 0.264±0.002 | 0.007±0.000 | 0.316±0.003 | **11.698±0.232** | 0.513±0.002 | **0.262±0.000** | **0.004±0.000** | **0.310±0.000** | 11.930±0.005 | **0.511±0.000** |
| | 720 | 0.326±0.001 | 0.006±0.000 | 0.367±0.002 | **10.920±0.188** | 0.571±0.001 | **0.324±0.000** | **0.003±0.000** | **0.362±0.000** | 11.236±0.009 | **0.569±0.000** |
| | AVG | 0.245±0.050 | 0.007±0.001 | 0.299±0.043 | **10.993±0.380** | 0.492±0.051 | **0.244±0.049** | **0.005±0.001** | **0.295±0.041** | 11.184±0.424 | **0.491±0.050** |

Table 16: Full results of the proposed AliO method on the ECL, ETTh1, ETTh2, ETTm1, ETTm2, ILI, Traffic, and Weather datasets and PatchTST [33]. The results are reported in terms of MSE, TAM, MAE, MAPE, and RMSE. The best results are highlighted in **bold**. We use official GitHub code (https://github.com/yuqinie98/PatchTST) to train the model with the official configuration. The average improvement of AliO over the baseline is MSE: 2.58%, TAM: 35.69%, MAE: 2.42%, MAPE: 4.40%, and RMSE: 1.34%. The maximum improvement of AliO over the baseline is MSE: 20.93%, TAM: 60.37%, MAE: 14.44%, MAPE: 29.81%, and RMSE: 11.01%. ILI, Traffic, and Weather are in Tab. 17.

| Models | | PatchTST | | | | | | | | | |
|---|---|---|---|---|---|---|---|---|---|---|---|
| Method | | Baseline | | | | | AliO | | | | |
| Metric | | MSE↓ | TAM↓ | MAE↓ | MAPE↓ | RMSE↓ | MSE↓ | TAM↓ | MAE↓ | MAPE↓ | RMSE↓ |
| ECL | 96 | 0.130±0.000 | 0.030±0.000 | 0.223±0.000 | 2.332±0.034 | 0.360±0.000 | 0.130±0.000 | 0.016±0.000 | 0.221±0.000 | 2.276±0.006 | 0.361±0.000 |
| | 192 | 0.149±0.001 | 0.032±0.002 | 0.241±0.001 | 2.488±0.040 | 0.386±0.001 | 0.148±0.001 | 0.021±0.000 | 0.238±0.000 | 2.462±0.004 | 0.385±0.001 |
| | 336 | 0.166±0.000 | 0.037±0.001 | 0.260±0.001 | 2.493±0.008 | 0.408±0.001 | 0.165±0.001 | 0.028±0.001 | 0.257±0.000 | 2.468±0.008 | 0.406±0.001 |
| | 720 | 0.203±0.001 | 0.049±0.002 | 0.293±0.001 | 2.583±0.027 | 0.451±0.001 | 0.202±0.001 | 0.030±0.000 | 0.289±0.001 | 2.554±0.012 | 0.449±0.001 |
| | AVG | 0.162±0.024 | 0.037±0.007 | 0.254±0.023 | 2.474±0.081 | 0.401±0.030 | 0.161±0.024 | 0.024±0.005 | 0.251±0.023 | 2.440±0.091 | 0.400±0.029 |
| ETTh1 | 96 | 0.379±0.000 | 0.065±0.002 | 0.401±0.000 | 9.413±0.032 | 0.616±0.000 | 0.378±0.001 | 0.053±0.002 | 0.401±0.001 | 9.358±0.026 | 0.615±0.001 |
| | 192 | 0.412±0.000 | 0.063±0.002 | 0.420±0.000 | 9.490±0.037 | 0.642±0.000 | 0.411±0.000 | 0.050±0.002 | 0.419±0.000 | 9.454±0.027 | 0.641±0.000 |
| | 336 | 0.435±0.002 | 0.072±0.007 | 0.436±0.001 | 9.524±0.066 | 0.659±0.001 | 0.432±0.001 | 0.045±0.002 | 0.434±0.000 | 9.425±0.038 | 0.657±0.000 |
| | 720 | 0.448±0.003 | 0.103±0.013 | 0.465±0.002 | 10.184±0.163 | 0.669±0.002 | 0.439±0.003 | 0.042±0.005 | 0.461±0.002 | 9.896±0.092 | 0.663±0.002 |
| | AVG | 0.418±0.023 | 0.076±0.014 | 0.431±0.021 | 9.653±0.277 | 0.646±0.018 | 0.415±0.021 | 0.048±0.004 | 0.429±0.020 | 9.533±0.190 | 0.644±0.017 |
| ETTh2 | 96 | 0.276±0.000 | 0.067±0.001 | 0.338±0.000 | 1.378±0.002 | 0.525±0.000 | 0.275±0.000 | 0.055±0.001 | 0.337±0.000 | 1.368±0.003 | 0.525±0.000 |
| | 192 | 0.336±0.000 | 0.062±0.001 | 0.378±0.000 | 1.510±0.003 | 0.580±0.000 | 0.337±0.000 | 0.051±0.001 | 0.377±0.001 | 1.495±0.024 | 0.580±0.000 |
| | 336 | 0.361±0.000 | 0.060±0.002 | 0.401±0.001 | 1.700±0.011 | 0.601±0.000 | 0.360±0.000 | 0.048±0.002 | 0.399±0.002 | 1.672±0.027 | 0.600±0.000 |
| | 720 | 0.391±0.000 | 0.055±0.002 | 0.429±0.000 | 2.013±0.003 | 0.625±0.000 | 0.391±0.001 | 0.044±0.002 | 0.429±0.001 | 2.010±0.003 | 0.625±0.001 |
| | AVG | 0.341±0.038 | 0.061±0.004 | 0.387±0.030 | 1.650±0.214 | 0.583±0.033 | 0.341±0.038 | 0.050±0.004 | 0.386±0.030 | 1.636±0.216 | 0.583±0.033 |
| ETTm1 | 96 | 0.290±0.001 | 0.052±0.001 | 0.342±0.001 | 2.175±0.015 | 0.538±0.001 | 0.285±0.001 | 0.029±0.000 | 0.334±0.001 | 2.102±0.015 | 0.534±0.001 |
| | 192 | 0.334±0.002 | 0.049±0.001 | 0.370±0.002 | 2.302±0.014 | 0.578±0.002 | 0.329±0.002 | 0.035±0.001 | 0.364±0.001 | 2.241±0.001 | 0.574±0.001 |
| | 336 | 0.366±0.001 | 0.049±0.001 | 0.391±0.001 | 2.350±0.014 | 0.605±0.001 | 0.367±0.002 | 0.026±0.000 | 0.386±0.001 | 2.284±0.008 | 0.606±0.002 |
| | 720 | 0.418±0.005 | 0.052±0.002 | 0.423±0.004 | 2.452±0.026 | 0.646±0.004 | 0.415±0.002 | 0.037±0.001 | 0.419±0.001 | 2.459±0.016 | 0.644±0.001 |
| | AVG | 0.352±0.042 | 0.050±0.001 | 0.382±0.026 | 2.320±0.089 | 0.592±0.035 | 0.349±0.043 | 0.032±0.004 | 0.376±0.028 | 2.271±0.114 | 0.589±0.036 |
| ETTm2 | 96 | 0.164±0.001 | 0.051±0.001 | 0.252±0.000 | 1.057±0.004 | 0.405±0.001 | 0.162±0.001 | 0.036±0.001 | 0.251±0.000 | 1.053±0.001 | 0.403±0.000 |
| | 192 | 0.220±0.000 | 0.059±0.002 | 0.292±0.001 | 1.194±0.004 | 0.469±0.000 | 0.218±0.001 | 0.033±0.001 | 0.289±0.001 | 1.194±0.006 | 0.467±0.001 |
| | 336 | 0.275±0.001 | 0.063±0.000 | 0.329±0.000 | 1.318±0.001 | 0.525±0.001 | 0.274±0.000 | 0.045±0.001 | 0.327±0.000 | 1.316±0.003 | 0.523±0.000 |
| | 720 | 0.368±0.002 | 0.063±0.002 | 0.384±0.001 | 1.478±0.003 | 0.606±0.001 | 0.366±0.001 | 0.044±0.000 | 0.383±0.001 | 1.480±0.002 | 0.605±0.001 |
| | AVG | 0.257±0.067 | 0.059±0.004 | 0.314±0.043 | 1.262±0.139 | 0.501±0.066 | 0.255±0.067 | 0.039±0.005 | 0.312±0.044 | 1.261±0.141 | 0.500±0.067 |

Table 17: Full results of the proposed AliO method on the ECL, ETTh1, ETTh2, ETTm1, ETTm2, ILI, Traffix, and Weather datasets and PatchTST [33]. The results are reported in terms of MSE, TAM, MAE, MAPE, and RMSE. The best results are highlighted in **bold**. We use official GitHub code (https://github.com/yuqinie98/PatchTST) to train the model with the official configuration. The average improvement of AliO over the baseline is MSE: 2.58%, TAM: 35.69%, MAE: 2.42%, MAPE: 4.40%, and RMSE: 1.34%. The maximum improvement of AliO over the baseline is MSE: 20.93%, TAM: 60.37%, MAE: 14.44%, MAPE: 29.81%, and RMSE: 11.01%. ECL, ETTh1, ETTh2, ETTm1, and ETTm2 are in Tab. 16.

| Models | | PatchTST | | | | | | | | | |
| Method | | Baseline | | | | | AliO | | | | |
| Metric | | MSE↓ | TAM↓ | MAE↓ | MAPE↓ | RMSE↓ | MSE↓ | TAM↓ | MAE↓ | MAPE↓ | RMSE↓ |
|---|---|---|---|---|---|---|---|---|---|---|---|
| ILI | 24 | 2.132±0.179 | 0.128±0.015 | 0.917±0.064 | 4.121±0.100 | 1.459±0.062 | **1.686±0.053** | **0.056±0.003** | **0.811±0.011** | **3.004±0.048** | **1.298±0.020** |
| | 36 | 1.559±0.045 | 0.106±0.017 | 0.830±0.013 | 2.305±0.216 | 1.248±0.018 | **1.391±0.047** | **0.044±0.002** | **0.767±0.004** | **1.751±0.077** | **1.179±0.020** |
| | 48 | 1.736±0.078 | 0.099±0.009 | 0.900±0.021 | 2.306±0.099 | 1.317±0.030 | **1.431±0.109** | **0.042±0.001** | **0.774±0.034** | **1.682±0.049** | **1.195±0.045** |
| | 60 | 1.824±0.140 | 0.107±0.005 | 0.923±0.046 | 2.476±0.098 | 1.349±0.053 | **1.479±0.097** | **0.042±0.002** | **0.790±0.034** | **1.738±0.096** | **1.216±0.040** |
| | AVG | 1.813±0.186 | 0.110±0.010 | 0.893±0.033 | 2.802±0.684 | 1.343±0.068 | **1.497±0.101** | **0.046±0.005** | **0.785±0.015** | **2.044±0.497** | **1.222±0.041** |
| Traffic | 96 | 0.358±0.000 | 0.040±0.000 | 0.245±0.000 | 2.553±0.018 | 0.598±0.000 | **0.357±0.000** | **0.029±0.001** | **0.242±0.000** | **2.472±0.005** | **0.598±0.000** |
| | 192 | **0.378±0.001** | 0.039±0.000 | 0.254±0.000 | 2.570±0.022 | **0.615±0.001** | 0.378±0.000 | **0.022±0.000** | **0.248±0.000** | **2.482±0.001** | 0.615±0.000 |
| | 336 | 0.391±0.001 | 0.040±0.001 | 0.262±0.001 | 2.620±0.004 | 0.626±0.001 | **0.390±0.001** | **0.027±0.001** | **0.257±0.001** | **2.542±0.003** | **0.625±0.000** |
| | 720 | 0.431±0.000 | 0.044±0.001 | 0.285±0.000 | 2.845±0.018 | 0.657±0.000 | **0.427±0.001** | **0.029±0.000** | **0.278±0.000** | **2.707±0.017** | **0.653±0.001** |
| | AVG | 0.390±0.024 | 0.041±0.002 | 0.262±0.013 | 2.647±0.105 | 0.624±0.019 | **0.388±0.023** | **0.027±0.003** | **0.256±0.012** | **2.551±0.084** | **0.623±0.018** |
| Weather | 96 | 0.152±0.002 | 0.020±0.000 | 0.201±0.003 | **11.697±0.303** | 0.390±0.003 | **0.151±0.000** | **0.015±0.000** | **0.197±0.000** | 11.913±0.090 | **0.388±0.000** |
| | 192 | 0.196±0.001 | 0.018±0.001 | 0.242±0.001 | **13.084±0.650** | 0.443±0.002 | **0.196±0.001** | **0.013±0.000** | **0.240±0.001** | 13.325±0.405 | **0.442±0.001** |
| | 336 | 0.247±0.000 | 0.017±0.000 | 0.282±0.000 | 14.374±0.254 | 0.497±0.000 | **0.246±0.000** | **0.012±0.000** | **0.279±0.001** | **14.242±0.366** | **0.496±0.000** |
| | 720 | **0.320±0.001** | 0.016±0.000 | 0.334±0.001 | **13.976±0.119** | **0.566±0.000** | 0.321±0.000 | **0.007±0.000** | **0.329±0.000** | 14.243±0.026 | 0.566±0.000 |
| | AVG | 0.229±0.056 | 0.018±0.001 | 0.265±0.044 | **13.283±0.919** | 0.474±0.058 | **0.228±0.056** | **0.012±0.003** | **0.261±0.044** | 13.431±0.853 | **0.473±0.059** |

Table 18: Full results of the proposed AliO method on the ECL, ETTh1, ETTh2, ETTm1, ETTm2, Exchange, Traffic, ILI, and Weather datasets and TimesNet [36]. The results are reported in terms of MSE, TAM, MAE, MAPE, and RMSE. The best results are highlighted in **bold**. We use official GitHub code (https://github.com/thuml/TimesNet) to train the model with the official configuration. The average improvement of AliO over the baseline is MSE: 3.42%, TAM: 33.36%, MAE: 1.81%, MAPE: 4.01%, and RMSE: 1.76%. The maximum improvement of AliO over the baseline is MSE: 29.03%, TAM: 51.96%, MAE: 9.34%, MAPE: 23.09%, and RMSE: 16.56%. Exchange, Traffic, ILI, and Weather are in Tab. 19.

| Models | | TimesNet | | | | | | | | | |
|---|---|---|---|---|---|---|---|---|---|---|---|
| Method | | Baseline | | | | | AliO | | | | |
| Metric | | MSE↓ | TAM↓ | MAE↓ | MAPE↓ | RMSE↓ | MSE↓ | TAM↓ | MAE↓ | MAPE↓ | RMSE↓ |
| ECL | 96 | 0.168±0.001 | 0.038±0.001 | 0.271±0.001 | 2.666±0.048 | 0.409±0.001 | **0.166±0.001** | **0.025±0.000** | **0.269±0.001** | **2.563±0.067** | **0.408±0.001** |
| | 192 | 0.186±0.001 | 0.038±0.002 | 0.288±0.002 | 2.787±0.097 | 0.431±0.001 | **0.183±0.001** | **0.023±0.000** | **0.284±0.001** | **2.712±0.013** | **0.428±0.002** |
| | 336 | 0.202±0.005 | 0.041±0.002 | 0.302±0.003 | **2.944±0.075** | 0.449±0.006 | **0.196±0.001** | **0.025±0.001** | **0.297±0.002** | 2.944±0.091 | **0.443±0.002** |
| | 720 | 0.228±0.012 | 0.044±0.005 | 0.322±0.011 | **2.883±0.092** | 0.477±0.013 | **0.219±0.002** | **0.039±0.003** | **0.313±0.001** | 2.933±0.027 | **0.468±0.002** |
| | AVG | 0.196±0.020 | 0.040±0.002 | 0.296±0.017 | 2.820±0.094 | 0.442±0.022 | **0.191±0.017** | **0.028±0.006** | **0.291±0.014** | **2.788±0.143** | **0.436±0.020** |
| ETTh1 | 96 | **0.409±0.010** | 0.055±0.001 | **0.425±0.006** | **10.015±0.311** | **0.640±0.008** | 0.410±0.010 | **0.045±0.001** | 0.426±0.006 | 10.036±0.280 | 0.641±0.008 |
| | 192 | 0.469±0.006 | 0.054±0.003 | 0.460±0.004 | 10.999±0.070 | 0.685±0.004 | **0.461±0.005** | **0.030±0.002** | **0.455±0.001** | **10.429±0.286** | **0.679±0.003** |
| | 336 | 0.507±0.013 | 0.046±0.003 | 0.478±0.008 | 10.926±0.266 | 0.712±0.009 | **0.497±0.003** | **0.027±0.001** | **0.472±0.002** | **10.342±0.156** | **0.705±0.002** |
| | 720 | 0.521±0.006 | 0.046±0.003 | 0.497±0.003 | 10.981±0.954 | 0.722±0.004 | **0.502±0.007** | **0.027±0.001** | **0.487±0.003** | **10.451±0.332** | **0.708±0.005** |
| | AVG | 0.476±0.039 | 0.050±0.004 | 0.465±0.024 | 10.730±0.370 | 0.689±0.029 | **0.468±0.033** | **0.032±0.007** | **0.460±0.020** | **10.314±0.148** | **0.683±0.024** |
| ETTh2 | 96 | 0.328±0.014 | 0.067±0.006 | 0.370±0.010 | 1.607±0.053 | 0.573±0.012 | **0.318±0.003** | **0.041±0.006** | **0.362±0.002** | **1.500±0.021** | **0.564±0.003** |
| | 192 | 0.429±0.030 | 0.089±0.034 | 0.424±0.019 | 1.646±0.069 | 0.655±0.023 | **0.398±0.011** | **0.044±0.002** | **0.408±0.005** | **1.626±0.040** | **0.631±0.008** |
| | 336 | 0.452±0.013 | 0.085±0.023 | 0.453±0.009 | **1.826±0.034** | 0.672±0.010 | **0.448±0.002** | **0.051±0.003** | **0.447±0.002** | 1.839±0.052 | **0.669±0.001** |
| | 720 | 0.454±0.006 | 0.075±0.025 | 0.463±0.006 | 2.177±0.158 | 0.673±0.005 | **0.446±0.016** | **0.036±0.005** | **0.459±0.010** | **2.064±0.132** | **0.668±0.012** |
| | AVG | 0.416±0.046 | 0.079±0.008 | 0.428±0.032 | 1.814±0.201 | 0.643±0.037 | **0.402±0.047** | **0.043±0.005** | **0.419±0.034** | **1.757±0.192** | **0.633±0.038** |
| ETTm1 | 96 | 0.334±0.005 | 0.052±0.001 | 0.374±0.003 | 2.366±0.011 | 0.578±0.004 | **0.328±0.005** | **0.029±0.001** | **0.368±0.002** | **2.308±0.045** | **0.572±0.004** |
| | 192 | 0.386±0.000 | 0.045±0.001 | 0.399±0.001 | 2.478±0.070 | 0.622±0.000 | **0.381±0.001** | **0.033±0.002** | **0.397±0.001** | **2.458±0.045** | **0.618±0.001** |
| | 336 | 0.429±0.007 | 0.044±0.002 | 0.427±0.003 | 2.593±0.091 | 0.655±0.005 | **0.409±0.003** | **0.022±0.001** | **0.417±0.001** | **2.483±0.014** | **0.640±0.003** |
| | 720 | 0.499±0.004 | 0.042±0.002 | 0.465±0.003 | 2.801±0.051 | 0.706±0.003 | **0.475±0.001** | **0.020±0.001** | **0.452±0.002** | **2.686±0.023** | **0.689±0.001** |
| | AVG | 0.412±0.054 | 0.045±0.003 | 0.416±0.030 | 2.560±0.144 | 0.640±0.042 | **0.398±0.048** | **0.026±0.005** | **0.409±0.027** | **2.484±0.121** | **0.630±0.038** |
| ETTm2 | 96 | 0.186±0.000 | 0.040±0.002 | 0.266±0.001 | 1.181±0.013 | 0.431±0.000 | **0.186±0.002** | **0.030±0.001** | **0.265±0.001** | **1.155±0.026** | **0.431±0.002** |
| | 192 | 0.255±0.002 | 0.037±0.001 | 0.308±0.002 | **1.290±0.020** | 0.505±0.002 | **0.253±0.001** | **0.020±0.001** | **0.307±0.002** | 1.292±0.036 | **0.503±0.001** |
| | 336 | **0.315±0.003** | 0.033±0.003 | **0.345±0.001** | 1.351±0.017 | **0.561±0.001** | 0.317±0.004 | **0.020±0.002** | 0.348±0.002 | **1.335±0.015** | 0.563±0.003 |
| | 720 | 0.430±0.001 | 0.034±0.004 | 0.410±0.001 | 1.584±0.039 | 0.656±0.001 | **0.423±0.001** | **0.017±0.001** | **0.406±0.000** | **1.545±0.021** | **0.650±0.000** |
| | AVG | 0.296±0.080 | 0.036±0.003 | 0.332±0.047 | 1.352±0.132 | 0.538±0.073 | **0.295±0.078** | **0.022±0.005** | **0.331±0.047** | **1.332±0.125** | **0.537±0.072** |

Table 19: Full results of the proposed AliO method on the ECL, ETTh1, ETTh2, ETTm1, ETTm2, Exchange, Traffic, ILI, and Weather datasets and TimesNet [36]. The results are reported in terms of MSE, TAM, MAE, MAPE, and RMSE. The best results are highlighted in **bold**. We use official GitHub code (https://github.com/thuml/TimesNet) to train the model with the official configuration. The average improvement of AliO over the baseline is MSE: 3.42%, TAM: 33.36%, MAE: 1.81%, MAPE: 4.01%, and RMSE: 1.76%. The maximum improvement of AliO over the baseline is MSE: 29.03%, TAM: 51.96%, MAE: 9.34%, MAPE: 23.09%, and RMSE: 16.56%. ECL, ETTh1, ETTh2, ETTm1, and ETTm2 are in Tab. 18.

| Models | | TimesNet | | | | | | | | | |
|---|---|---|---|---|---|---|---|---|---|---|---|
| Method | | Baseline | | | | | AliO | | | | |
| Metric | | MSE↓ | TAM↓ | MAE↓ | MAPE↓ | RMSE↓ | MSE↓ | TAM↓ | MAE↓ | MAPE↓ | RMSE↓ |
| Exchange | 96 | 0.112±0.001 | 0.038±0.004 | 0.241±0.001 | 1.405±0.100 | 0.334±0.002 | **0.105±0.002** | **0.023±0.001** | **0.232±0.001** | **1.300±0.041** | **0.324±0.003** |
| | 192 | 0.217±0.008 | 0.036±0.004 | 0.337±0.005 | **1.951±0.061** | 0.466±0.009 | **0.207±0.002** | **0.020±0.000** | **0.330±0.001** | 2.002±0.054 | **0.455±0.003** |
| | 336 | 0.366±0.008 | 0.031±0.003 | 0.440±0.006 | **3.197±0.076** | 0.605±0.007 | **0.364±0.005** | **0.029±0.004** | **0.439±0.004** | 3.249±0.070 | **0.603±0.004** |
| | 720 | 0.964±0.025 | 0.033±0.005 | 0.746±0.012 | 6.519±0.188 | 0.982±0.013 | **0.944±0.034** | **0.024±0.004** | **0.740±0.015** | **6.473±0.189** | **0.972±0.017** |
| | AVG | 0.415±0.295 | 0.035±0.003 | 0.441±0.170 | 3.268±1.777 | 0.597±0.217 | **0.405±0.290** | **0.024±0.003** | **0.435±0.171** | **3.256±1.775** | **0.588±0.217** |
| Traffic | 96 | 0.593±0.007 | 0.042±0.001 | 0.313±0.002 | 2.957±0.073 | 0.770±0.004 | **0.589±0.003** | **0.028±0.001** | **0.310±0.001** | **2.798±0.051** | **0.767±0.002** |
| | 192 | 0.618±0.005 | 0.041±0.004 | 0.325±0.004 | 3.115±0.107 | 0.786±0.003 | **0.614±0.003** | **0.025±0.001** | **0.320±0.001** | **2.882±0.053** | **0.783±0.002** |
| | 336 | 0.631±0.008 | 0.041±0.005 | 0.334±0.006 | 3.420±0.219 | 0.794±0.005 | **0.448±0.128** | **0.026±0.001** | **0.303±0.018** | **2.712±0.320** | **0.663±0.092** |
| | 720 | 0.661±0.002 | 0.035±0.002 | 0.347±0.001 | 3.477±0.016 | 0.813±0.001 | **0.653±0.002** | **0.032±0.005** | **0.344±0.002** | **3.456±0.094** | **0.808±0.001** |
| | AVG | 0.626±0.022 | 0.040±0.003 | 0.330±0.011 | 3.242±0.192 | 0.791±0.014 | **0.576±0.069** | **0.028±0.003** | **0.319±0.014** | **2.962±0.261** | **0.755±0.050** |
| ILI | 24 | 2.759±0.839 | 0.371±0.026 | 0.972±0.077 | 4.452±0.551 | 1.643±0.246 | **2.239±0.347** | **0.256±0.011** | **0.924±0.045** | **3.424±0.129** | **1.492±0.114** |
| | 36 | 1.949±0.033 | 0.330±0.031 | 0.919±0.006 | 2.976±0.224 | 1.396±0.012 | **1.734±0.025** | **0.262±0.027** | **0.846±0.007** | **2.508±0.167** | **1.317±0.010** |
| | 48 | 1.976±0.112 | 0.305±0.030 | 0.896±0.029 | 2.906±0.050 | 1.405±0.040 | **1.911±0.033** | **0.263±0.016** | **0.862±0.015** | **2.686±0.047** | **1.382±0.012** |
| | 60 | 2.002±0.098 | 0.289±0.006 | 0.919±0.033 | 3.060±0.179 | 1.414±0.035 | **1.970±0.054** | **0.251±0.021** | **0.899±0.002** | **2.779±0.123** | **1.403±0.019** |
| | AVG | 2.171±0.304 | 0.324±0.028 | 0.926±0.025 | 3.348±0.572 | 1.465±0.092 | **1.963±0.162** | **0.258±0.004** | **0.883±0.027** | **2.849±0.309** | **1.399±0.056** |
| Weather | 96 | **0.174±0.003** | 0.021±0.001 | 0.223±0.002 | 11.767±0.301 | **0.417±0.004** | 0.174±0.003 | **0.017±0.000** | 0.223±0.001 | **11.659±0.718** | 0.417±0.003 |
| | 192 | **0.231±0.005** | 0.017±0.002 | **0.271±0.004** | 14.158±0.953 | **0.481±0.006** | 0.237±0.007 | **0.014±0.001** | 0.276±0.006 | **13.831±0.810** | 0.487±0.007 |
| | 336 | 0.285±0.002 | 0.020±0.002 | 0.307±0.001 | **13.782±0.351** | 0.534±0.002 | **0.280±0.001** | **0.011±0.001** | **0.302±0.000** | 13.819±0.139 | **0.529±0.001** |
| | 720 | 0.359±0.001 | 0.015±0.001 | 0.353±0.002 | **14.045±0.438** | 0.599±0.001 | **0.358±0.001** | **0.012±0.001** | 0.353±0.001 | 14.367±0.365 | **0.599±0.001** |
| | AVG | **0.262±0.061** | 0.018±0.002 | 0.289±0.043 | 13.438±0.871 | **0.508±0.060** | 0.262±0.060 | **0.014±0.002** | 0.289±0.042 | **13.419±0.930** | 0.508±0.059 |

Table 20: Full results of the proposed AliO method on the ECL, ETTh1, ETTh2, ETTm1, ETTm2, Solar, PEMS03, PEMS04, PEMS07, PEMS08, Exchange, Traffix, and Weather datasets and iTransformer [30]. The results are reported in terms of MSE, TAM, MAE, MAPE, and RMSE. The best results are highlighted in **bold**. We use official GitHub code (`https://github.com/thuml/iTransformer`) to train the model with the official configuration. The average improvement of AliO over the baseline is MSE: 6.59%, TAM: 28.04%, MAE: 5.41%, MAPE: 0.46%, and RMSE: 3.85%. The maximum improvement of AliO over the baseline is MSE: 88.52%, TAM: 60.37%, MAE: 73.75%, MAPE: 18.04%, and RMSE: 66.10%. Solar, PEMS03, PEMS04, PEMS07, and PEMS08 are in Tab. 21. Exchange, Traffic, and Weather are in Tab. 22.

| Models | | | iTransformer | | | | | | | | | |
| --- | --- | --- | --- | --- | --- | --- | --- | --- | --- | --- | --- | --- |
| Method | | | Baseline | | | | | AliO | | | | |
| Metric | | | MSE↓ | TAM↓ | MAE↓ | MAPE↓ | RMSE↓ | MSE↓ | TAM↓ | MAE↓ | MAPE↓ | RMSE↓ |
| ECL | 96 | | 0.148±0.000 | 0.046±0.000 | 0.240±0.000 | 2.525±0.030 | 0.385±0.000 | 0.147±0.000 | 0.034±0.000 | 0.238±0.000 | 2.512±0.018 | 0.383±0.000 |
| | 192 | | 0.165±0.001 | 0.042±0.000 | 0.256±0.001 | 2.732±0.033 | 0.406±0.001 | 0.160±0.000 | 0.031±0.000 | 0.250±0.000 | 2.671±0.023 | 0.400±0.000 |
| | 336 | | 0.180±0.001 | 0.049±0.001 | 0.273±0.001 | 2.736±0.005 | 0.424±0.001 | 0.175±0.001 | 0.036±0.001 | 0.267±0.000 | 2.802±0.025 | 0.418±0.001 |
| | 720 | | 0.211±0.000 | 0.063±0.000 | 0.301±0.000 | 3.068±0.024 | 0.459±0.000 | 0.207±0.001 | 0.025±0.001 | 0.294±0.001 | 3.073±0.032 | 0.455±0.001 |
| | AVG | | 0.176±0.021 | 0.050±0.007 | 0.267±0.020 | 2.765±0.174 | 0.418±0.024 | 0.172±0.020 | 0.031±0.004 | 0.262±0.019 | 2.764±0.184 | 0.414±0.024 |
| ETTh1 | 96 | | 0.386±0.001 | 0.083±0.001 | 0.404±0.001 | 10.066±0.052 | 0.621±0.001 | 0.380±0.000 | 0.057±0.000 | 0.397±0.000 | 9.673±0.026 | 0.616±0.000 |
| | 192 | | 0.443±0.001 | 0.083±0.001 | 0.437±0.001 | 9.867±0.093 | 0.666±0.001 | 0.431±0.000 | 0.064±0.001 | 0.428±0.000 | 9.624±0.060 | 0.657±0.000 |
| | 336 | | 0.487±0.003 | 0.089±0.002 | 0.459±0.002 | 10.520±0.334 | 0.698±0.002 | 0.471±0.001 | 0.038±0.000 | 0.445±0.001 | 10.034±0.076 | 0.686±0.001 |
| | 720 | | 0.506±0.007 | 0.098±0.001 | 0.492±0.003 | 11.807±0.649 | 0.711±0.005 | 0.471±0.008 | 0.048±0.002 | 0.469±0.004 | 11.258±0.051 | 0.687±0.006 |
| | AVG | | 0.455±0.041 | 0.088±0.005 | 0.448±0.029 | 10.565±0.676 | 0.674±0.031 | 0.438±0.034 | 0.052±0.009 | 0.435±0.023 | 10.147±0.591 | 0.661±0.026 |
| ETTh2 | 96 | | 0.300±0.001 | 0.080±0.002 | 0.350±0.001 | 1.469±0.010 | 0.548±0.001 | 0.300±0.000 | 0.045±0.001 | 0.348±0.000 | 1.422±0.004 | 0.548±0.000 |
| | 192 | | 0.378±0.000 | 0.076±0.001 | 0.398±0.001 | 1.596±0.014 | 0.615±0.000 | 0.375±0.001 | 0.058±0.002 | 0.396±0.000 | 1.560±0.006 | 0.612±0.000 |
| | 336 | | 0.421±0.002 | 0.073±0.002 | 0.431±0.001 | 1.775±0.008 | 0.649±0.001 | 0.413±0.001 | 0.055±0.003 | 0.427±0.001 | 1.712±0.002 | 0.643±0.001 |
| | 720 | | 0.429±0.001 | 0.069±0.002 | 0.447±0.000 | 2.018±0.012 | 0.655±0.001 | 0.419±0.002 | 0.050±0.002 | 0.441±0.001 | 1.981±0.024 | 0.647±0.001 |
| | AVG | | 0.382±0.046 | 0.074±0.003 | 0.406±0.033 | 1.714±0.184 | 0.617±0.038 | 0.377±0.042 | 0.052±0.005 | 0.403±0.032 | 1.669±0.186 | 0.613±0.036 |
| ETTm1 | 96 | | 0.343±0.003 | 0.067±0.001 | 0.377±0.001 | 2.354±0.012 | 0.586±0.002 | 0.330±0.001 | 0.037±0.000 | 0.361±0.000 | 2.142±0.005 | 0.575±0.001 |
| | 192 | | 0.380±0.002 | 0.060±0.001 | 0.394±0.001 | 2.383±0.013 | 0.617±0.001 | 0.375±0.001 | 0.031±0.000 | 0.383±0.001 | 2.203±0.011 | 0.612±0.001 |
| | 336 | | 0.418±0.000 | 0.058±0.001 | 0.418±0.000 | 2.443±0.006 | 0.647±0.000 | 0.408±0.001 | 0.037±0.000 | 0.406±0.000 | 2.302±0.005 | 0.639±0.001 |
| | 720 | | 0.488±0.001 | 0.056±0.001 | 0.457±0.001 | 2.670±0.012 | 0.699±0.001 | 0.472±0.001 | 0.026±0.000 | 0.441±0.001 | 2.451±0.005 | 0.687±0.001 |
| | AVG | | 0.407±0.048 | 0.060±0.004 | 0.412±0.027 | 2.462±0.111 | 0.637±0.037 | 0.396±0.046 | 0.033±0.004 | 0.398±0.026 | 2.275±0.104 | 0.628±0.037 |
| ETTm2 | 96 | | 0.185±0.001 | 0.050±0.001 | 0.270±0.001 | 1.169±0.009 | 0.430±0.001 | 0.184±0.000 | 0.038±0.000 | 0.267±0.000 | 1.160±0.002 | 0.428±0.000 |
| | 192 | | 0.254±0.000 | 0.050±0.001 | 0.314±0.001 | 1.302±0.018 | 0.504±0.000 | 0.250±0.001 | 0.025±0.000 | 0.310±0.001 | 1.287±0.005 | 0.500±0.001 |
| | 336 | | 0.315±0.004 | 0.049±0.006 | 0.352±0.003 | 1.411±0.005 | 0.562±0.004 | 0.312±0.001 | 0.022±0.000 | 0.348±0.001 | 1.399±0.003 | 0.559±0.001 |
| | 720 | | 0.413±0.001 | 0.044±0.003 | 0.407±0.001 | 1.615±0.006 | 0.643±0.001 | 0.411±0.001 | 0.019±0.000 | 0.404±0.001 | 1.594±0.009 | 0.641±0.001 |
| | AVG | | 0.292±0.075 | 0.048±0.002 | 0.336±0.045 | 1.374±0.146 | 0.534±0.070 | 0.289±0.075 | 0.026±0.007 | 0.332±0.045 | 1.360±0.142 | 0.532±0.070 |

Table 21: Full results of the proposed AliO method on the ECL, ETTh1, ETTm1, ETTh2, ETTm2, Solar, PEMS03, PEMS04, PEMS07, PEMS08, Exchange, Traffix, and Weather datasets and iTransformer [30]. The results are reported in terms of MSE, TAM, MAE, MAPE, and RMSE. The best results are highlighted in **bold**. We use official GitHub code (https://github.com/thuml/iTransformer) to train the model with the official configuration. The average improvement of AliO over the baseline is MSE: 6.59%, TAM: 5.41%, MAE: 0.46%, and RMSE: 3.85%. The maximum improvement of AliO over the baseline is MSE: 88.52%, TAM: 60.37%, MAE: 73.75%, MAPE: 18.04%, and RMSE: 66.10%. ECL, ETTh1, ETTh2, ETTm1, and ETTm2 are in Tab. 20. Exchange, Traffix, and Weather are in Tab. 22

| Models | | | iTransformer | | | | | | | | | |
| --- | --- | --- | --- | --- | --- | --- | --- | --- | --- | --- | --- | --- |
| Method | | | Baseline | | | | | AliO | | | | |
| Metric | | | MSE↓ | TAM↓ | MAE↓ | MAPE↓ | RMSE↓ | MSE↓ | TAM↓ | MAE↓ | MAPE↓ | RMSE↓ |
| Solar | 96 | | 0.206±0.002 | 0.038±0.002 | 0.237±0.002 | 1.846±0.033 | 0.454±0.002 | 0.192±0.002 | 0.021±0.001 | 0.217±0.002 | 1.843±0.035 | 0.438±0.002 |
| | 192 | | 0.237±0.001 | 0.040±0.000 | 0.263±0.001 | 1.976±0.014 | 0.487±0.001 | 0.225±0.001 | 0.020±0.000 | 0.241±0.001 | 1.980±0.008 | 0.474±0.001 |
| | 336 | | 0.250±0.001 | 0.038±0.001 | 0.275±0.001 | 2.011±0.011 | 0.500±0.001 | 0.242±0.001 | 0.019±0.000 | 0.257±0.000 | 2.017±0.011 | 0.492±0.001 |
| | 720 | | 0.251±0.000 | 0.035±0.000 | 0.275±0.000 | 2.066±0.025 | 0.501±0.000 | 0.247±0.001 | 0.016±0.000 | 0.261±0.000 | 2.092±0.012 | 0.497±0.001 |
| | AVG | | 0.236±0.016 | 0.038±0.002 | 0.263±0.014 | 1.975±0.072 | 0.485±0.017 | 0.227±0.019 | 0.019±0.002 | 0.244±0.015 | 1.983±0.081 | 0.475±0.021 |
| PEMS03 | 12 | | 0.069±0.000 | 0.048±0.000 | 0.175±0.001 | 1.417±0.006 | 0.263±0.001 | 0.069±0.000 | 0.028±0.000 | 0.175±0.000 | 1.424±0.006 | 0.263±0.000 |
| | 24 | | 0.098±0.001 | 0.052±0.000 | 0.209±0.001 | 1.671±0.011 | 0.313±0.001 | 0.098±0.000 | 0.041±0.000 | 0.209±0.000 | 1.663±0.006 | 0.313±0.001 |
| | 48 | | 0.163±0.001 | 0.061±0.001 | 0.274±0.001 | 2.022±0.009 | 0.404±0.001 | 0.162±0.001 | 0.047±0.001 | 0.273±0.001 | 2.000±0.015 | 0.402±0.001 |
| | 96 | | 0.918±0.310 | 0.117±0.026 | 0.720±0.127 | 3.747±0.531 | 0.944±0.162 | 0.576±0.044 | 0.080±0.009 | 0.575±0.028 | 3.383±0.266 | 0.759±0.029 |
| | AVG | | 0.312±0.314 | 0.069±0.025 | 0.345±0.196 | 2.214±0.814 | 0.481±0.243 | 0.226±0.183 | 0.049±0.017 | 0.308±0.141 | 2.117±0.679 | 0.434±0.173 |
| PEMS04 | 12 | | 0.081±0.000 | 0.042±0.000 | 0.188±0.000 | 1.283±0.007 | 0.284±0.000 | 0.082±0.000 | 0.033±0.000 | 0.190±0.001 | 1.294±0.013 | 0.286±0.000 |
| | 24 | | 0.100±0.000 | 0.041±0.000 | 0.212±0.000 | 1.512±0.019 | 0.316±0.000 | 0.101±0.000 | 0.031±0.000 | 0.213±0.000 | 1.521±0.009 | 0.317±0.001 |
| | 48 | | 0.134±0.002 | 0.041±0.000 | 0.248±0.002 | 1.782±0.029 | 0.366±0.002 | 0.131±0.001 | 0.029±0.000 | 0.245±0.001 | 1.771±0.017 | 0.362±0.002 |
| | 96 | | 0.169±0.001 | 0.041±0.000 | 0.280±0.000 | 2.090±0.015 | 0.411±0.001 | 0.166±0.002 | 0.028±0.000 | 0.277±0.002 | 2.081±0.008 | 0.407±0.003 |
| | AVG | | 0.121±0.030 | 0.041±0.001 | 0.232±0.031 | 1.667±0.270 | 0.344±0.043 | 0.120±0.028 | 0.030±0.002 | 0.231±0.030 | 1.667±0.262 | 0.343±0.041 |
| PEMS07 | 12 | | 0.067±0.000 | 0.044±0.000 | 0.164±0.001 | 1.670±0.003 | 0.258±0.000 | 0.070±0.001 | 0.036±0.001 | 0.168±0.001 | 1.706±0.017 | 0.264±0.001 |
| | 24 | | 0.087±0.000 | 0.042±0.000 | 0.190±0.001 | 1.898±0.019 | 0.295±0.001 | 0.090±0.001 | 0.035±0.000 | 0.193±0.001 | 1.943±0.028 | 0.300±0.001 |
| | 48 | | 0.995±0.054 | 0.020±0.019 | 0.832±0.032 | 1.540±0.702 | 0.997±0.027 | 0.114±0.001 | 0.031±0.001 | 0.218±0.002 | 2.158±0.017 | 0.338±0.002 |
| | 96 | | 1.172±0.189 | 0.021±0.023 | 0.895±0.050 | 2.412±1.881 | 1.079±0.085 | 0.402±0.161 | 0.052±0.017 | 0.435±0.089 | 4.015±0.667 | 0.623±0.121 |
| | AVG | | 0.580±0.454 | 0.031±0.010 | 0.520±0.308 | 1.880±0.298 | 0.657±0.342 | 0.169±0.121 | 0.038±0.007 | 0.254±0.095 | 2.455±0.818 | 0.381±0.127 |
| PEMS08 | 12 | | 0.088±0.001 | 0.053±0.000 | 0.193±0.001 | 1.637±0.004 | 0.297±0.001 | 0.089±0.000 | 0.046±0.000 | 0.193±0.000 | 1.643±0.003 | 0.298±0.001 |
| | 24 | | 0.138±0.000 | 0.059±0.000 | 0.243±0.001 | 2.074±0.005 | 0.371±0.001 | 0.138±0.000 | 0.051±0.000 | 0.242±0.000 | 2.076±0.011 | 0.371±0.001 |
| | 48 | | 0.247±0.010 | 0.052±0.001 | 0.287±0.006 | 2.242±0.070 | 0.497±0.010 | 0.232±0.003 | 0.037±0.001 | 0.271±0.002 | 2.085±0.038 | 0.481±0.003 |
| | 96 | | 0.452±0.020 | 0.065±0.005 | 0.431±0.013 | 2.921±0.217 | 0.672±0.015 | 0.282±0.003 | 0.030±0.001 | 0.312±0.002 | 2.394±0.052 | 0.531±0.003 |
| | AVG | | 0.231±0.125 | 0.057±0.005 | 0.288±0.079 | 2.219±0.413 | 0.459±0.127 | 0.185±0.068 | 0.041±0.007 | 0.255±0.039 | 2.050±0.239 | 0.420±0.082 |

Table 22: Full results of the proposed AliO method on the ECL, ETTh1, ETTh2, ETTm1, ETTm2, Solar, PEMS03, PEMS04, PEMS07, PEMS08, Exchange, Traffix, and Weather datasets and iTransformer [30]. The results are reported in terms of MSE, TAM, MAE, MAPE, and RMSE. The best results are highlighted in **bold**. We use official GitHub code (https://github.com/thuml/iTransformer) to train the model with the official configuration. The average improvement of AliO over the baseline is MSE: 6.59%, TAM: 28.04%, MAE: 5.41%, MAPE: 0.46%, and RMSE: 3.85%. The maximum improvement of AliO over the baseline is MSE: 88.52%, TAM: 60.37%, MAE: 73.75%, MAPE: 18.04%, and RMSE: 66.10%. ECL, ETTh1, ETTh2, ETTm1, and ETTm2 are in Tab. 20. Solar, PEMS03, PEMS04, PEMS07, and PEMS08 are in Tab. 21.

| Models | | iTransformer | | | | | | | | | |
|---|---|---|---|---|---|---|---|---|---|---|---|
| Method | | Baseline | | | | | AliO | | | | |
| Metric | | MSE↓ | TAM↓ | MAE↓ | MAPE↓ | RMSE↓ | MSE↓ | TAM↓ | MAE↓ | MAPE↓ | RMSE↓ |
| Exchange | 96 | 0.086±0.001 | 0.035±0.001 | 0.206±0.001 | 1.287±0.007 | 0.294±0.001 | **0.086±0.000** | **0.030±0.000** | **0.205±0.001** | **1.280±0.006** | **0.293±0.001** |
| | 192 | 0.179±0.001 | 0.037±0.000 | 0.302±0.001 | 1.926±0.011 | 0.423±0.001 | **0.178±0.001** | **0.031±0.000** | **0.300±0.001** | **1.895±0.003** | **0.421±0.001** |
| | 336 | 0.335±0.003 | 0.042±0.000 | 0.420±0.002 | 2.952±0.014 | 0.579±0.003 | **0.331±0.002** | **0.032±0.001** | **0.417±0.001** | **2.914±0.013** | **0.576±0.001** |
| | 720 | 0.865±0.006 | 0.038±0.002 | 0.702±0.003 | 6.170±0.019 | 0.930±0.003 | **0.857±0.006** | **0.031±0.002** | **0.699±0.002** | **6.150±0.021** | **0.926±0.003** |
| | AVG | 0.366±0.270 | 0.038±0.002 | 0.408±0.166 | 3.084±1.680 | 0.557±0.213 | **0.363±0.267** | **0.031±0.001** | **0.405±0.166** | **3.060±1.679** | **0.554±0.212** |
| Traffic | 96 | 0.394±0.001 | 0.066±0.001 | 0.269±0.001 | 2.897±0.005 | 0.628±0.001 | **0.393±0.001** | **0.049±0.000** | **0.264±0.000** | **2.723±0.010** | **0.627±0.001** |
| | 192 | 0.413±0.001 | 0.061±0.000 | 0.277±0.000 | 2.948±0.011 | 0.643±0.001 | **0.412±0.000** | **0.043±0.000** | **0.271±0.000** | **2.757±0.011** | **0.642±0.000** |
| | 336 | **0.425±0.001** | 0.059±0.000 | 0.283±0.001 | 2.947±0.021 | **0.652±0.001** | 0.428±0.000 | **0.041±0.000** | **0.278±0.000** | **2.788±0.009** | 0.654±0.000 |
| | 720 | **0.457±0.002** | 0.060±0.000 | 0.300±0.001 | 3.108±0.034 | **0.676±0.001** | 0.460±0.000 | **0.041±0.000** | **0.296±0.000** | **2.889±0.008** | 0.678±0.000 |
| | AVG | **0.422±0.020** | 0.061±0.002 | 0.282±0.010 | 2.975±0.071 | **0.649±0.016** | 0.423±0.022 | **0.044±0.003** | **0.277±0.011** | **2.789±0.055** | 0.650±0.017 |
| Weather | 96 | 0.176±0.002 | 0.024±0.001 | 0.216±0.002 | 16.182±0.603 | 0.420±0.002 | **0.175±0.000** | **0.021±0.000** | **0.215±0.000** | **14.334±0.300** | **0.418±0.000** |
| | 192 | 0.225±0.001 | 0.024±0.002 | 0.257±0.001 | 16.029±0.036 | 0.474±0.001 | **0.223±0.000** | **0.019±0.000** | **0.257±0.000** | **15.938±0.131** | **0.473±0.000** |
| | 336 | 0.282±0.001 | 0.024±0.000 | 0.299±0.001 | 15.771±0.132 | 0.531±0.001 | **0.281±0.001** | **0.016±0.000** | **0.298±0.000** | **15.376±0.069** | **0.530±0.001** |
| | 720 | 0.359±0.001 | 0.023±0.001 | 0.350±0.001 | **15.409±0.151** | 0.599±0.001 | **0.357±0.000** | **0.013±0.000** | **0.347±0.000** | 15.819±0.052 | **0.597±0.000** |
| | AVG | 0.260±0.061 | 0.024±0.000 | 0.281±0.045 | 15.848±0.262 | 0.506±0.059 | **0.259±0.061** | **0.017±0.003** | **0.279±0.044** | **15.367±0.565** | **0.505±0.060** |

Table 23: Full results of the proposed AliO method on the ECL, ETTh1, ETTh2, ETTm1, ETTm2, ILI, Traffix, and Weather datasets and GPT4TS [43]. The results are reported in terms of MSE, TAM, MAE, MAPE, and RMSE. The best results are highlighted in **bold**. We use official GitHub code (https://github.com/DAMO-DI-ML/NeurIPS2023-One-Fits-All) to train the model with the official configuration. The average improvement of AliO over the baseline is MSE: 2.27%, TAM: 32.13%, MAE: 2.43%, MAPE: 1.56%, and RMSE: 1.14%. The maximum improvement of AliO over the baseline is MSE: 11.54%, TAM: 62.06%, MAE: 10.98%, MAPE: 8.16%, and RMSE: 5.93%. ILI, Traffix, and Weather are in Tab. 24.

| Models | | GPT4TS | | | | | | | | | |
| --- | --- | --- | --- | --- | --- | --- | --- | --- | --- | --- | --- |
| Method | | Baseline | | | | | AliO | | | | |
| Metric | | MSE↓ | TAM↓ | MAE↓ | MAPE↓ | RMSE↓ | MSE↓ | TAM↓ | MAE↓ | MAPE↓ | RMSE↓ |
| ETTh1 | 96 | 0.380±0.003 | 0.060±0.001 | 0.401±0.001 | 9.556±0.065 | 0.616±0.002 | 0.374±0.003 | 0.034±0.001 | 0.393±0.001 | 9.675±0.079 | 0.612±0.002 |
| | 192 | 0.418±0.001 | 0.051±0.001 | 0.419±0.003 | 9.502±0.101 | 0.646±0.001 | 0.412±0.002 | 0.037±0.001 | 0.416±0.001 | 9.532±0.276 | 0.642±0.001 |
| | 336 | 0.441±0.007 | 0.054±0.004 | 0.435±0.004 | 9.507±0.157 | 0.664±0.006 | 0.431±0.004 | 0.031±0.000 | 0.427±0.003 | 9.519±0.180 | 0.656±0.003 |
| | 720 | 0.458±0.006 | 0.058±0.005 | 0.465±0.003 | 10.152±0.159 | 0.677±0.004 | 0.452±0.009 | 0.053±0.002 | 0.456±0.004 | 10.804±0.153 | 0.672±0.007 |
| | AVG | 0.424±0.026 | 0.056±0.003 | 0.430±0.021 | 9.679±0.245 | 0.651±0.020 | 0.417±0.026 | 0.039±0.008 | 0.423±0.020 | 9.883±0.479 | 0.646±0.020 |
| ETTh2 | 96 | 0.292±0.003 | 0.053±0.003 | 0.353±0.001 | 1.377±0.031 | 0.540±0.002 | 0.288±0.003 | 0.041±0.001 | 0.348±0.004 | 1.352±0.023 | 0.537±0.003 |
| | 192 | 0.368±0.008 | 0.054±0.006 | 0.400±0.004 | 1.484±0.017 | 0.606±0.006 | 0.351±0.001 | 0.033±0.002 | 0.388±0.001 | 1.470±0.024 | 0.592±0.001 |
| | 336 | 0.376±0.003 | 0.041±0.002 | 0.414±0.002 | 1.674±0.030 | 0.613±0.002 | 0.373±0.002 | 0.029±0.001 | 0.409±0.002 | 1.622±0.025 | 0.611±0.001 |
| | 720 | 0.414±0.004 | 0.042±0.003 | 0.449±0.005 | 2.051±0.056 | 0.644±0.003 | 0.404±0.002 | 0.027±0.002 | 0.441±0.002 | 1.981±0.017 | 0.635±0.002 |
| | AVG | 0.363±0.040 | 0.048±0.005 | 0.404±0.031 | 1.646±0.230 | 0.601±0.034 | 0.354±0.038 | 0.033±0.005 | 0.396±0.030 | 1.606±0.212 | 0.594±0.032 |
| ETTm1 | 96 | 0.292±0.002 | 0.047±0.001 | 0.347±0.002 | 2.184±0.033 | 0.540±0.002 | 0.286±0.001 | 0.036±0.000 | 0.340±0.001 | 2.136±0.011 | 0.535±0.001 |
| | 192 | 0.330±0.001 | 0.043±0.002 | 0.369±0.001 | 2.241±0.011 | 0.575±0.001 | 0.325±0.001 | 0.029±0.000 | 0.362±0.001 | 2.220±0.004 | 0.570±0.001 |
| | 336 | 0.365±0.002 | 0.042±0.002 | 0.392±0.003 | 2.350±0.054 | 0.605±0.002 | 0.361±0.001 | 0.027±0.001 | 0.385±0.000 | 2.305±0.008 | 0.601±0.000 |
| | 720 | 0.418±0.001 | 0.043±0.001 | 0.423±0.001 | 2.506±0.031 | 0.646±0.001 | 0.414±0.001 | 0.033±0.001 | 0.418±0.001 | 2.469±0.009 | 0.643±0.001 |
| | AVG | 0.351±0.041 | 0.044±0.002 | 0.383±0.025 | 2.320±0.110 | 0.592±0.035 | 0.346±0.042 | 0.031±0.003 | 0.376±0.026 | 2.283±0.110 | 0.587±0.036 |
| ETTm2 | 96 | 0.170±0.002 | 0.039±0.002 | 0.259±0.002 | 1.096±0.010 | 0.412±0.002 | 0.168±0.000 | 0.033±0.001 | 0.257±0.000 | 1.079±0.003 | 0.410±0.001 |
| | 192 | 0.233±0.004 | 0.042±0.003 | 0.305±0.003 | 1.224±0.020 | 0.483±0.004 | 0.225±0.000 | 0.025±0.001 | 0.297±0.001 | 1.199±0.012 | 0.474±0.001 |
| | 336 | 0.297±0.009 | 0.048±0.007 | 0.348±0.005 | 1.363±0.011 | 0.545±0.008 | 0.284±0.003 | 0.026±0.001 | 0.337±0.002 | 1.323±0.016 | 0.533±0.002 |
| | 720 | 0.382±0.006 | 0.041±0.003 | 0.401±0.005 | 1.532±0.034 | 0.618±0.005 | 0.382±0.001 | 0.028±0.000 | 0.400±0.000 | 1.531±0.015 | 0.618±0.001 |
| | AVG | 0.270±0.070 | 0.044±0.003 | 0.328±0.047 | 1.304±0.145 | 0.514±0.068 | 0.265±0.071 | 0.028±0.003 | 0.323±0.047 | 1.283±0.150 | 0.509±0.068 |
| ECL | 96 | 0.138±0.000 | 0.034±0.000 | 0.237±0.000 | 2.442±0.003 | 0.372±0.000 | 0.138±0.000 | 0.025±0.000 | 0.237±0.001 | 2.412±0.003 | 0.372±0.000 |
| | 192 | 0.155±0.000 | 0.034±0.000 | 0.252±0.001 | 2.679±0.026 | 0.393±0.000 | 0.154±0.000 | 0.025±0.000 | 0.251±0.000 | 2.652±0.033 | 0.392±0.000 |
| | 336 | 0.169±0.000 | 0.036±0.000 | 0.266±0.001 | 2.707±0.030 | 0.411±0.001 | 0.169±0.001 | 0.021±0.000 | 0.266±0.001 | 2.665±0.023 | 0.411±0.001 |
| | 720 | 0.206±0.000 | 0.042±0.001 | 0.297±0.001 | 2.915±0.033 | 0.454±0.000 | 0.206±0.000 | 0.028±0.003 | 0.296±0.000 | 2.818±0.063 | 0.454±0.000 |
| | AVG | 0.167±0.023 | 0.036±0.003 | 0.263±0.020 | 2.686±0.150 | 0.407±0.027 | 0.167±0.022 | 0.025±0.002 | 0.262±0.020 | 2.637±0.130 | 0.407±0.027 |

Table 24: Full results of the proposed AliO method on the ECL, ETTh1, ETTh2, ETTm1, ETTm2, ILI, Traffix, and Weather datasets and GPT4TS [43]. The results are reported in terms of MSE, TAM, MAE, MAPE, and RMSE. The best results are highlighted in **bold**. We use official GitHub code (https://github.com/DAMO-DI-ML/NeurIPS2023-One-Fits-All) to train the model with the official configuration. The average improvement of AliO over the baseline is MSE: 2.27%, TAM: 32.13%, MAE: 2.43%, MAPE: 1.56%, and RMSE: 1.14%. The maximum improvement of AliO over the baseline is MSE: 11.54%, TAM: 62.06%, MAE: 10.98%, MAPE: 8.16%, and RMSE: 5.93%. ECL, ETTh1, ETTh2, ETTm1, and ETTm2 are in Tab. 23.

| Models | | GPT4TS | | | | | | | | | |
|---|---|---|---|---|---|---|---|---|---|---|---|
| Method | | Baseline | | | | | AliO | | | | |
| Metric | | MSE↓ | TAM↓ | MAE↓ | MAPE↓ | RMSE↓ | MSE↓ | TAM↓ | MAE↓ | MAPE↓ | RMSE↓ |
| ILI | 24 | 1.965±0.046 | 0.137±0.010 | 0.862±0.009 | 3.994±0.129 | 1.402±0.016 | **1.897±0.036** | **0.115±0.007** | **0.839±0.007** | **3.967±0.117** | **1.377±0.013** |
| | 36 | 1.914±0.090 | 0.104±0.026 | 0.913±0.037 | 3.408±0.334 | 1.383±0.033 | **1.693±0.044** | **0.090±0.007** | **0.812±0.020** | **3.182±0.090** | **1.301±0.017** |
| | 48 | 1.851±0.096 | 0.112±0.014 | 0.915±0.032 | **3.073±0.103** | 1.360±0.036 | **1.722±0.041** | **0.091±0.008** | **0.852±0.006** | 3.268±0.148 | **1.312±0.015** |
| | 60 | 1.861±0.056 | 0.114±0.011 | 0.927±0.031 | **3.225±0.188** | 1.364±0.020 | **1.707±0.032** | **0.097±0.005** | **0.880±0.011** | 3.376±0.155 | **1.306±0.012** |
| | AVG | 1.898±0.041 | 0.117±0.011 | 0.904±0.023 | **3.425±0.313** | 1.377±0.015 | **1.755±0.074** | **0.098±0.009** | **0.846±0.022** | 3.448±0.275 | **1.324±0.028** |
| Traffic | 96 | 0.386±0.001 | 0.056±0.000 | 0.279±0.000 | 2.996±0.001 | 0.621±0.001 | **0.377±0.002** | **0.039±0.000** | **0.267±0.002** | **2.819±0.007** | **0.614±0.002** |
| | 192 | 0.404±0.001 | 0.058±0.002 | 0.286±0.003 | 3.081±0.043 | 0.636±0.001 | **0.399±0.001** | **0.032±0.001** | **0.276±0.001** | **2.854±0.002** | **0.632±0.001** |
| | 336 | 0.411±0.000 | 0.061±0.001 | 0.290±0.000 | 3.106±0.016 | 0.641±0.000 | **0.406±0.002** | **0.032±0.000** | **0.282±0.002** | **2.888±0.011** | **0.637±0.002** |
| | 720 | 0.444±0.000 | 0.063±0.001 | 0.304±0.001 | 3.212±0.029 | 0.667±0.000 | **0.438±0.000** | **0.024±0.000** | **0.295±0.000** | **2.950±0.007** | **0.662±0.000** |
| | AVG | 0.411±0.019 | 0.059±0.002 | 0.289±0.008 | 3.099±0.069 | 0.641±0.015 | **0.405±0.020** | **0.032±0.005** | **0.280±0.009** | **2.878±0.043** | **0.636±0.015** |
| Weather | 96 | 0.148±0.002 | 0.026±0.001 | 0.199±0.003 | 12.027±0.525 | 0.385±0.002 | **0.147±0.001** | **0.018±0.000** | **0.194±0.002** | **11.946±0.167** | **0.384±0.002** |
| | 192 | 0.195±0.000 | 0.024±0.001 | 0.244±0.000 | 14.653±0.401 | 0.442±0.000 | **0.192±0.001** | **0.017±0.000** | **0.237±0.001** | **14.616±0.251** | **0.439±0.001** |
| | 336 | 0.246±0.002 | 0.024±0.000 | 0.284±0.001 | 15.431±0.356 | 0.496±0.002 | **0.244±0.001** | **0.013±0.000** | **0.276±0.001** | **14.887±0.367** | **0.494±0.001** |
| | 720 | 0.319±0.001 | 0.022±0.001 | 0.336±0.002 | **14.770±0.506** | 0.565±0.001 | **0.316±0.001** | **0.011±0.000** | **0.328±0.001** | 14.877±0.092 | **0.562±0.000** |
| | AVG | 0.227±0.057 | 0.024±0.001 | 0.266±0.045 | 14.220±1.163 | 0.472±0.060 | **0.225±0.056** | **0.015±0.002** | **0.259±0.044** | **14.081±1.107** | **0.469±0.059** |

Table 25: Full results of the proposed AliO method on the ECL, ETTh1, ETTh2, ETTm1, ETTm2, Solar, traffic, and Weather datasets and CycleNet [27]. The results are reported in terms of MSE, TAM, MAE, MAPE, and RMSE. The best results are highlighted in **bold**. We use official GitHub code (https://github.com/ACAT-SCUT/CycleNet) to train the model with the official configuration. The average improvement of AliO over the baseline is MSE: 0.52%, TAM: 37.91%, MAE: 0.91%, MAPE: 1.84%, and RMSE: 0.26%. The maximum improvement of AliO over the baseline is MSE: 2.73%, TAM: 74.35%, MAE: 6.79%, MAPE: 13.27%, and RMSE: 1.38%. Solar, Traffix, and Weather are in Tab. 26.

| Models | | | CycleNet | | | | | | | | | |
| Method | | | Baseline | | | | | AliO | | | | |
| Metric | | MSE↓ | TAM↓ | MAE↓ | MAPE↓ | RMSE↓ | MSE↓ | TAM↓ | MAE↓ | MAPE↓ | RMSE↓ |
| ECL | 96 | **0.141±0.000** | **0.019±0.000** | **0.234±0.000** | 2.305±0.002 | **0.376±0.000** | 0.142±0.000 | 0.019±0.000 | 0.234±0.000 | **2.304±0.004** | 0.376±0.000 |
| | 192 | 0.156±0.000 | 0.017±0.002 | 0.247±0.000 | 2.450±0.007 | 0.395±0.000 | **0.156±0.000** | **0.015±0.000** | **0.247±0.000** | **2.444±0.003** | **0.394±0.000** |
| | 336 | 0.173±0.000 | 0.014±0.000 | 0.265±0.000 | 2.439±0.003 | 0.415±0.000 | **0.173±0.000** | **0.010±0.000** | **0.265±0.000** | **2.439±0.001** | **0.415±0.000** |
| | 720 | 0.211±0.000 | 0.013±0.000 | 0.297±0.000 | **2.583±0.001** | 0.459±0.000 | **0.211±0.000** | **0.010±0.000** | **0.297±0.000** | 2.584±0.002 | **0.459±0.000** |
| | AVG | 0.170±0.023 | 0.016±0.002 | 0.261±0.021 | 2.444±0.088 | 0.411±0.028 | **0.170±0.023** | **0.014±0.003** | **0.261±0.021** | **2.443±0.089** | **0.411±0.028** |
| ETTh1 | 96 | 0.380±0.002 | 0.054±0.011 | 0.393±0.002 | 9.380±0.023 | 0.616±0.002 | **0.376±0.000** | **0.016±0.001** | **0.387±0.000** | **9.142±0.027** | **0.613±0.000** |
| | 192 | 0.425±0.001 | 0.043±0.005 | 0.418±0.001 | 9.471±0.032 | 0.652±0.001 | **0.422±0.000** | **0.013±0.001** | **0.414±0.000** | **9.185±0.033** | **0.650±0.000** |
| | 336 | 0.462±0.001 | 0.041±0.005 | 0.438±0.001 | 9.443±0.011 | 0.680±0.001 | **0.459±0.001** | **0.012±0.001** | **0.434±0.000** | **9.210±0.029** | **0.678±0.000** |
| | 720 | 0.461±0.000 | 0.038±0.002 | 0.460±0.000 | 9.559±0.015 | 0.679±0.000 | **0.458±0.001** | **0.011±0.000** | **0.454±0.000** | **9.408±0.029** | **0.676±0.000** |
| | AVG | 0.432±0.030 | 0.044±0.005 | 0.427±0.022 | 9.463±0.058 | 0.657±0.023 | **0.429±0.030** | **0.013±0.002** | **0.422±0.022** | **9.236±0.091** | **0.654±0.024** |
| ETTh2 | 96 | 0.285±0.001 | 0.047±0.002 | 0.335±0.001 | 1.343±0.005 | 0.534±0.001 | **0.283±0.000** | **0.032±0.000** | **0.334±0.000** | **1.324±0.001** | **0.532±0.000** |
| | 192 | 0.373±0.002 | 0.042±0.002 | 0.392±0.002 | 1.512±0.011 | 0.611±0.001 | **0.371±0.000** | **0.029±0.001** | **0.390±0.000** | **1.504±0.004** | **0.609±0.000** |
| | 336 | 0.424±0.004 | 0.048±0.013 | 0.435±0.002 | 1.770±0.012 | 0.651±0.003 | **0.418±0.000** | **0.015±0.000** | **0.431±0.000** | **1.735±0.005** | **0.647±0.000** |
| | 720 | 0.456±0.003 | 0.053±0.013 | 0.459±0.002 | 2.066±0.009 | 0.675±0.003 | **0.449±0.000** | **0.023±0.005** | **0.456±0.000** | **2.049±0.001** | **0.670±0.000** |
| | AVG | 0.385±0.058 | 0.047±0.003 | 0.405±0.042 | 1.672±0.244 | 0.618±0.048 | **0.381±0.056** | **0.025±0.006** | **0.402±0.041** | **1.653±0.243** | **0.615±0.047** |
| ETTm1 | 96 | 0.326±0.001 | 0.039±0.003 | 0.364±0.001 | 2.208±0.001 | 0.571±0.001 | **0.320±0.000** | **0.017±0.000** | **0.355±0.000** | **2.112±0.002** | **0.566±0.000** |
| | 192 | 0.366±0.000 | 0.032±0.003 | 0.382±0.000 | 2.263±0.005 | 0.605±0.000 | **0.364±0.000** | **0.013±0.000** | **0.376±0.000** | **2.204±0.000** | **0.603±0.000** |
| | 336 | 0.396±0.001 | 0.031±0.003 | 0.402±0.001 | 2.316±0.002 | 0.629±0.000 | **0.394±0.000** | **0.012±0.000** | **0.396±0.000** | **2.261±0.000** | **0.627±0.000** |
| | 720 | 0.457±0.000 | 0.026±0.001 | 0.434±0.000 | 2.476±0.003 | 0.676±0.000 | **0.455±0.000** | **0.016±0.001** | **0.431±0.000** | **2.449±0.007** | **0.675±0.000** |
| | AVG | 0.386±0.043 | 0.032±0.004 | 0.395±0.023 | 2.316±0.089 | 0.620±0.034 | **0.383±0.044** | **0.014±0.002** | **0.390±0.025** | **2.256±0.110** | **0.618±0.035** |
| ETTm2 | 96 | 0.167±0.001 | 0.033±0.002 | 0.248±0.001 | 1.041±0.004 | 0.408±0.001 | **0.166±0.000** | **0.022±0.000** | **0.247±0.000** | **1.032±0.000** | **0.407±0.000** |
| | 192 | 0.233±0.001 | 0.029±0.002 | 0.291±0.000 | 1.158±0.002 | 0.482±0.001 | **0.232±0.000** | **0.020±0.000** | **0.290±0.000** | **1.151±0.000** | **0.481±0.000** |
| | 336 | 0.294±0.001 | 0.025±0.001 | 0.330±0.000 | 1.265±0.005 | 0.542±0.001 | **0.293±0.000** | **0.018±0.000** | **0.330±0.000** | **1.264±0.000** | **0.541±0.000** |
| | 720 | 0.394±0.000 | 0.022±0.000 | 0.389±0.000 | 1.470±0.004 | 0.628±0.000 | **0.394±0.000** | **0.017±0.000** | **0.389±0.000** | **1.469±0.000** | **0.628±0.000** |
| | AVG | 0.272±0.075 | 0.027±0.004 | 0.315±0.046 | 1.233±0.141 | 0.515±0.072 | **0.271±0.075** | **0.019±0.002** | **0.314±0.047** | **1.229±0.144** | **0.514±0.072** |

Table 26: Full results of the proposed AliO method on the ECL, ETTh1, ETTh2, ETTm1, ETTm2, Solar, traffic, and Weather datasets and CycleNet [27]. The results are reported in terms of MSE, TAM, MAE, MAPE, and RMSE. The best results are highlighted in **bold**. We use official GitHub code (https://github.com/ACAT-SCUT/CycleNet) to train the model with the official configuration. The average improvement of AliO over the baseline is MSE: 0.52%, TAM: 37.91%, MAE: 0.91%, MAPE: 1.84%, and RMSE: 0.26%. The maximum improvement of AliO over the baseline is MSE: 2.73%, TAM: 74.35%, MAE: 6.79%, MAPE: 13.27%, and RMSE: 1.38%. ECL, ETTh1, ETTh2, ETTm1, and ETTm2 are in Tab. 25.

| Models | | | CycleNet | | | | | | | | | |
| Method | | | Baseline | | | | | AliO | | | | |
| Metric | | MSE↓ | TAM↓ | MAE↓ | MAPE↓ | RMSE↓ | MSE↓ | TAM↓ | MAE↓ | MAPE↓ | RMSE↓ |
|---|---|---|---|---|---|---|---|---|---|---|---|
| Solar | 96 | 0.250±0.001 | 0.030±0.002 | 0.278±0.001 | 1.984±0.005 | 0.500±0.001 | 0.250±0.000 | 0.026±0.000 | 0.277±0.000 | 1.982±0.000 | 0.500±0.000 |
| | 192 | 0.289±0.000 | 0.027±0.000 | 0.298±0.001 | 2.101±0.005 | 0.538±0.000 | 0.290±0.000 | 0.026±0.000 | 0.299±0.000 | 2.110±0.001 | 0.539±0.000 |
| | 336 | 0.338±0.000 | 0.024±0.000 | 0.322±0.000 | 2.335±0.001 | 0.582±0.000 | 0.338±0.000 | 0.026±0.002 | 0.323±0.001 | 2.328±0.003 | 0.581±0.000 |
| | 720 | 0.351±0.000 | 0.021±0.001 | 0.327±0.000 | 2.478±0.005 | 0.593±0.000 | 0.352±0.000 | 0.020±0.000 | 0.327±0.000 | 2.480±0.001 | 0.593±0.000 |
| | AVG | 0.307±0.036 | 0.026±0.003 | 0.306±0.018 | 2.224±0.173 | 0.553±0.033 | 0.308±0.036 | 0.024±0.002 | 0.307±0.018 | 2.225±0.172 | 0.553±0.033 |
| Traffic | 96 | 0.481±0.000 | 0.028±0.003 | 0.314±0.001 | 3.675±0.005 | 0.693±0.000 | 0.468±0.000 | 0.007±0.000 | 0.293±0.001 | 3.187±0.028 | 0.684±0.000 |
| | 192 | 0.482±0.001 | 0.021±0.000 | 0.313±0.000 | 3.619±0.004 | 0.694±0.000 | 0.469±0.000 | 0.007±0.000 | 0.295±0.000 | 3.181±0.007 | 0.685±0.000 |
| | 336 | 0.480±0.005 | 0.015±0.004 | 0.308±0.006 | 3.428±0.058 | 0.693±0.004 | 0.481±0.000 | 0.006±0.000 | 0.301±0.000 | 3.185±0.009 | 0.694±0.000 |
| | 720 | 0.507±0.005 | 0.022±0.001 | 0.324±0.006 | 3.490±0.038 | 0.712±0.003 | 0.506±0.005 | 0.012±0.004 | 0.323±0.006 | 3.465±0.050 | 0.711±0.004 |
| | AVG | 0.487±0.010 | 0.021±0.004 | 0.315±0.005 | 3.553±0.088 | 0.698±0.007 | 0.481±0.014 | 0.008±0.002 | 0.303±0.011 | 3.255±0.109 | 0.693±0.010 |
| Weather | 96 | 0.170±0.000 | 0.010±0.000 | 0.216±0.000 | 12.814±0.055 | 0.413±0.000 | 0.171±0.000 | 0.010±0.000 | 0.217±0.001 | 12.942±0.059 | 0.413±0.000 |
| | 192 | 0.223±0.001 | 0.013±0.003 | 0.260±0.000 | 13.923±0.100 | 0.472±0.001 | 0.222±0.000 | 0.010±0.002 | 0.259±0.000 | 13.935±0.035 | 0.472±0.000 |
| | 336 | 0.276±0.000 | 0.009±0.001 | 0.297±0.000 | 14.299±0.037 | 0.525±0.000 | 0.276±0.000 | 0.007±0.000 | 0.297±0.000 | 14.302±0.006 | 0.525±0.000 |
| | 720 | 0.350±0.000 | 0.009±0.002 | 0.345±0.000 | 14.876±0.083 | 0.591±0.000 | 0.350±0.000 | 0.006±0.000 | 0.345±0.000 | 14.851±0.003 | 0.591±0.000 |
| | AVG | 0.255±0.059 | 0.010±0.001 | 0.280±0.042 | 13.978±0.674 | 0.500±0.059 | 0.255±0.059 | 0.008±0.002 | 0.280±0.042 | 14.007±0.623 | 0.500±0.059 |

## J    Pronunciation of `a.li.o`

To prevent confusion regarding the pronunciation, this section explicitly describes how to pronounce `a.li.o`.

$$[\text{a.li.o}]$$

The pronunciation consists of three syllables:

- **a**: Pronounced like the 'a' in *father* (ah).
- **li**: Pronounced with a clear 'L' sound, as in *Lee* (lee).
- **o**: Pronounced like the 'o' in *go* (oh).

Consequently, it is pronounced as ***"Ah-Lee-Oh"***, ensuring the 'L' is clearly articulated.

