# OpenReview forum: "AliO: Output Alignment Matters in Long-Term Time Series Forecasting"
_NeurIPS.cc/2025/Conference — NeurIPS 2025 poster_

### Official Review · Reviewer_HfaX · 2025-06-23

**Clarity:** 3
**Significance:** 3
**Originality:** 3
**Rating:** 4
**Confidence:** 4

**Summary:**

This paper addresses output alignment issues in long-term time series forecasting (LTSF) by proposing AliO (Align Outputs), a novel training approach that minimizes prediction discrepancies for overlapping timestamps across lagged input sequences.

**Questions:**

Can you provide concrete evidence or case studies demonstrating that output alignment issues cause significant problems in actual forecasting deployments? What specific industries or applications would benefit most from this approach?

How does AliO compare against ensemble methods or Bayesian approaches that naturally provide prediction uncertainty and consistency? Have you considered comparing against methods that use prediction intervals or confidence bounds?

**Ethical Concerns:**

["NO or VERY MINOR ethics concerns only"]

**Final Justification:**

I have already read all the replies from the authors, I would like to to change my rate to "Borderline accept".

**Limitations:**

yes

**Paper Formatting Concerns:**

No major formatting issues observed. The paper follows NeurIPS formatting guidelines appropriately.

**Quality:**

3

**Strengths And Weaknesses:**

Strengths:
The paper identifies an interesting and previously unaddressed issue in LTSF - inconsistent predictions for the same timestamps when using lagged input sequences, which could impact practical deployment reliability.

Extensive experiments across 7 state-of-the-art models (Autoformer, DLinear, PatchTST, TimesNet, iTransformer, GPT4TS, CycleNet) and multiple datasets demonstrate broad applicability.

Weaknesses:

The core scenario of using lagged input sequences for prediction is artificially constructed and rarely encountered in real forecasting applications, where models typically use the most recent available data. The claimed financial and operational costs from output misalignment lack empirical validation.

The "regression pulling" mechanism appears heuristic without strong theoretical foundations. The stop-gradient operation seems like an ad-hoc solution to prevent performance degradation rather than a principled design choice. The necessity of simultaneous time and frequency domain alignment is insufficiently justified given computational overhead.

The paper assumes that forecasts for the same timestamp should remain consistent, but this ignores the dynamic nature of time series. Forecasts will naturally change as new information is added, which may actually reflect the model's adaptability to new data rather than a flaw.

---

> ### Author Rebuttal · Authors · 2025-07-31
>
> We sincerely thank all the reviewers for their thorough and insightful feedback. We have carefully considered all the comments, which have helped us to clarify and strengthen our paper's contributions. We hope that our detailed responses below successfully address the main concerns raised, and we are committed to incorporating these valuable suggestions into the final camera-ready version of our paper.
>
> ---
> [W1] We thank the reviewer for this question on practical applications. A key feature of AliO is that it is a training-time-only methodology, meaning the inference process is identical to the baseline with no additional overhead. The training scenario, which uses lagged inputs, is a direct reflection of the rolling forecast method used in real-world deployments.
> * **Zero Inference Overhead** AliO is applied as a loss function only during the training phase. Therefore, at inference time, the model operates as a standard single model, making its deployment pipeline and computational cost completely identical to the baseline.
> * **Training for Real-World Rolling Forecasts** The lagged input scenario used during training is not artificial; it is a direct situation of the rolling forecast method. Real-world systems constantly update predictions using a sliding window of the most recent data (as shown in Figure 1). Our training method prepares the model to be consistent and accurate in exactly this type of real-world, sequential deployment.
>
> ---
> [W2] We thank the reviewer for this question. The Regression Pulling (RegPull) mechanism is not a heuristic but a principled design rooted in the well-established methodology of knowledge distillation. It functions as a form of dynamic self-distillation, and the dual-domain alignment is a well-justified choice for recent forecasting.
>
> * **Principled Design via Self-Distillation** RegPull functions as a dynamic self-distillation. At each step, the more accurate prediction acts as a ‘teacher’ for the less accurate ‘student’. The stop-gradient is a standard, principled tool in this framework to ensure a one-way correction, preventing the student’s error propagation from corrupting the teacher and ensuring our alignment loss constructively reinforces the primary regression objective.
> * **Justified Dual-Domain Approach** We align in both time and frequency domains to capture complementary data characteristics, a practice supported by recent theoretical research like [Learning to Forecast in the Frequency Domain], which highlights the importance of using both domains for robust forecasting. The computational overhead of the frequency alignment (FFT) is an efficient O(NlogN).
> * **Training-Time-Only Mechanism** It is important to emphasize that the entire AliO mechanism, including RegPull and the dual-domain loss calculation, is applied only during the training phase.
>
> ---
> [W3] We thank the reviewer for their insightful point. We agree that adaptability is a key virtue of a good forecaster. We argue that AliO enhances this adaptability by enabling the model to adapt with long-term consistency, which is crucial for the specific challenges of LTSF.
>
> * **Distinguishing Adaptability and Consistency** We differentiate between adaptability and consistency. Adaptability is the ability to rationally update predictions based on new information, quantified by accuracy. Consistency is the stability of predictions for the same future point despite minor input changes.
> * **Adaptation in LTSF** For LTSF, which focuses on long-term trends, a good model should not change its forecast drastically based on new short-term information. Instead, it should demonstrate a gradual analysis, reflecting a continuous analysis of long-term trends from the past to ensure a consistent evolution of its forecast (Figure 2).
> * **How AliO Achieves Consistent Adaptation** AliO creates a dependency between past and future data points during the backward pass. The RegPull mechanism guides this process towards the ground truth, ensuring the consistent adaptation is also an accurate adaptation. Figure 2 demonstrates the result, showing how the model learns to adapt consistently from past time steps.
>
> From a more fundamental perspective, this notion of consistent adaptation aligns with the definition of an ideal model, which must ultimately follow the singular Ground Truth.
> * **The Goal of Adaptation** The ultimate goal of an adaptive model is to follow the ground truth. As the reviewer noted, predictions can naturally fluctuate, but the Ground Truth itself for a given future point does not fluctuate.
> * **Low Variance as an Ideal Trait** Therefore, a model that truly follows the Ground Truth should exhibit low prediction variance for that specific future point. Reducing this variance is a step toward becoming a more ideal model.
> * **AliO’s Guiding Principle** Following this principle, AliO guides the model to produce adaptively consistent predictions, enhancing both its accuracy and reliability.
>
> ---
> [W4] We thank the reviewer for this critical question, which prompts us to provide more concrete evidence by connecting our work to established principles in economics and management science.
>
> **Evidence from Economic and Management Science Principles**
> * **The Economic Dilemma of the Planner**
> * * Real-world planners operate under Bounded Rationality, making decisions with limited information and cognitive capacity [A Behavioral Model of Rational Choice].
> * * Output inconsistency forces a costly dilemma: either pay tangible re-planning costs to change course or incur intangible opportunity costs by sticking to a now-outdated plan.
> * * This constant re-evaluation can lead to suboptimal decisions, a pattern often explained by the Sunk Cost Fallacy [The Sunk Cost and Concorde Effects].
>
> * **Systemic Cost Amplification (The Bullwhip Effect)**
> * * This individual planning dilemma often amplifies into massive systemic costs across the supply chain, a phenomenon famously known as the Bullwhip Effect.
> * * Seminal research identifies demand forecast updating as a primary cause of this effect, where small forecast instabilities are magnified upstream [The Bullwhip Effect in Supply Chains].
> * * AliO's goal of improving output alignment is a direct technical approach to mitigate this specific root cause. AliO achieves this through its core mechanisms:
> * * * AliO is designed to achieve two simultaneous goals: it regularizes forecast instabilities via an explicit loss term ($L_{AliO}$), while also ensuring that the model's primary forecasting performance is maintained or enhanced.
> * * * Crucially, the Regression Pulling (RegPull) mechanism ensures this stabilization process is constructive. It forces the alignment to occur only in the direction of the prediction that is closer to the ground truth.
> * * * This principled correction prevents the model from converging to a stable but inaccurate forecast, ensuring the resulting consistency is also reliable while maintaining or enhancing overall forecasting performance.
>
> **Most Benefiting Industries and Applications**
>
> Based on these principles, AliO is most beneficial in industries where such planning dilemmas and systemic effects lead to significant costs. This includes any sector with a critical value chain:
>
> * Manufacturing and Supply Chain Management: Where unstable forecasts directly fuel MRP Nervousness and the Bullwhip Effect.
> * Retail and Consumer Goods: Where inventory and stockout costs are directly tied to forecast accuracy and stability.
> * Energy: Where inconsistent load forecasts lead to inefficient and costly power grid dispatch decisions.
> * Public Health: Where stable forecasts are critical for reliable resource allocation during health crises.
>
> **Commitment to Paper Enhancement**
>
> We thank the reviewer for prompting this clarification. We will enhance this economic justification to the final camera-ready version, taking advantage of the increased page limit to better motivate our work.
>
> ---
> [W5] We thank the reviewer for this important comparison. We argue that AliO is not a direct competitor but a complementary tool to Ensemble and Bayesian methods. While these methods provide crucial model robustness, they do not inherently guarantee the temporal consistency that AliO is specifically designed to enforce.
> * **Different Types of Consistency** The 'consistency' from Ensemble/Bayesian methods refers to model robustness — stability against variations in parameters or initializations, as discussed in works like iTransformer. AliO, however, targets temporal consistency — the stability of predictions for the same future point across slightly shifted time windows.
> * **Lack of Inherent Alignment Mechanism** Ensemble and Bayesian methods lack an explicit temporal alignment algorithm. Averaging a set of temporally inconsistent models will still produce an inconsistent average. Furthermore, without a directional correction mechanism like our RegPull , simple averaging could even pull the final forecast away from the ground truth, whereas AliO always aligns towards the more accurate prediction.
> * **Practical Advantages and Synergy** From a practical standpoint, AliO adds zero inference overhead, unlike ensembles which have a K-fold cost. It is also a computationally efficient loss function. In contrast, Bayesian methods are often significantly more computationally expensive than standard training. AliO can be used with these powerful methods to ensure the central forecast path is stable, while they quantify its uncertainty.

---

> > ### Author Response · Authors · 2025-08-05
> >
> > Dear Reviewer,
> >
> > Thank you for your thoughtful feedback on our submission.
> >
> > In our rebuttal, we have done our best to provide a thorough response to all the points you raised. We are reaching out because the discussion period ends in just one day, and we have not yet received your response.
> >
> > We understand that you are taking the time for careful consideration, and we truly appreciate it. Please let us know if we can provide any further information or clarification that might be helpful.
> >
> > Thank you again for your time and consideration.
> >
> > Best regards,

---

> > > ### Comment · Reviewer_HfaX · 2025-08-06
> > >
> > > Based on the authors' rebuttal, while I appreciate their efforts to address the concerns, I remain unconvinced about the practical significance of the alignment problem and the theoretical soundness of the proposed solution, therefore I will maintain my current rating.

---

> > > > ### Author Response · Authors · 2025-08-06
> > > > **Regarding practical significance of alignment**
> > > >
> > > > Thank you for your valuable feedback.
> > > >
> > > > # Regarding practical significance of alignment
> > > >
> > > > Regarding the question raised during the discussion period about the practical significance of alignment, we'd like to clarify that this is a recognized issue in various fields, including management sectors that adopt rolling forecasts, industrial applications, and weather forecasting.
> > > >
> > > > We have compiled the relevant references and case studies on this matter below.
> > > >
> > > > * [The Impact of Forecast Inconsistency and Probabilistic Forecasts on Users’ Trust and Decision-Making]:
> > > > This report states that national agencies like the National Oceanic and Atmospheric Administration (NOAA) consider forecast consistency a crucial value, arguing that a lack of consistency erodes public trust. The report raises the concern that traditional methodologies have attempted to maintain consistency at the expense of accuracy. As a technique that improves both accuracy and consistency (Table 1, Appendix H), AliO is a more advanced methodology that accomplishes both objectives simultaneously.
> > > >
> > > > * [The Management Forecast Consistency]:
> > > > This report argues that when evaluating forecast quality, consistency (the variability of forecast error) is a more important metric than accuracy (the magnitude of the error). Even if forecast errors are small (high accuracy), if they are random and unpredictable, it is difficult for information users like investors and analysts to trust and utilize those predictions for decision-making. Conversely, it explains that even if forecast errors are large (low accuracy), if the error is consistent and predictable (e.g., always 3% lower than the actual), the information is more valuable because users can adjust for the systematic bias and trust the resulting value. Our methodology improves both accuracy and consistency across multiple predictions (Table 1, Appendix H), thereby resolving both challenges at once.
> > > >
> > > > * [Rolling Forecast Best Practices <WallStreetPrep>]:
> > > > This material describes the difficulty of implementation and management as a primary challenge of rolling forecasts, which generate forecast volatility (misalignment) through constant prediction. It reports that due to the difficulty of achieving consistent predictions, managing rolling forecasts is much harder than managing the static results of a traditional fixed budget, even if the forecasts are accurate. It reports that this consistency problem has led to failure in 20% of companies that adopted rolling forecasts.
> > > >
> > > > * [Rolling forecasting in budgetary systems – case study]:
> > > > This material reports that due to consistency problems, resources are required from employees outside the finance department, consuming unnecessary organizational time. It argues that the resulting re-budgeting—contrary to the original goal of rolling forecasting to reduce resources—actually forces more resources to be invested.
> > > >
> > > > As the reports and case studies above confirm, the alignment problem is a common issue faced in society. Our AliO is a methodology that can effectively solve these problems at the AI model level, as it improves consistency while maintaining or enhancing accuracy.

---

> > > > > ### Author Response · Authors · 2025-08-06
> > > > > **Regarding methodology**
> > > > >
> > > > > We would like to express our sincere gratitude to the reviewers for their thoughtful and constructive feedback.
> > > > >
> > > > > We would like to clarify the mechanism for our approach.
> > > > >
> > > > > When used alongside the regression loss, AliO utilizes a stop-gradient to ensure the alignment loss is always calculated in a direction consistent with the regression loss. By applying the stop-gradient to the prediction that is closer to the ground truth, the model updates are always guided to move the predictions closer to the ground truth.
> > > > >
> > > > > Methodologies that improve performance by using a model's own output as a learning target already exist. The validity of the AliO method, when used with regression loss, can be explained based on established research papers.
> > > > >
> > > > > * Data-Distortion Guided Self-Distillation for Deep Neural Networks
> > > > > * Self-Distillation From the Last Mini-Batch for Consistency Regularization
> > > > > * Consistency Regularization forSelf-Distillation With Augmentation in Feature Space
> > > > >
> > > > > These papers improved model performance by taking two augmented data points as input and training the model to make their outputs similar. AliO is considered a redesign of these methodologies, tailored for time series data. In AliO, lagged inputs can be viewed as a form of augmentation, similar to the approach in TS2Vec; furthermore, AliO advances upon the presented papers by being designed not just to make outputs similar, but to align them toward the better one.
> > > > >
> > > > > More directly, as we mentioned in Section 6, a body of work in time series research also exists that improves performance by enforcing consistency in the latent space for lagged inputs.
> > > > > * Unsupervised Representation Learning for Time Series with Temporal Neighborhood Coding
> > > > > * TimeTS2Vec: Towards Universal Representation of Time Series
> > > > > * Multi-Task Self-Supervised Time-Series Representation Learnin
> > > > > * Self-supervised Learning for Electroencephalogram: A Systematic Survey
> > > > >
> > > > > These works improve model performance by taking lagged inputs, which have an overlapping period, and training the model to make the corresponding latent space representations identical.
> > > > >
> > > > > AliO was developed by adapting these preceding methodologies, which in turn supports the validity of our own approach.

---

### Official Review · Reviewer_Q8Hn · 2025-06-28

**Clarity:** 3
**Significance:** 3
**Originality:** 4
**Rating:** 5
**Confidence:** 4

**Summary:**

This paper identifies a crucial but overlooked issue in long-term time series forecasting (LTSF): the lack of consistency when models generate predictions for overlapping futrue timestamps using lagged input windows. To address this, the authors introduce Alio, a simple yet effective framework that explicitly aligns prediction outputs by minimizing discrepancies across overlapping segments in both the time and frequency domains.

To quantify this aspect, the authors propose Time Alignment Metric (TAM) to measure the consistency of overlapping outputs. To enhance the consistency, the authors design a novel loss function by adding an alignment loss alongside traditional forecasting objective, using a regression pulling mechanism with a stop-gradient to ensure that the improving alignment does not degrade predictive accuracy.

The authors conduct extensive experiments on multiple benchmark datasets and popular LTSF models to demonstrate the effectiveness of the method.

**Questions:**

1. How do authors get the results of baselines? As Alio framework includes forecasting objective, it will inherently enhance the performance on MSE metric. For fair comparison, I think the authors should ensure that the training of baselines are sufficient.

**Ethical Concerns:**

["NO or VERY MINOR ethics concerns only"]

**Final Justification:**

I think authors have resolved all my concerns. So I keep my score to support this paper.

**Limitations:**

yes

**Quality:**

3

**Strengths And Weaknesses:**

Strengths:

* This paper identifies a crucial yet previously overlooked problem in long-term time series forecasting: existing models often fail to produce consistent predictions for overlapping forecasting windows. To address this, the authors introduce a new metric, TAM, which effectively captures prediction alignment and complements traditional error-based metrics such as MSE and MAE.

* The paper proposes a novel method that successfully mitigates these inconsistencies and demonstrates its effectiveness through comprehensive experiments.
* This paper also provides solid mathematical proofs, which strengthen the theoretical foundation of the work.

* Overall, the paper is clearly written and easy to follow, with well-designed and readable figures that help illustrate the key ideas.

Weaknesses:

* For metric design, authors should propose more showcases or human study results to demonstrate the metric could effectively measure the degree of alignment.
* The related work section could be expanded. A more comprehensive review and clearer positioning with respect to recent advances would help highlight the unique aspects of this work and situate its contributions more effectively.
* There are some typos/errors. For example, in Traffic results of Table 1, AliO achieve 0.280 in GPT4TS in TAM, which is higher than baseline (0.059). But the authors use bold to indicate it is the best result. Moreover, also in this table, some results are significantly higher than baseline. The authors should also check the correctness.

---

> ### Author Rebuttal · Authors · 2025-07-31
>
> We are grateful to all the reviewers for their comprehensive and constructive feedback. We have carefully considered every comment, which has helped us to clarify and strengthen our paper's contributions.
>
> ---
> Thank you for correcting the typo. The results for 'Traffic' in Table 1 were indeed incorrect (they were shifted). The actual results are as follows.
>
> |Models|CycleNet||GPT4TS||iTransformer||PatchTST||TimesNet||DLinear||Autoformer||
> |-|-|-|-|-|-|-|-|-|-|-|-|-|-|-|
> |Metric|MSE|TAM2|MSE|TAM2|MSE|TAM2|MSE|TAM2|MSE|TAM2|MSE|TAM2|MSE|TAM2|
> |baseline|0.487|0.021|0.411|0.059|0.422|0.061|0.389|0.041|0.626|0.04|0.436|0.025|0.634|0.048|
> |AliO|0.481|0.008|0.405|0.032|0.423|0.044|0.388|0.027|0.554|0.03|0.434|0.009|0.624|0.045|
>
> ---
>
> **Core Motivation: The Economic Costs and Foundational Questions of Inconsistency**
>
> Before addressing the specific points, we wish to reiterate the core motivation for this work. Our research was driven by a set of fundamental questions regarding the practical reliability of forecasting models:
> * Can a model whose predictions for the same future point fluctuate erratically with minimal new data truly be considered ideal?
> * Can a user trust a system whose predictions are constantly volatile?
> * Given two systems with equal accuracy, will a user not always prefer the stable system over the unstable one?
>
> We believe the answers to these questions highlight the critical need for output alignment. Prediction inconsistencies are not a theoretical issue; they cause  sunk costs when plans are modified and opportunity costs when they are maintained, leading to significant real-world losses. These losses can amplify across an entire organization through phenomena like MRP Nervousness and the Bullwhip Effect. Therefore, our methodology is critical in fields where value chains are vital, such as energy, public health, manufacturing, and supply chain management.
>
> ---
> [W1] We thank the reviewer for this valuable suggestion on validating our metric. Our paper demonstrates TAM's effectiveness through visual showcases and a practical, quantitative design.
> * **Visual Showcases**: The paper provides several visual showcases of TAM's effectiveness. In Figure 2 and the figures in Appendix C, it is visually clear that scenarios with high TAM values exhibit poor alignment, while scenarios with low TAM values show strong, consistent alignment .
>
> * **Quantitative & Practical Design**: TAM is designed to be a direct and interpretable metric for planning. By calculating the point-wise MAE between predictions for the same timestamp, it directly quantifies the value difference that is critical for resource allocation. This differs from shape-based metrics like DTW. However, the TAM framework is flexible, and for applications where shape consistency is the priority, DTW could certainly be adopted as the distance function within TAM.
>
> * **User Study as Future Work**: We agree that a user study is a valuable validation approach. Given that this is a technical paper proposing a new quantitative method, we prioritized establishing an objective and reproducible metric first. We take this suggestion seriously and are considering a user-focused study as a separate, independent paper to properly explore the perceptual aspects of alignment.
>
> ---
> [W2] We thank the reviewer for this valuable suggestion. We agree that a more comprehensive related work section will better situate our contribution. Taking advantage of the increased page limit for the camera-ready version, we will expand this section to more clearly differentiate AliO from two key areas: regression losses and time-series contrastive learning.
> * **vs. Regression Losses (e.g., MSE, Soft-DTW)**: While regression losses focus on improving regression accuracy (the relationship between prediction and ground truth), AliO is distinct in that it improves reliability/consistency (the relationship between predictions) while maintaining or enhancing accuracy via the RegPull mechanism. Our metric, TAM, is similarly designed to quantify this reliability, whereas standard metrics quantify accuracy.
> * **vs. Contrastive Learning (e.g., TS2Vec)**: Time-series contrastive learning methods that design inputs to have overlapping timestamps [Towards Universal Representation of Time Series, Unsupervised Representation Learning for Time Series with Temporal Neighborhood Coding] align the overlapping inputs in the latent space. In contrast, AliO performs alignment directly in the output space. Critically, our RegPull mechanism ensures this alignment process is constructive, simultaneously aligning predictions with each other and with the ground truth, which is a unique contribution.
>
> ---
> [W4] We thank the reviewer for this critical question on experimental fairness. We ensured a fair comparison through two key principles: 1) faithfully reproducing strong, well-trained baselines, and 2) designing our RegPull mechanism to be a constructive, rather than merely additive, loss.
>
> * **Faithful Baseline Reproduction**: To ensure our baselines were sufficient and strong, we used the official code and hyperparameters (e.g., learning rate, epochs) from each SOTA paper . Our reproduced baseline performance closely matches the results reported in the original papers, confirming they are well-trained.
> * **Principled Loss Design (RegPull)**: To ensure AliO's loss term does not unfairly enhance performance, the RegPull mechanism is designed to be constructive. It ensures alignment occurs only in the direction of the better prediction (the one closer to the ground truth) . This prevents the alignment loss from conflicting with the primary accuracy objective.
> In fact, our internal experiments confirmed that removing RegPull worsened MSE performance by 1-3%. This demonstrates that the MSE improvement is a direct result of the intentionally designed RegPull algorithm, not merely an artifact of adding a [regression loss + new loss].

---

> > ### Comment · Reviewer_Q8Hn · 2025-08-03
> >
> > Thank you very much for your response. I will keep my score to support this paper.

---

> > > ### Author Response · Authors · 2025-08-04
> > > **Thank you for your constructive feedback**
> > >
> > > Dear Reviewer Q8Hn,
> > >
> > > Thank you very much for your constructive feedback on our paper. We sincerely appreciate your comments; they have provided us with an opportunity to further highlight the significance of our research and have helped us strengthen the manuscript.
> > >
> > > We will ensure that all the typos you pointed out are corrected in the camera-ready version. We will also incorporate the other valuable suggestions you have made.
> > >
> > > Thank you again for your time and effort.
> > >
> > > Best regards

---

### Official Review · Reviewer_5d8j · 2025-07-01

**Clarity:** 3
**Significance:** 2
**Originality:** 3
**Rating:** 3
**Confidence:** 4

**Summary:**

This paper addresses the problem of improving output consistency in long-term time series forecasting. The authors propose a method called OliO, which jointly considers alignment in both the time and frequency domains to reduce inconsistencies in forecast outputs. They also introduce a new evaluation metric—Time Alignment Metric (TAM)—to quantify consistency. To demonstrate the effectiveness of their approach, the authors conduct experiments using multiple model architectures across seven standard benchmark datasets and show that OliO can reduce TAM, indicating improved forecasting consistency.

**Questions:**

- **Figure 7** makes it difficult to clearly assess the gains over baselines. More comparisons—similar to table 1—across different values of \(N\) and \(l\) would better highlight the effectiveness of OliO.
- The paper does not specify the **context length** used during training. It would be valuable to understand how varying context lengths impact the effectiveness of OliO.
- **Figure 6** suggests that increasing the time alignment coefficient consistently improves performance. Why not explore even larger values to assess the full potential of the method?
- It remains unclear whether a model trained on a specific lag value \(l\) can **generalize** to other lag settings. This is an important aspect of the method’s robustness.
- Is the **inconsistency issue** targeted by OliO specific to long-term forecasting, or does it also appear in short- and mid-term settings? A broader evaluation would clarify the scope of the problem.

**Ethical Concerns:**

["NO or VERY MINOR ethics concerns only"]

**Final Justification:**

Thank you for the rebuttal. While it partially addressed my concerns, I agree with another reviewer that, although the problem is interesting, its practical significance and theoretical soundness remain in question. Given the overall status of the paper, I tend to maintain my current score.

**Limitations:**

- A more extensive exploration of hyperparameter settings (e.g., N, l) is needed to fully assess the robustness of the proposed method.

- The paper would benefit from a more thorough discussion and comparison with existing approaches, particularly recent pretrained time series foundation models.

- A broader range of datasets should be evaluated to better understand the generalizability of the method across diverse forecasting scenarios.

**Quality:**

3

**Strengths And Weaknesses:**

### **Strengths**
- The paper tackles an interesting and underexplored problem: improving output consistency in long-term time series forecasting (LTSF).
- The proposed approach, OliO, incorporates both time-domain and frequency-domain alignment, which is a thoughtful and well-motivated design. The final loss function is carefully crafted to encourage consistent  as well as accurate predictions.
- The method is evaluated across various LTSF architectures, demonstrating its potential generality.
- A new evaluation metric—Time Alignment Metric (TAM)—is introduced to quantify consistency, contributing a novel perspective to the field.

### **Weaknesses**
- While consistency is an important property, a good forecaster should ultimately aim to match the ground truth. In many cases, lower consistency may indicate appropriate variability or uncertainty. The paper does not compare against models trained with **Weighted Quantile Loss**, which explicitly models uncertainty and could offer a more principled baseline than OliO.
- The study only evaluates a very limited range of lag values (\(l\)), with Table 1 focusing solely on \(l=1\). This setting is not representative of practical long-term forecasting, and limits insight into the generalization capabilities of the proposed method. Results in Figure 7 further suggest that OliO underperforms for larger \(l\).
- The paper omits discussion and evaluation of recent **pretrained time series foundation models** (e.g., Morai, Morai-MoE, Time-MoE, Chronos). A comparison against these strong baselines is essential to assess the true effectiveness of OliO.
- The results on the **Traffic** dataset raise concerns (not sure if this a typo) : OliO appears to degrade alignment significantly
- TAM is already low on these datasets. The authors should consider identifying datasets with higher TAM values to better evaluate the method’s strengths and limitations.

---

> ### Author Rebuttal · Authors · 2025-07-31
>
> We are grateful to all the reviewers for their comprehensive and constructive feedback. We have carefully considered every comment, which has helped us to clarify and strengthen our paper's contributions.
>
> We apologize for the typo in Table 1. Due to the character limits of the rebuttal platform, the corrected results have been organized in a clear table in our first response to reviewer Q8Hn. We would appreciate it if you could refer to that section for the accurate figures.
>
> ---
> [W1] We agree that a good model must align with the Ground Truth, and we argue that this fundamental premise necessitates the temporal consistency that AliO is designed to enforce.
> * **Purpose of AliO** The purpose of AliO is maintaining or enhancing the regression performance while enhancing the consistency. This means the purpose of AliO aims to match the ground truth while the prediction is stable.
>
> * **Ideal Model’s Requirement** An ideal model predicting a single, unique Ground Truth for a future point T must produce the same output, regardless of minor input variations (e.g., from $X_n$ to $X_{n+1}$). If the model's output changes for the same single time point, it either contradicts its ideal nature (by failing to pinpoint the single Ground Truth) or implies a physical impossibility (multiple Ground Truths for one time point). Therefore, consistency is an essential property of an ideal model. AliO is the mechanism designed to explicitly enforces this property, which standard regression loss only addresses implicitly.
>
> Regarding the comparison to Weighted Quantile Loss (WQL), we posit that AliO and WQL are not competing baselines but are complementary tools that address different, essential aspects of a reliable forecast.
> * **WQL** is designed to quantify uncertainty by providing a prediction interval. It answers the question: “What is the probable range where the true value might lie?”. It cannot guarantee both accuracy and temporal consistency.
> * **AliO** is designed to ensure consistency. It answers the question: “Is the single most probable prediction path stable and reliable over time? It can guarantee consistency while maintaining or enhancing performance (Table 1 and Appendix H)”
>
> An ideal system could use both WQL to define the uncertainty bounds and AliO to ensure the central forecast within those bounds is stable. A direct comparison is thus less a baseline and more a future direction for a combined approach.
>
> ---
> [W2,6,9,11] We thank the reviewer for these important questions regarding the hyperparameter l and generalization. Our focus on l=1 is a deliberate choice to address the most fundamental case of temporal alignment, and our results indicate that this approach confers a more general alignment to the model.
> * **Rationale for Focusing on l=1** We centered our analysis on l=1 as it represents the most fundamental scenario. A model that is inconsistent for immediately adjacent inputs exhibits a core instability. Furthermore, this setting creates a chain of local alignments: the prediction at time t (Y_t) is aligned with Y_{t+1}, which is then aligned with Y_{t+2}, and so on. This series of constraints encourages the model to learn a global alignment over larger lags as an emergent property.
> * **Interpreting Performance at Larger l (Figure 7)** The performance trend for larger l in Figure 7 reflects the task’s inherent difficulty, not a limitation of AliO. As l increases, the overlapping window for alignment shrinks, making the task harder. The MSE results show a trade-off between l and N, indicating a need to find an appropriate context window size, as discussed in Section 5.3.
> * **Evidence of Generalization** Our results suggest that a model trained on a specific $l$ does generalize. The visualizations in Figure 2 and Appendix C show that a model trained with a fixed l exhibits improved alignment across a wider range of effective lags (e.g., Prediction 0 through 5) compared to the baseline.
> * **Commitment to Further Analysis** To address this, we will add more detailed tabular comparisons for different (N, l) settings to the appendix in the camera-ready version.
>
> The table below shows the extended results normalized for each lag value (l = 16, 32, 48, 64).
>
> |iTransformer-ETTh1 (MSE)|16|32|48|64|
> |-|-|-|-|-|
> |N=2|0.59|0.59|0.60|0.61|
>
> |iTransformer-ETTh1 (TAM)|16|32|48|64|
> |-|-|-|-|-|
> |N=2|0.16|0.38|0.54|0.70|
>
> ---
> [W3,W12] We thank the reviewer for this important suggestion regarding recent foundation models. Our experiments were designed to rigorously validate AliO's primary strength as a model-agnostic method, and the results suggest its applicability to these newer architectures.
> * **Proven Model-Agnostic Performance** Our main goal was to demonstrate that AliO is a model-agnostic technique. We validated this by applying it to a diverse range of architectures — including Transformer, Linear, CNN, and even a Pre-trained Model (GPT4TS) — and showed the performance of AliO across all of them.
> * **Expected Efficacy on PFMs** Since recent foundation models like Chronos or Morai are built upon these fundamental architectural principles (e.g., Transformer, MLPs), we anticipate that AliO, as a model-agnostic loss function, would similarly enhance their output consistency.
> * **Commitment to Further Validation** We agree that an explicit experiment on a recent PFM would further strengthen our paper. While this is not feasible during the short rebuttal period, we will add these results to the appendix of the camera-ready version to enhance our contribution.
>
> ---
> [W10,13] The inconsistency problem does appear in short-term settings; however, AliO is most effective in LTSF due to its reliance on a long overlapping sequence for alignment.
> * **Problem Presence in Short-Term** The inconsistency issue is not exclusive to long-term horizons. As shown in Figure 2, significant variance between predictions exists even in the short- and mid-term intervals of the forecast.
> * **Methodological Suitability for LTSF** AliO’s effectiveness relies on having a sufficiently long overlapping sequence to perform alignment. LTSF provides these long overlapping windows, making it the ideal setting. In short-term forecasting, the overlapping period is too brief for the alignment mechanism to be as impactful, a trend suggested by the results in Figure 7 and Appendix H.
> * **Commitment to Broader Scope** We agree that a broader evaluation is a valuable direction. We will add a discussion to the camera-ready paper clarifying this scope and explicitly mentioning the application to short-term forecasting as an area for future work.
>
> ---
> [W5] We thank the reviewer for this suggestion, which allows us to highlight a key strength of our experimental design. Our results demonstrate that AliO is highly effective in the high-TAM scenarios you suggested, and also provides significant benefits for datasets with already low TAM.
> * **Proven Efficacy in High-TAM Scenarios** We included the ILI dataset specifically to test our method in a high-TAM environment. As shown in Table 2 and Figure 4, on this dataset, AliO improved TAM by up to 58.1% and, critically, improved the regression metric (MSE) by up to 17.4%, demonstrating its strength in the exact scenario you described.
> * **General Applicability in Low-TAM Scenarios** AliO is also effective on standard benchmarks with lower initial TAM. For instance, on models like iTransformer and Autoformer, AliO still improved TAM by an average of 31.08% and MSE by 6.79% (Appendix H), showcasing its broad utility.
> * **Theoretical Rationale** This performance characteristic is explained by viewing AliO as a form of iterative self-distillation. The large initial gap (high TAM) between the teacher (better prediction) and student (worse prediction) provides a strong learning signal, leading to greater performance gains, a principle supported by recent distillation research [Understanding the Gains from Repeated Self-Distillation]. [This point is also discussed in our response to reviewer HfaX.]
>
> ---
> [W8] We thank the reviewer for this insightful question. As you mentioned TAM performance in Figure 6 improves as the $\lambda_T$ increases. This allows us to clarify the trade-off in our methodology. Our goal is not just to maximize consistency at all costs, but to balance between improving consistency and maintaining or improving forecasting performance (MSE).
> * **Trade-Off Between Consistency and Accuracy** While it is true that TAM performance improves with a larger coefficient, as you rightly observed, the model’s core performance (MSE) does not follow the same trend. The optimal average MSE was already achieved at a coefficient ($\lambda_T$= 2), after which performance began to degrade, as shown in Figure 6.
> * **Risk of Overfitting to the Alignment Objective** This indicates that exploring even larger coefficient values would likely continue to improve TAM but at the cost of overfitting to the consistency objective. This would diminish the influence of the primary regression loss, which is counterproductive. (While RegPull exists, its purpose is to ensure the direction of the alignment loss matches that of the regression loss; it is not a direct regression loss itself.)
>
> To further validate these findings, we will add experiments with a broader coefficient range to the appendix.
>
> ---
> [W7] We thank the reviewer for pointing out this omission and for the valuable suggestion.
>
> Our experiments used the official context lengths from each SOTA paper. These specifics, listed below, will be added to the appendix for full transparency:
> * CycleNet, Autoformer, iTransformer: 96
> * TimesNet: 96 (36 for the ILI dataset)
> * PatchTST, DLinear: 336 (104 for the ILI dataset)
>
> We agree that analyzing AliO's effectiveness across various context lengths is a valuable direction for future work. However, our study's primary goal was to isolate the performance improvement solely from AliO under conditions identical to the SOTA baseline settings.

---

> > ### Author Response · Authors · 2025-08-05
> >
> > Dear Reviewer,
> >
> > Thank you again for your insightful review of our submission.
> >
> > We have provided a detailed response to your points in our rebuttal. As the discussion period deadline is tomorrow, we wanted to gently follow up.
> >
> > We trust that you are giving the matter careful thought, and we are ready to provide any additional materials if needed.
> >
> > Thank you for your attention to our work.
> >
> > Sincerely,

---

> > > ### Author Response · Authors · 2025-08-05
> > > **Additional data for Figure 7**
> > >
> > > To further elaborate on Figure 7, we ran additional experiments on the TimesNet ETTh1 dataset. We extended the lag to 64 and increased num_samples (N) to {2, 4, 8, 16} across prediction lengths of 96, 192, 336, and 720 (detailed in the Appendix). The full range of lag values {1, 2, ..., 64} was tested for all prediction lengths except for 96.
> > >
> > > The results show that, similar to our findings in Section 5.3, TAM performance is generally superior with a small lag and a large N.
> > >
> > > Moreover, as seen in Figure 7, both TAM and MSE performance tend to decline as the lag increases. Nevertheless, we also noted temporary improvements in MSE at points like lag=48. This is because, as discussed in Section 5.3, a larger lag indirectly increases the size of the context window.
> > >
> > > ### Prediction Length = 96
> > >
> > > |MSE|lag = 1|2|4|8|16|32|
> > > |-|-|-|-|-|-|-|
> > > |num_samples = 16|0.084682|0.604413|0.673358|-|-|-|
> > > |8|0.104499|0.393956|0.744772|0.73569|-|-|
> > > |4|0.121304|0.215247|0.668411|0.842757|0.850005|-|
> > > |2|0|0.143544|0.400155|0.952206|0.944189|1|
> > >
> > >
> > > |TAM|lag = 1|2|4|8|16|32|
> > > |-|-|-|-|-|-|-|
> > > |num_samples = 16|0.391769|0.374947|0.304065|-|-|-|
> > > |8|0.155449|0.144612|0.370511|0.286691|-|-|
> > > |4|0|0.048982|0.128199|0.31205|0.31205|-|
> > > |2|0.514371|0.310482|0.148924|0.489335|1|0.916545|
> > >
> > > ### Prediction Length = 192
> > >
> > > |MSE|lag = 1|2|4|8|16|32|48|64|
> > > |-|-|-|-|-|-|-|-|-|
> > > |num_samples = 16|0|0.57251|0.615467|-|-|-|-|-|
> > > |8|0.056027|0.340819|0.677425|0.654278|0.671174|-|-|-|
> > > |4|0.062936|0.145248|0.593142|0.737499|0.727006|0.770648|-|-|
> > > |2|0.023953|0.108073|0.34194|0.921574|0.890429|0.909058|0.050314|1|
> > >
> > >
> > > |TAM|lag = 1|2|4|8|16|32|48|64|
> > > |-|-|-|-|-|-|-|-|-|
> > > |num_samples = 16|0.30632|0.419096|0.433462|0.460702|-|-|-|-|
> > > |8|0.101933|0.224263|0.334436|0.32699|0.357816|-|-|-|
> > > |4|0|0.057982|0.230441|0.266493|0.377638|0.409745|-|-|
> > > |2|0.506196|0.129541|0.082475|0.350737|0.864099|0.421465|0.674514|1|
> > >
> > >
> > > ### Prediction Length = 336
> > >
> > > |MSE|lag = 1|2|4|8|16|32|48|64|
> > > |-|-|-|-|-|-|-|-|-|
> > > |num_samples = 16|0|0.405|0.464225|0.480078|0.507374|-|-|-|
> > > |8|0.100982|0.287249|0.543376|0.542814|0.563105|0.571398|-|-|
> > > |4|0.093844|0.167532|0.557489|0.665419|0.667408|0.667576|-|-|
> > > |2|0.167667|0.069781|0.279253|0.822194|0.872746|0.847946|0.144109|1|
> > >
> > >
> > > |TAM|lag = 1|2|4|8|16|32|48|64|
> > > |-|-|-|-|-|-|-|-|-|
> > > |num_samples = 16|0.878688|0.78596|0.994856|0.985302|0.572348|-|-|-|
> > > |8|0.620765|0.397376|0.675665|0.910008|1|0.963233|-|-|
> > > |4|0.627545|0.582182|0.663796|0.744864|0.643243|0.659094|-|-|
> > > |2|0.695673|0.797483|0.458442|0|0.310831|0.065214|0.733293|0.556883|
> > >
> > >
> > >
> > > ### Prediction Length = 720
> > >
> > > |MSE|lag = 1|2|4|8|16|32|48|64|
> > > |-|-|-|-|-|-|-|-|-|
> > > |num_samples = 16|0|0.405|0.464225|0.480078|0.507374|-|-|-|
> > > |8|0.100982|0.287249|0.543376|0.542814|0.563105|0.571398|-|-|
> > > |4|0.093844|0.167532|0.557489|0.665419|0.667408|0.667576|-|-|
> > > |2|0.167667|0.069781|0.279253|0.822194|0.872746|0.847946|0.144109|1|
> > >
> > >
> > > |TAM|lag = 1|2|4|8|16|32|48|64|
> > > |-|-|-|-|-|-|-|-|-|
> > > |num_samples = 16|0.158379|0.431538|0.480193|0.458286|0.464605|-|-|-|
> > > |8|0.166214|0.245022|0.391437|0.358413|0.375742|0.390881|-|-|
> > > |4|0.151747|0.14241|0.39425|0.423479|0.421364|0.452096|-|-|
> > > |2|0.053845|0.131887|0.240306|0.674344|1|0.52441|0|0.933229|

---

### Official Review · Reviewer_J5tA · 2025-07-03

**Clarity:** 2
**Significance:** 3
**Originality:** 2
**Rating:** 4
**Confidence:** 4

**Summary:**

This paper introduces a method for aligning predictions of time series samples that share overlapping prediction periods. The proposed alignment technique ensures consistency across both temporal and spectral domains. To quantify the alignment of time series samples that share overlapping periods, the authors propose a new evaluation metric called the Time Alignment Metric.

The approach is evaluated through integration with several state-of-the-art models and tested on multiple standard multivariate time series datasets, demonstrating its potential to improve prediction consistency in overlapping time periods.

**Questions:**

### Implementation Concerns
The paper raises significant questions about the practical implementation of the proposed AliO method. I would disagree with the statement "not [...] requiring modifications to the model". A crucial implementation detail involves how batches of samples are created and processed. For AliO to function effectively, it needs access to both the current sample ($x_k$) and previous samples ($x_{k-t}$) with their corresponding predictions ($y_k$ and $y_{k-t}$), where $t \in ${$1$, ..., number_previous_samples}. However, most time series models process samples in batches that typically don't maintain chronological ordering, making it unclear how the authors ensure access to consecutive samples.

The code provided only includes the AliO implementation itself, not its integration with existing prediction models. This omission makes it difficult to assess how modifications were made to models like DLinear, CycleNet, or iTransformer, where timestamps are typically separated from the data. Additionally, the implementation code for the Time Alignment Metric (TAM) is missing, which is essential for reproducing the results and understanding the evaluation methodology.

### Motivation and Practical Relevance
While the concept of aligning predictions for overlapping time periods is theoretically interesting, its practical motivation requires further justification. The assumption that close samples should provide similar forecasts isn't necessarily problematic, but strict alignment may not always be necessary or beneficial in real-world applications. In resource planning scenarios, the temporal resolution of forecasts is crucial. For hourly or half-hourly data, aligning outputs for consecutive timesteps may not provide significant practical benefits, as planning and allocation decisions are typically made at coarser temporal resolutions (e.g., daily).

The paper's motivation should address how often planning and allocation processes are updated in practice. Many real-world systems run models at regular intervals (e.g., daily for electricity consumption forecasting) rather than at the highest available resolution. For weekly datasets like ILI, alignment makes sense as consecutive timesteps represent different weeks. However, for hourly datasets like ETT or Electricity, the benefits are less clear. In addition, Figure 2 shows that predictions diverge significantly at the beginning of overlapping intervals but converge toward the end, suggesting that alignment might not be critical for all applications, depending on the planning horizon.

### Theoretical Considerations
The theoretical justification for alignment needs clarification. If a model minimizes error such that $\hat{y}$ is close to $y$ based on $x$, and $\hat{y_l}$ is close to $y_l$ based on $x_l$, then $\hat{y}[1:]$ should naturally be close to $\hat{y_l}[:-1]$ because they're compared to the same ground truth $y[1:] = y_l[:-1]$. This suggests that some alignment is inherently ensured through proper training. The authors argue that differences in input lead to misalignment, implying that the method effectively makes the model "remember" previous predictions. This raises concerns about error propagation—if an initial prediction is poor, subsequent aligned predictions might compound this error.

### Reproducibility concerns include:

 * Missing values for λ_T and λ_F in Table 1
 * Unspecified input and prediction lengths for Table 1 results
 * Unclear whether results represent averages across multiple runs
 * Undisclosed datasets used for Figures 6 and 7, with a need for similar plots across all datasets to avoid overgeneralization

### Clarity
Several clarity issues should be addressed:

 * The phrase "the same timestamps" is misleading—"overlapping timestamps across different predictions" would be more accurate
 * Table 1 should consistently highlight improvements (e.g., GPT4TS and ECL results should match the formatting used for CycleNet and Weather)
 * Results for DLinear and Traffic with AliO appear reversed, showing worse TAM performance—such performance needs further discussion why TAM is worsened where it should be better as it is specifically designed to align
 * The claim of "achieving gains of 70.5% for CycleNet" should specify that this is the maximum gain for ETTh1, not an average (the average gain is 50%, with minimum 12.5% and maximum 70.45%)
 * The notation for lag samples should use n - l to clearly indicate historical samples

### Conclusion
Given these concerns, particularly regarding implementation details and practical motivation, I currently score this submission as borderline reject. However, my assessment could change during the rebuttal phase depending on how well the authors address these points, especially the implementation questions, and based on discussions with other reviewers. A strong rebuttal that clarifies the implementation details and better justifies the practical relevance of strict alignment could potentially improve my evaluation of this work.

**Ethical Concerns:**

["NO or VERY MINOR ethics concerns only"]

**Final Justification:**

Considering a proposed revisions including an enhanced narrative, improved figures, additional results and visualizations, deeper insights into the key issues, and expanded discussions of the results in light of new findings and rebuttal points, I believe the submission has strong potential for NeurIPS. My primary remaining concern is whether the authors can effectively incorporate all these improvements and clearly present them in the camera-ready version.
However, I'm willing to reconsider and increase my evaluation score to borderline accept, in recognition of the progress made.

**Limitations:**

### Comparison and Evaluation Needs
A comparison between the proposed Time Alignment Metric (TAM) and Dynamic Time Warping (DTW) would provide valuable context for understanding TAM's advantages and limitations. Why using TAM instead of DTW for overlapping period? This comparison should address how TAM differs from established alignment metrics and what specific benefits it offers for time series prediction tasks.

### Ablation Study Limitations
The current ablation study lacks sufficient clarity in the main paper regarding the individual contributions of $\lambda_F$ and $\lambda_T$ components. Specifically, the analysis should more clearly demonstrate the impact of each component when the other is set to zero ($\lambda_F=0$ or $\lambda_T=0$).

### Results Discussion
The paper would benefit from a more comprehensive discussion of the results. Currently, there's insufficient analysis of whether the observed improvements are consistent across all datasets or vary significantly between them. A detailed examination of the homogeneity of improvements would help readers understand the method's general applicability. Furthermore, the paper should provide insights into why certain datasets show greater benefits from the proposed alignment method. Are there specific characteristics (such as temporal resolution, seasonality patterns, or noise levels) that make some datasets more responsive to the alignment technique? Offering hypotheses or intuitions about these variations would significantly enhance the paper's contribution.

**Quality:**

2

**Strengths And Weaknesses:**

### Strengths
 * Proposes an interesting method for aligning predictions across overlapping time periods in both temporal and spectral domains
 * Integrates the method with multiple state-of-the-art models and tests on common multivariate time series datasets
 * Introduces a Time Alignment Metric (TAM) specifically designed to evaluate alignment accuracy

### Weaknesses
 * Lacks clear explanation of how consecutive samples are accessed in batch processing
 * Provides only AliO implementation without showing integration with existing models or TAM implementation
 * Needs better justification for strict alignment requirements in real-world applications
 * Lack demonstration of the necessity of TAM (why not using regular metrics on the overlapping period)
 * Lacks analysis of improvement consistency across datasets and explanations for varying performance
 * Contains formatting inconsistencies and unclear result highlighting

---

> ### Author Rebuttal · Authors · 2025-07-31
>
> We thank the reviewers for their thorough and insightful feedback. We have carefully considered all comments and will revise the paper to reflect these valuable suggestions (including typo). Below, we address each point in detail.
>
> ---
> We apologize for the typo in Table 1. Due to the character limits, the corrected results have been organized in a clear table in our first response to reviewer Q8Hn. We would appreciate it if you could refer to that section for the accurate figures.
>
> ---
> [W5,21] [Improvements] Regarding the suggestion to delve deeper into performance variations, our analysis shows that the primary factor determining AliO’s performance gain is the baseline model’s initial instability on a given dataset, which we quantify as the initial TAM [Section 5.2].
> * **Empirical Evidence** As shown in Figure 4, there is a positive correlation between a high initial TAM (poor initial alignment) and greater MSE improvement. This is confirmed by the results on the high-TAM ILI dataset, where AliO achieved the largest MSE gain of up to 17.4%
> * **Theoretical Mechanism (Iterative Dynamic Self-Distillation)** This performance gain is driven by AliO's mechanism, which functions as an iterative, dynamic self-distillation. At each overlapping timestamp, the more accurate prediction (closer to the ground truth) acts as a dynamic Teacher for the less accurate one (the Student) . When the initial gap between them (a high Initial TAM) is large, the alignment loss provides a strong and informative gradient, guiding the model toward a more accurate solution. This principle, where a larger initial gap allows for greater gains through repeated distillation, is supported by recent literature [Understanding the Gains from Repeated Self-Distillation]. Also, for well-aligned data at the beginning (Low initial TAM), it is natural that there is little room for performance improvement since it is already well-aligned.
> * **Root Cause and Future Analysis** We hypothesize that the data characteristics (especially high noise) you mentioned are the root causes that result in a high initial TAM since high noise makes the model increase the prediction variation. Crucially, this means the initial TAM is a characteristic of the baseline model’s performance on the data, measured before our method is applied; it is not an artifact of AliO. Acknowledging this valuable feedback, we will add a dedicated analysis of this relationship in the final paper to further strengthen our contribution.
>
> ---
> [W1,2,7] Regarding the implementation, we acknowledge this section could be clearer and will revise the paper as follows:
> * **Clarification of No Model Modification** We will clarify in the paper that our claim refers to the model’s architecture (e.g., Transformer blocks, forward function), which remains entirely unchanged. This highlights that AliO is a model-agnostic method.
> * **Explanation of Data Pipeline** We agree that the data loading pipeline requires adjustment. We will add a detailed explanation to the appendix demonstrating how we access consecutive samples by pairing them in the Dataset class. This is a standard preprocessing technique in time-series [Towards Universal Representation of Time Series, Unsupervised Representation Learning for Time Series] and does not alter the core model.
> This modified pipeline is used only during the training phase and remains the same as baseline during the inference phase.
> * **Ensuring Reproducibility** To address reproducibility concerns, we provide the main GitHub link in Abstract.
>
> ---
> [W10,11,12] In response to the request for Table 1: All results presented in Table 1 are the average of three runs with different random seeds [Table 1 caption].
> * **Input and Prediction Lengths** The specific prediction lengths used for each dataset are detailed in Appendix F. We acknowledge that the context lengths were not explicitly listed. They followed the official SOTA settings, and we will add this to the appendix F for full transparency:
> * * CycleNet, Autoformer, iTransformer: 96
> * * TimesNet: 96, 36(for ili dataset)
> * * PatchTST, DLinear: 336, 104 (for ili dataset)
> * **lambda values** The values for lambdas in Table 1 represent the best results achieved from a grid search over the hyper-parameter space defined in Appendix G. Reporting the best performance found via a search is a standard practice in research [Learning to Forecast in the Frequency Domain, Shape and Time Distortion Loss, etc.].
>
> ---
> [W3,8] Regarding the important question about the relationship between sampling frequency and planning cycles, this is critical because it directly impacts operational stability and user trust. We argue that high-frequency forecast consistency remains crucial even for coarser planning cycles for the following reasons:
> * **Impact on Trust and Operational Monitoring** Even with daily planning, volatile hourly forecasts for the same future period erode planners' confidence and can trigger unnecessary monitoring alarms.
> * **Evidence of Real-World Costs** This instability is not theoretical; it leads to significant, real-world costs. This is a well-documented industrial and economical phenomenon known as MRP Nervousness, where frequent re-planning ultimately incurs substantial economic losses (e.g., sunk and opportunity costs). AliO directly mitigates this costly issue.
> * **Importance of Mid-Term Alignment** Planning considers the entire forecast period, not just the final endpoint. The high short- and mid-term misalignment shown in Figure 2 is precisely the problem that erodes trust and forces costly re-evaluations, even if the predictions eventually converge.
>
> As you mentioned, not all methodologies may be important in every situation, and AliO can be appropriately used in situations like ILI mentioned by the reviewer.
>
> ---
> [W4,19] Regarding the important questions about our proposed metric, TAM is necessary because it is specifically designed to measure prediction consistency, a dimension distinct from the accuracy measure by regression metrics. We chose MAE over DTW as the default distance function to directly address the practical need for point-wise stability.
> * **Necessity of TAM (vs. regression metrics)** While regression metrics like MSE measure accuracy (prediction vs. ground truth), TAM is essential for measuring consistency (prediction vs. prediction). This allows us to distinguish between a model that is accurate but unreliable (volatile) and a model that is both accurate and robust. (Table 1).
> * **MAE vs. DTW** We use MAE as the default because our goal is to ensure point-wise consistency — the stability of the predicted value for a specific future timestamp, which is critical for planning. DTW, which measures shape similarity by warping time, is unsuited for this specific, time-aligned problem and could distort the direct value difference.
> * **Practicality and Future Work** MAE is also far more computationally efficient than DTW. However, we agree that a DTW-based TAM could be a valuable tool for the different goal of measuring shape consistency, which we consider a promising future research direction.
>
> ---
> [W9] We agree that regression loss provides some implicit alignment but argue this is often insufficient. RegPull mechanism is designed to enforce stronger consistency while preventing the error propagation the reviewer rightly pointed out.
> * **Limits of Implicit Alignment** While regression loss provides some alignment, it doesn’t guarantee consistency. The non-zero baseline TAM values in Table are the evidence. We posit that AliO provides the proper training the reviewer alluded to by adding the necessary explicit loss for consistency while maintaining or enhancing accuracy.
> * **Preventing Error Propagation with RegPull** RegPull mechanism is specifically designed to prevent error propagation. At each timestamp, it identifies the more accurate prediction (teacher) and only pulls the less accurate one (student) towards it. This one-way correction is enforced by applying a stop-gradient to the teacher (point-wise), which blocks any gradient flow from the student back to the teacher. Therefore, a poor prediction cannot corrupt a good one, ensuring the alignment process constructively reinforces the main accuracy objective.
>
> ---
> [W13] We thank the reviewer for these valuable suggestions, which allow us to both address concerns about overgeneralization from our main paper's heatmaps and clarify the indicidual and synergistic effects. To address the valid concern about Figures 6 and 7, we will add a comprehensive set of individual heatmaps for each model and dataset to the appendix. This detailed view also allows us to clarify the role of each component. We provide additional result as below (X-axis: $\lambda_F$, Y-axis: $\lambda_T$):
>
> |TimesNet-ETTm1 (MSE)|0|0.5|1|2|
> |-|-|-|-|-|
> |5|0.01|0.01|0|0.01|
> |2|0.06|0.06|0.05|0.06|
> |1|0.11|0.11|1|0.12|
> |0|-|0.11|0.12|0.14|
>
> |TimesNet-ETTm1 (TAM)|0|0.5|1|2|
> |-|-|-|-|-|
> |5|0|0|0.01|0|
> |2|0.33|0.33|0.33|0.33|
> |1|0.55|0.54|0.46|0.55|
> |0|-|1|1|1|
>
> * **Individual Component Analysis** As the reviewer proposed, the impact of each component can be seen by setting the other to zero. The effect of the time domain alone is visible in Figure 6. Similarly, the effect of the frequency domain alone is demonstrated in the heatmaps in Appendix E (single model).
> * **Synergistic Effect** The optimal accuracy in Figure 6 (i.e., lowest MSE) is achieved when both time and frequency coefficients are non-zero. This experimentally demonstrates that the two domains contain complementary information and create a synergistic effect to maximize prediction accuracy when used together. This finding aligns with empirical support for the ideas presented in prior works such as [Learning to Forecast in the Frequency Domain].
>
> To make this point clearer for all readers, we will add a sentence about Figure 6 in the main paper and present the individual effect heatmaps as in Appendix E.

---

> > ### Author Response · Authors · 2025-08-05
> >
> > Dear Reviewer,
> >
> > Thank you for your valuable feedback on our paper.
> >
> > We have submitted our rebuttal, in which we provided a thorough response to the comments you raised.
> > However, we notice that the discussion period ends tomorrow, and we have not yet received a response.
> >
> > We understand that you are taking the time for careful consideration, and we appreciate your diligence. Please let us know if we can provide any further information or clarification.
> >
> > Thank you for your time and consideration.
> >
> > Best regards,

---

> ### Author Response · Authors · 2025-08-05
> **Additional data for Figure 6.**
>
> For Figure 6, the performance when lambda is 0 is as follows.
>
> In general, it was observed that TAM is worst when the $\lambda_T=0$. For MSE, performance improves with an appropriate combination of time and frequency coefficients. This is consistent with the argument in the paper [Learning to Forecast in the Frequency Domain], which claims that performance is enhanced when the time and frequency domains are used together.
>
> ---
>
> ## TimesNet
>
> |ETTh1 MSE|$\lambda_F$=0|0.5|1|2|
> |-|-|-|-|-|
> |$\lambda_T$=5|0.16|0.15|0.16|0.15|
> |2|0.11|0.08|0.09|0|
> |1|0.05|0.04|1|0.04|
> |0|-|0.03|0.04|0.04|
>
> |ETTh1 TAM|$\lambda_F$=0|0.5|1|2|
> |-|-|-|-|-|
> |$\lambda_T$=5|0|0|0|0|
> |2|0.25|0.25|0.25|0.26|
> |1|0.46|0.46|0.43|0.47|
> |0|-|1|0.99|0.98|
>
> ---
>
> |ETTm2 MSE|$\lambda_F$=0|0.5|1|2|
> |-|-|-|-|-|
> |$\lambda_T$=5|0.03|0.04|0.06|0|
> |2|0.1|0.16|0.08|0.08|
> |1|0.14|0.12|1|0.11|
> |0|-|0.29|0.24|0.22|
>
> |ETTm2 TAM|$\lambda_F$=0|0.5|1|2|
> |-|-|-|-|-|
> |$\lambda_T$=5|0|0|0.02|0.01|
> |2|0.24|0.26|0.26|0.25|
> |1|0.45|0.44|0.42|0.43|
> |0|-|1|1|0.97|
>
> ---
>
> |Traffic MSE|$\lambda_F$=0|0.5|1|2|
> |-|-|-|-|-|
> |$\lambda_T$=5|0.05|0.06|0.03|0.06|
> |2|0.01|0.02|0.02|0|
> |1|0.04|0.03|1|0|
> |0|-|0.44|0.07|0.06|
>
> |Traffic TAM|$\lambda_F$=0|0.5|1|2|
> |-|-|-|-|-|
> |$\lambda_T$=5|0.01|0|0.09|0.04|
> |2|0.27|0.3|0.26|0.29|
> |1|0.52|0.48|0.28|0.5|
> |0|-|0.88|1|0.90|
>
> ---
>
> ## iTransformer
>
>
> |ETTh1 MSE|$\lambda_F$=0|0.5|1|2|
> |-|-|-|-|-|
> |$\lambda_T$=5|0|0|0|0|
> |2|0.17|0.17|0.17|0.17|
> |1|0.93|0.93|0.57|0.93|
> |0|-|1|1|0.99|
>
> |ETTh1 TAM|$\lambda_F$=0|0.5|1|2|
> |-|-|-|-|-|
> |$\lambda_T$=5|0|0|0|0|
> |2|0.24|0.24|0.24|0.23|
> |1|0.46|0.46|0.63|0.46|
> |0|-|1|0.99|0.98|
>
> ---
>
> |ETTm1 MSE|$\lambda_F$=0|0.5|1|2|
> |-|-|-|-|-|
> |$\lambda_T$=5|0.1|0.1|0.1|0.1|
> |2|0|0|0|0|
> |1|0.2|0.2|0.37|0.2|
> |0|0|1|1|0.99|
>
> |ETTm1 TAM|$\lambda_F$=0|0.5|1|2|
> |-|-|-|-|-|
> |$\lambda_T$=5|0|0|0|0|
> |2|0.25|0.25|0.25|0.25|
> |1|0.46|0.46|0.59|0.45|
> |0|-|1|0.99|0.98|
>
> ---
>
> |Traffic MSE|$\lambda_F$=0|0.5|1|2|
> |-|-|-|-|-|
> |$\lambda_T$=5|0.99|0.99|0.99|1|
> |2|0.21|0.2|0.21|0.2|
> |1|0.03|0.01|0.35|0|
> |0|-|0.33|0.35|0.33|
>
> |Traffic TAM|$\lambda_F$=0|0.5|1|2|
> |-|-|-|-|-|
> |$\lambda_T$=5|0|0|0|0|
> |2|0.21|0.21|0.21|0.21|
> |1|0.42|0.41|0.51|0.41|
> |0|-|1|0.99|0.98|
>
> ---
>
> |Weather MSE|$\lambda_F$=0|0.5|1|2|
> |-|-|-|-|-|
> |$\lambda_T$=5|0.97|0.95|0.91|1|
> |2|0.14|0.2|0.12|0.16|
> |1|0.12|0.06|0|0.02|
> |0|-|0.52|0.42|0.4|
>
> |Weather TAM|$\lambda_F$=0|0.5|1|2|
> |-|-|-|-|-|
> |$\lambda_T$=5|0|0.01|0|0.01|
> |2|0.27|0.27|0.26|0.26|
> |1|0.47|0.46|0.58|0.46|
> |0|-|1|0.99|0.96|
>
> ## DLinear
>
> |ETTh1 MSE|$\lambda_F$=0|0.5|1|2|
> |-|-|-|-|-|
> |$\lambda_T$=5|0|0|0|0|
> |2|0.16|0.16|0.16|0.16|
> |1|0.25|0.24|1|0.24|
> |0|-|0.37|0.37|0.37|
>
> |ETTh1 TAM|$\lambda_F$=0|0.5|1|2|
> |-|-|-|-|-|
> |$\lambda_T$=5|0|0|0|0|
> |2|0.35|0.34|0.34|0.34|
> |1|0.57|0.57|0.6|0.57|
> |0|-|1|1|0.98|
>
> ---
>
> |ETTm1 MSE|$\lambda_F$=0|0.5|1|2|
> |-|-|-|-|-|
> |$\lambda_T$=5|0|0.18|0.18|0.18|
> |2|0.62|0.62|0.62|0.61|
> |1|0.83|0.82|1|0.82|
> |0|-|0.89|0.89|0.88|
>
> |ETTm1 TAM|$\lambda_F$=0|0.5|1|2|
> |-|-|-|-|-|
> |$\lambda_T$=5|0|0.01|0.01|0.01|
> |2|0.31|0.31|0.31|0.31|
> |1|0.53|0.53|0.47|0.52|
> |0|-|1|1|0.98|
>
> ---
>
> |Traffic MSE|$\lambda_F$=0|0.5|1|2|
> |-|-|-|-|-|
> |$\lambda_T$=5|0.48|0.48|0.48|0.48|
> |2|0.62|0.62|0.62|0.62|
> |1|0.75|0.75|0|0.76|
> |0|-|1|1|1|
>
> |Traffic TAM|$\lambda_F$=0|0.5|1|2|
> |-|-|-|-|-|
> |$\lambda_T$=5|0|0|0|0|
> |2|0.37|0.37|0.37|0.37|
> |1|0.59|0.59|0.45|0.59|
> |0|-|1|1|0.99|
>
>
> ---
>
> |Weather MSE|$\lambda_F$=0|0.5|1|2|
> |-|-|-|-|-|
> |$\lambda_T$=5|0.44|0.44|0.44|0.44|
> |2|0.25|0.25|0.25|0.25|
> |1|0.15|0.15|1|0.15|
> |0|-|0|0|0.01|
>
> |Weather TAM|$\lambda_F$=0|0.5|1|2|
> |-|-|-|-|-|
> |$\lambda_T$=5|0|0|0|0|
> |2|0.24|0.24|0.24|0.24|
> |1|0.46|0.46|0.65|0.45|
> |0|-|1|0.99|0.98|

---

> > ### Comment · Reviewer_J5tA · 2025-08-06
> >
> > Sorry for the late reply and thank you for your detailed explanations and the additional experiments/results. I appreciate the time you've taken to address these points.
> >
> > ## Clarification Requests
> >
> > Regarding Figure 4 and the correlation between TAM and MSE improvement:
> > > As shown in Figure 4, there is a positive correlation between a high initial TAM (poor initial alignment) and greater MSE improvement
> >
> > Could you please clarify:
> > - Is Figure 4 based on a single dataset or multiple datasets?
> > - Does each point in the figure represent:
> >   - A single dataset, but different runs?
> >   - One run of a given dataset?
> >
> > ## Discussion on Loss, Alignment, and Practical Benefits
> >
> > While I understand your perspective that:
> > > We agree that regression loss provides some implicit alignment but argue this is often insufficient
> >
> > I remain concerned about the practical significance of the MSE improvements achieved with AliO, particularly for LTSF datasets where:
> > - The gains appear minimal (primarily affecting the third decimal place)
> > - These improvements must be weighed against the computational overhead introduced by AliO (cf. latest point)
> >
> > In my understanding, for these (LTSF) fine-grained datasets (hourly or XX-minute resolution), the differences between consecutive samples ($l=1$) may mostly be insignificant (plotting distribution of samples difference for each dataset, different input length and different lags could greatly help readers confirm this point or discard it). In addition, as I mentioned previously, current loss induces some kind of alignment and Figure 9-11 demonstrates that without AliO, when advancing the sampling window by one step, predictions already show good alignment, and results show minor improvement in MSE.
> >
> > I hypothesize that AliO's benefits might become more apparent either with larger sampling lags which will introduce more differences in input or with dataset using coarser resolution (such as ILI, as demonstrated by Authors' results).
> > The additional experiments provided by Authors in the response to reviewer 5d8j (with different lags and number of samples) do not confirm this hypothesis, but these experiments are only done on ETTh1, therefore it doesn't provide sufficient evidence to confirm this trend across any datasets (more comprehensive testing across multiple datasets would be necessary to be able to generalize this. In addition, Authors posit that the main factor for misalignment in predictions is the differences between consecutive input samples. Therefore, Authors should confirm this point with input sample difference distribution).
> >
> > ## Implementation Details
> >
> > Regarding your clarification about model architecture:
> > > We will clarify in the paper that our claim refers to the model's architecture (e.g., Transformer blocks, forward function), which remains entirely unchanged
> >
> > While I understand the backbone architecture remains unchanged, AliO does appear to modify:
> > 1. The sampling process for training data to ensure having consecutive samples
> > 2. The training function (e.g., train from Exp_Main in DLinear) by incorporating TAM alongside the standard loss metric (MSE) in order to guide training to ensure prediction alignment
> >
> > ## Number of samples compared using TAM
> >
> > What is the exact relationship between $N$ (number of samples compared using TAM) and batch size?
> >   - Is $N$ typically smaller than batch size?
> >   - Or equal to batch size?
> > Larger $N$ seemingly provide performance, but does $N$ influence computation cost during training?
> >
> > ## Computational Cost Analysis
> >
> > You mentioned in your response to reviewer HfaX:
> > > Zero Inference Overhead: AliO is applied as a loss function only during the training phase. Therefore, at inference time, the model operates as a standard single model, making its deployment pipeline and computational cost completely identical to the baseline
> >
> > While the zero-inference overhead is clear, I would appreciate more details about the specific computational overhead introduced by AliO during training?
> > This would help discuss the trade-off between the computational cost and the observed improvements in MSE.
> > (Such information can serve as guidelines for practitioners about when the benefits might justify the additional training costs)

---

> > > ### Author Response · Authors · 2025-08-06
> > >
> > > Thank you very much for your detailed and insightful early review. Your thorough feedback has been invaluable in refining our work, and we greatly appreciate the time and effort you invested.
> > >
> > > # Regarding Clarification Requests:
> > > Figure 4 is based on multiple datasets. Each point in the plot represents the average performance for a single model on a single dataset, averaged over all prediction lengths and three random seeds.
> > >
> > > # Regarding Discussion on Loss, Alignment, and Practical Benefits
> > > We thank the reviewer for these detailed points on practical benefits and the nature of the input data. We argue that even numerically small MSE improvements have significant practical value in high-stakes systems.
> > >
> > > * Practical Significance of Minimal MSE Gains:
> > > > While some MSE improvements may appear small in isolation, their practical impact in high-stakes planning environments is significant. In complex value chains governed by phenomena like MRP Nervousness or the Bullwhip Effect, even minor forecast stabilization can prevent extremely costly re-planning cycles and generate substantial downstream savings. The goal is to reduce the frequency of these high-cost events.
> > >
> > > The following pertains to the performance improvement averaged over all prediction lengths and three random seeds.
> > >
> > > |Model|MSE(Mean)|TAM(Mean)|
> > > |-|-|-|
> > > |Autoformer|7.0%|24.2%|
> > > |DLinear|5.6%|36.2%|
> > > |PatchTST|2.6%|35.7%|
> > > |TimesNet|3.4%|33.4%|
> > > |iTransformer|6.6%|28.0%|
> > > |GPT4TS|2.3%|32.1%|
> > >
> > > * The Model's Reaction to Small Input Differences
> > >
> > > > The reviewer's suggestion to analyze the distribution of differences between consecutive samples is good feedback. Since it is not possible to attach media files in the review system, this will be incorporated in the camera-ready version.
> > >
> > > > Generally, the data undergoes normalization, such as through RevIN, before being fed into the backbone model. In other words, the issue is not that the inputs differ significantly, but rather that the baseline model is overly sensitive to statistically minor input variations, resulting in unstable predictions (as demonstrated by the initial TAM and high MSE). AliO is a methodology designed to regulate this hypersensitivity and train the model to be more robust against such subtle fluctuations.
> > >
> > > > The following are the results for the exchange data with daily resolution. For the exchange rate data, the baseline TAM ranged from about 0.02 to 0.04, which is relatively low. Therefore, rather than the coarser resolution itself, the limitation of the baseline model seems to lie in its unstable predictions (high MSE, high TAM) in response to the inputs.
> > > |Model|MSE(Mean)|
> > > |-|-|
> > > |Autoformer|11.0%|
> > > |DLinear|3.9%|
> > > |TimesNet|3.4%|
> > >
> > > # Implementation Details
> > > As you mentioned, we have modified the training pipeline, and this is the same approach we adopted for the learning method that receives overlapping data, as shown below. Although we modified the pipeline, the model itself was not changed; therefore, this can be considered a model-agnostic approach.
> > >
> > > * Unsupervised Representation Learning for Time Series with Temporal Neighborhood Coding
> > > * TimeTS2Vec: Towards Universal Representation of Time Series
> > > * Multi-Task Self-Supervised Time-Series Representation Learning
> > > * Self-supervised Learning for Electroencephalogram: A Systematic Survey
> > >
> > > # Number of samples compared using TAM
> > > The precise relationship is that the effective batch size becomes original_batch_size * N, as the model's forward pass is called N times. Therefore, the computation cost increases linearly with N.
> > >
> > >
> > >
> > > # Computational Cost Analysis
> > > > We fully agree with the reviewer. In our experiments with N=2, the training time per epoch increased by approximately x1.6 (TimesNet), x2.0 (PatchTST), x1.6(iTransformer) on average, even with parallel processing. While we believe the zero inference overhead holds greater practical value, practitioners can use this training cost increase to evaluate the trade-off. We will add a more detailed analysis to the camera-ready version.

---

> > > > ### Author Response · Authors · 2025-08-06
> > > > **Additional Data for Figure 7 [1st]**
> > > >
> > > > Below are inference results of the DLinear model on the ETTh2 dataset, with prediction lengths of 96, 192, 336, and 720.
> > > > The results show that, consistent with our findings in Section 5.3, TAM performance is generally better with a small lag and a large N.
> > > >
> > > > |ETTh2 MSE 96|1|2|4|8|16|32|
> > > > |-|-|-|-|-|-|-|
> > > > |16|0.55|0.30|0.89|-|-|-|
> > > > |8|0.27|0.19|0.16|0.29|-|-|
> > > > |4|0.58|0.32|0.62|0.74|0.55|-|
> > > > |2|0.49|0.02|0.00|0.80|0.56|1.00|
> > > >
> > > > |ETTh2 TAM 96|1|2|4|8|16|32|
> > > > |-|-|-|-|-|-|-|
> > > > |16|0.03|0.59|0.00|-|-|-|
> > > > |8|0.23|0.38|0.65|0.03|-|-|
> > > > |4|0.14|0.17|0.28|1.00|0.30|-|
> > > > |2|0.33|0.27|0.28|0.09|0.43|0.30|
> > > >
> > > > |ETTh2 MSE 192|1|2|4|8|16|32|48|64|
> > > > |-|-|-|-|-|-|-|-|-|
> > > > |16|1.00|0.09|0.76|0.97|-|-|-|-|
> > > > |8|0.33|0.34|0.92|0.00|0.35|-|-|-|
> > > > |4|0.46|0.21|0.27|0.33|0.72|0.64|-|-|
> > > > |2|0.67|0.67|0.62|0.32|0.65|0.87|0.43|0.73|
> > > >
> > > > |ETTh2 TAM 192|1|2|4|8|16|32|48|64|
> > > > |-|-|-|-|-|-|-|-|-|
> > > > |16|0.21|0.07|0.19|0.00|-|-|-|-|
> > > > |8|0.12|0.23|0.20|0.70|0.25|-|-|-|
> > > > |4|1.00|0.45|0.50|0.65|0.65|0.28|-|-|
> > > > |2|0.12|0.05|0.06|0.33|0.35|0.19|0.43|0.17|
> > > >
> > > > |ETTh2 MSE 336|1|2|4|8|16|32|48|64|
> > > > |-|-|-|-|-|-|-|-|-|
> > > > |16|0.85|0.76|0.13|0.09|0.93|-|-|-|
> > > > |8|0.23|0.83|0.81|0.34|0.00|0.31|-|-|
> > > > |4|0.39|0.55|0.77|0.61|0.98|0.69|0.36|-|
> > > > |2|0.72|0.82|0.80|0.69|0.59|0.88|0.89|1.00|
> > > >
> > > > |ETTh2 TAM 336|1|2|4|8|16|32|48|64|
> > > > |-|-|-|-|-|-|-|-|-|
> > > > |16|0.00|0.21|0.34|1.00|0.38|-|-|-|
> > > > |8|0.05|0.41|0.13|0.77|0.38|0.99|-|-|
> > > > |4|0.16|0.16|0.35|0.09|0.04|0.18|0.45|-|
> > > > |2|0.33|0.34|0.09|0.32|0.72|0.11|0.20|0.07|
> > > >
> > > > |ETTh2 MSE 720|1|2|4|8|16|32|48|64|
> > > > |-|-|-|-|-|-|-|-|-|
> > > > |16|0.48|0.64|0.40|0.14|0.73|0.73|-|-|
> > > > |8|0.23|0.79|0.55|0.30|1.00|0.29|0.10|0.37|
> > > > |4|0.00|0.69|0.62|0.67|0.74|0.43|0.56|0.50|
> > > > |2|0.59|0.59|0.58|0.48|0.44|0.54|0.78|0.63|
> > > >
> > > > |ETTh2 TAM 720|1|2|4|8|16|32|48|64|
> > > > |-|-|-|-|-|-|-|-|-|
> > > > |16|0.24|0.12|0.07|0.06|0.02|0.03|-|-|
> > > > |8|0.23|0.17|0.00|0.21|0.68|0.31|1.00|0.24|
> > > > |4|0.27|0.06|0.28|0.48|0.00|0.09|0.18|0.36|
> > > > |2|0.15|0.10|0.08|0.08|0.19|0.43|0.28|0.20|

---

> > > > > ### Comment · Reviewer_J5tA · 2025-08-07
> > > > >
> > > > > I'm glad my comments have been helpful in refining your work. And I appreciate your efforts in addressing them.
> > > > >
> > > > > ## Computational cost
> > > > > I was expecting the cost to decrease with greater $N$ as you will have less TAM to compute, but as batch_size increase it is not the case, I see.
> > > > >
> > > > > ## Suggestions for Enhancing Figure 4
> > > > >
> > > > > Regarding your explanation:
> > > > > > Each point in the plot represents the average performance for a single model on a single dataset
> > > > >
> > > > > To make this visualization more informative, I suggest the following enhancements:
> > > > >
> > > > > 1. **Color Coding**:
> > > > >    - Use different colors for points corresponding to different datasets
> > > > >    - Plot regression lines for each dataset using matching colors (while keeping the regression lines considering all points)
> > > > >
> > > > > 2. **Model Differentiation**: Use distinct markers for each model type
> > > > >
> > > > > Such an enhanced visualization would provide valuable insights. In fact, it could help readers better understand model sensitivity. If a model is particularly sensitive as you suggest, it would likely show high initial TAM values. The plot could also reveal whether this sensitivity stems from the model architecture (consistent across datasets) or is dataset-dependent. Being able to extract such conclusions from the plot would significantly strengthen your work.
> > > > >
> > > > >
> > > > > ## Importance of Input Variation Distribution
> > > > >
> > > > > Regardless of whether we term it "input sample difference" or "input variations", analyzing these distributions serves an important purpose. It provides evidence to support your claims about sensitivity to minor input variations, complementing the enhanced Figure 4 analysis.
> > > > >
> > > > > I believe presenting these variation distributions per dataset and for different numbers of samples ($N$) would be valuable additions to your paper. This information could form part of practical guidelines for future readers, helping them determine when AliO might be particularly beneficial. By examining their own datasets' input variation distributions, along with the sensitivity of the model they are planning to use as well as their forecasting objectives, readers could better assess whether AliO would be crucial for their specific applications.
> > > > >
> > > > > ## Addressing Practicality Concerns
> > > > >
> > > > > While I understand your perspective on the importance of forecast consistency, I think most reviewers' concerns about practical applications stems from the current version of the paper that lacks mentioned to the reports supporting the need for precise forecasting alignment. Furthermore, there is some inconsistency between the models mentioned in these references to reports (such as those from NOAA) with those tested in your study, (the report I have checked primarily focus on statistical methods like linear mixed regression).
> > > > >
> > > > > To create a more compelling narrative, consider explicitly referencing these reports and demonstrating that:
> > > > > - Baseline statistical methods show sensitivity to input variations
> > > > > - Advanced models, even without AliO, exhibit similar sensitivity (as shown by initial TAM values)
> > > > > - AliO effectively addresses this sensitivity in advanced models
> > > > > - Your solution outperforms previous latent space alignment approaches
> > > > >
> > > > > Following this path would better establish the problem you're addressing and clearly position AliO as an effective solution. It would demonstrate how your method improves upon both traditional statistical approaches and recent alignment techniques, providing a more complete picture of its advantages.

---

> > > > > > ### Author Response · Authors · 2025-08-07
> > > > > >
> > > > > > We are sincerely grateful to the reviewers for their valuable and constructive feedback. Your suggestions have been instrumental in helping us strengthen our paper. We are committed to incorporating all suggested revisions into the camera-ready version.
> > > > > >
> > > > > > # Regarding Figure 4
> > > > > > Thank you for the insightful feedback on our work. We agree that the suggested improvements to Figure 4 will provide crucial insights that more effectively support our claims. As you suggested, by distinguishing between datasets and model types in the visualization, we can more clearly determine whether a model's sensitivity is due to its architecture or is dataset-dependent.
> > > > > >
> > > > > > We will actively incorporate these enhancements into the final camera-ready version to help readers better understand and be persuaded by the core arguments of our research.
> > > > > >
> > > > > > # On Strengthening the Motivation with Real-World Evidence
> > > > > > We thank the reviewer for their highly constructive feedback on strengthening our paper's narrative. Following your advice, we will revise Section 2 (Motivation) in the camera-ready version. We will explicitly cite practitioner reports (e.g., from NOAA) to first establish that forecast instability is a significant, long-standing problem, even for the traditional statistical models used in these domains. This will ground the importance of the consistency problem in real-world practice, a property we argue will be essential for the future adoption of deep learning models in these critical industries.
> > > > > >
> > > > > > # On Differentiating from Contrastive Learning
> > > > > > We will also expand our Related Work section to more clearly position AliO. AliO and latent space alignment methods like Contrastive Learning are not direct competitors. Contrastive learning is typically used for pre-training without ground truth, focusing on representation, where regression performance is not guaranteed. In contrast, AliO is a full training methodology that operates in the output space alongside a regression loss. This ensures the alignment process is always anchored to the ground truth, making the two approaches complementary.
> > > > > >
> > > > > > # On Analyzing the Input Variation Distribution
> > > > > > Thank you for the excellent suggestion, which will significantly strengthen our paper. We will add an analysis of the input variation distribution (e.g., histograms of sample differences) for key datasets (e.g., ETT, ILI) to the appendix. This will provide quantitative evidence for our core argument that baseline models are oversensitive to minor input variations (leading to a high initial TAM). Furthermore, as you pointed out, this analysis will serve as a practical guideline for future readers to determine when AliO might be particularly beneficial for their applications.
> > > > > >
> > > > > > Thank you again for your time and effort.

---

### Note · Authors · 2025-08-12

We sincerely thank all the reviewers for their thorough and insightful feedback. Your comments have been instrumental in helping us clarify the core of our research and significantly strengthen the paper. We are committed to faithfully incorporating all the improvements discussed during the rebuttal period into the final camera-ready version to present a more complete and polished paper.

During this rebuttal period, we focused on demonstrating the practical utility of Alignment, a common question from the reviewers. To this end, we explained how prediction inconsistency leads to the rolling forecast dilemma, creating sunk and opportunity costs that amplify into real industrial problems like MRP Nervousness and the Bullwhip Effect, based on established economic theories. We also conducted additional experiments to provide results for a broader range of hyperparameters.

Methodologically, we showed that AliO's effectiveness is not just an empirical improvement but can be explained by the theoretical principle of knowledge distillation. Furthermore, we demonstrated that improving alignment performance on average contributes to improving regression performance, revealing a positive relationship between reliability and accuracy. We clarified the unique role of the TAM metric, justified our dual-domain (time and frequency) alignment with recent literature, and positioned AliO as a complementary tool - not a competitor or baseline - to other powerful methods like Ensembles, Bayesian approaches, WQL, and Contrastive Learning. This entire process was shared transparently through our fully public code, including all training parameters, reconfirming that AliO is a model-agnostic method that requires no changes to the backbone architecture.

Finally, our entire research process was guided by the following fundamental questions:

* Can a model whose predictions for the same future point fluctuate erratically with minimal new data truly be considered ideal?

* Can a user trust a system whose predictions are constantly volatile?

* Given two systems with equal accuracy, will a user not always prefer the stable system over the unstable one?

We believe AliO can be one answer to these questions, and we trust that our detailed rebuttal, supported by sufficient references and new experimental results, has successfully demonstrated the necessity of our methodology.

We thank you once again for your valuable time and for helping us improve our paper.

Best regards,

---

### Decision · Program_Chairs · 2025-09-17

**Decision:**

Accept (poster)

**Comment:**

The paper identifies the problem of inconsistent predictions across overlapping forecasting windows in long-term time series forecasting (LTSF), arguing that such inconsistencies reduce reliability in real-world deployment. To address this, this paper proposes AliO, a method that enforces consistency through “regression pulling” and joint time–frequency alignment. It also introduces a new evaluation metric, TAM, to measure alignment quality. Experiments across several architectures (e.g., Autoformer, PatchTST, TimesNet, GPT4TS) and datasets are presented, with results suggesting AliO improves both predictive accuracy and alignment.

Strengths: This paper addresses a novel and underexplored issue in LTSF, by introducing a new evaluation metric (TAM) that complements existing error-based measures. Experimental results have demonstrated broad applicability via evaluating across multiple models and datasets. This paper provides mathematical formulations and reasonably clear writing with illustrative figures.

Weaknesses: There are concerns about the practical necessity of strict prediction consistency, which is not convincingly justified; reviewers question whether variability may instead reflect appropriate adaptation to new data. The “regression pulling” and stop-gradient mechanisms appear heuristic, with limited theoretical grounding. There are also concerns about the missing evaluation against recent pretrained foundation models, and unclear generalization across different lag values. The TAM metric’s interpretability and necessity relative to established metrics remain underexplored, with limited validation (e.g., user studies).

Rebuttal: The rebuttal provided clarifications to these concerns but did not fully resolve concerns ---- the real-world necessity of strict consistency was not substantiated, the design choices remained ad hoc, and the scope of baselines remained limited. As a result, reviewers maintained reservations, and after weighing these points, I conclude that the weaknesses outweigh the strengths.

Summary:  Despite the above limitations, this paper addresses an interesting and less studied issue in time series forecasting and offers a practical method with encouraging empirical results. While some aspects of the methodology are heuristic and the broader impact is not yet fully proven, the novelty of the problem formulation, the introduction of a new evaluation metric, and the experimental evidence across diverse models make this work a valuable contribution to the community. These strengths outweigh the shortcomings.